# Organic NIR-II molecule with long blood half-life for in vivo dynamic vascular imaging

Benhao Li[1,6], Mengyao Zhao[1,6], Lishuai Feng[2], Chaoran Dou[3], Suwan Ding [1], Gang Zhou [4], Lingfei Lu[1], Hongxin Zhang[1], Feiya Chen[1], Xiaomin Li [1], Guangfeng Li[4], Shichang Zhao[5], Chunyu Jiang[2], Yan Wang[3], Dongyuan Zhao [1], Yingsheng Cheng[2] & Fan Zhang [1✉]

Real-time monitoring of vessel dysfunction is of great significance in preclinical research. Optical bioimaging in the second near-infrared (NIR-II) window provides advantages including high resolution and fast feedback. However, the reported molecular dyes are hampered by limited blood circulation time (~ 5–60 min) and short absorption and emission wavelength, which impede the accurate long-term monitoring. Here, we report a NIR-II molecule (LZ-1105) with absorption and emission beyond 1000 nm. Thanks to the long blood circulation time (half-life of 3.2 h), the fluorophore is used for continuous real-time monitoring of dynamic vascular processes, including ischemic reperfusion in hindlimbs, thrombolysis in carotid artery and opening and recovery of the blood brain barrier (BBB). LZ-1105 provides an approach for researchers to assess vessel dysfunction due to the long excitation and emission wavelength and long-term blood circulation properties.

[1] Department of Chemistry, State Key Laboratory of Molecular Engineering of Polymers, Shanghai Key Laboratory of Molecular Catalysis and Innovative Materials and iChem, Fudan University, Shanghai 200433, PR China. [2] Department of Radiology, Shanghai Jiao Tong University Affiliated Sixth People's Hospital, 600 Yishan Road, Shanghai 200233, PR China. [3] Department of Ultrasound in Medicine, Shanghai Institute of Ultrasound in Medicine, Shanghai Jiao Tong University Affiliated Sixth People's Hospital, 600 Yishan Road, Shanghai 200233, PR China. [4] Lab of Advanced Materials & Department of Macromolecular Science, Collaborative Innovation Center of Chemistry for Energy Materials, Fudan University, Shanghai 200438, PR China. [5] Department of Orthopedics, Shanghai Sixth People's Hospital, Shanghai Jiao Tong University, 600 Yishan Road, Shanghai 200233, PR China. [6] These authors contributed equally: Benhao Li, Mengyao Zhao. ✉email: zhang_fan@fudan.edu.cn

A healthy and functional vasculature plays the important role of delivering nutrients to cells and protecting organs[1]. Real-time detection of vascular dysfunction in small animals is of great significance in preclinical biomedical research[2]. However, obtaining vascular dynamic information requires more than taking a static measurement at a single timepoint, it also requires the ability to noninvasively monitor functional dynamic processes in real-time with high spatial and temporal resolution[3]. Optical bioimaging exhibits advantages of fast feedback and nonionizing radiation[4]. Although the optical imaging penetration depth is limited to several centimeters, continuously monitoring dynamic physical process of small animals would be helpful to broaden the understanding of vascular dysfunction and recovery process[5]. Compared with the traditional near-infrared window (780–900 nm), recently developed second near-infrared window (NIR-II; 1000–1700 nm) fluorophores have received considerable attention for noninvasive in vivo imaging because of higher resolution and signal-to-background ratio (SBR)[5–17]. Unfortunately, widely used inorganic NIR-II contrast agents such as single-walled carbon nanotubes[18,19], quantum dots[11,20,21], and rare-earth doped downconversion nanoparticles[22–28] hold unknown long-term toxicity concerns. Thus development of organic small-molecular NIR-II fluorophores for in vivo imaging is in urgent demand[29–34]. Several organic NIR-II contrast agents have been designed for in vivo bioimaging. However, due to the short blood circulation half-life time (5–60 min) (Supplementary Table 1)[35–38], continuous monitoring of dynamic physical processes is problematic.

Herein, we report the small molecule LZ-1105 as a NIR-II probe for long-term in vivo imaging of dynamic vascular structure changes in small animals. LZ-1105 exhibits a high aqueous solubility, a peak fluorescent emission at 1105 nm and a molecular mass of 1.5 kDa. Meanwhile, LZ-1105 shows high resolution and SBR for noninvasive imaging of mouse cerebral vasculatures, hindlimb vasculatures, and lymphatic vasculatures. Significantly, pharmacokinetics of LZ-1105 demonstrates a half-life of over 3 h in blood circulation, which provides a long imaging window for long-term monitoring of dynamic vascular structure changes. LZ-1105 is capable of real-time quantification of femoral artery blood velocity in ischemic perfusion hindlimbs, thrombolysis process in carotid arteries, and the temporary opening and recovery of the blood–brain barrier (BBB) in living mice. Thus, NIR-II imaging with the long-term angiography probe LZ-1105 in small animals might help researchers to better understand dynamic vascular processes.

## Results

**Synthesis and characterization of LZ dyes for imaging**. The LZ series dyes were easily synthesized in two steps with overall yields of 85–92% from commercial available benzopyrrole derivatives (Fig. 1a). Alkylated reaction and Vilsmeier–Haack reaction were key steps during the synthesis process (Supplementary Fig. 1). To enhance the water solubility of the LZ dyes, four sulfonic groups were introduced into the chemical structures. All the LZ dyes displayed the maximal absorptions and emissions in different solvents within the NIR-II window in the range of 1003–1118 nm (Fig. 1b, Supplementary Figs. 2–5, and Supplementary Table 2). Besides, the high molar extinction coefficients ($\varepsilon_{max}$, 1.01–1.99 × $10^5$ $cm^{-1}$ mol $L^{-1}$) were comparable with the clinically approved ICG, which ensured the signal brightness during NIR-II imaging (Supplementary Table 3). Among all the LZ dyes, LZ-1105 exhibited the highest fluorescence intensity (4.1-fold higher than that of ICG) with a fluorescence quantum yield ($\Phi_f$) of 1.69% in mice blood due to the reduced quenching effect (Fig. 1c and Supplementary Figs. 6–8). Moreover, superior photostability of

LZ-1105 was observed by exposing ICG and LZ-1105 to the continuous laser irradiation in deionized water, PBS (pH = 7.4), urine, serum and blood for 3 h, respectively, (Supplementary Fig. 9). The full width at half maximum (FWHM)[3,10,21,32,35,37,39], contrast,[37,40] and SBR[4,10,19] were used to measure the resolution and clarity of the images (see "Methods" section and Supplementary Fig. 10). Before ICG and LZ-1105 administration for NIR-II imaging, signal collection condition was optimized in vitro and in vivo. Overall, 1300 and 1400 nm long-pass filters were finally selected to acquire optimal NIR-II imaging for ICG and LZ-1105, respectively, (Supplementary Figs. 11–23 and Supplementary Note 1). When capillary tubes filled with LZ-1105 or ICG solution were immersed in 1% intralipid solution at increased phantom depth, bioimaging results of LZ-1105 resolve sharper edges of the capillary at a depth up to 6 mm than that of ICG (Fig. 1d). The FWHM of capillary tubes filled with ICG increased more significantly than that of LZ-1105 with increasing penetration depth, demonstrating a higher imaging resolution achieved by LZ-1105 (Fig. 1e). The potential cytotoxicity of LZ-1105 was evaluated in human umbilical vein endothelial cells (HUVEC). The cells exhibited over 95% viability after incubation with 350 μM of LZ-1105 for 24 h (Supplementary Fig. 24). Moreover, the nanosecond life time (0.15 ns) of LZ-1105 molecule under the excitation of 1064 nm laser illustrated that the triplet state was not involved in the fluorescence process at 1105 nm, excluding the phototoxicity caused by the generation of singlet oxygen (Supplementary Fig. 25).

To evaluate the spatial resolving ability of LZ-1105 for in vivo imaging, brain, hindlimb, and lymphatic system in healthy nude mice (brain) and ICR mice (hindlimb and lymphatic system) were detected with ICG as a comparison. The penetration depths were measured as 1.31, 1.31, and 0.50 mm for brain, hindlimb, and lymphatic system (Supplementary Fig. 26). Noninvasive cerebral vessel imaging was performed with higher resolution through the intact scalp and skull with LZ-1105 administration (Fig. 1f). The SBR of LZ-1105 injected group (10.1) was ~8.4-fold higher than that of the ICG group (1.2) (Fig. 1g). Moreover, the FWHM of the vessels at the same position were measured as 105.1 μm (LZ-1105) and 231.8 μm (ICG), respectively. In addition, contrast was also used to evaluate the resolving ability of the images. The contrast of LZ-1105 group (0.42) was approximately twofold higher than that of the ICG group (0.21), demonstrating the superior SBR and spatial resolution achieved by LZ-1105. Besides, similar results were obtained during NIR-II imaging of blood vessels and lymphatic system of the hindlimb achieved by LZ-1105 administration (Fig. 1h–k and Supplementary Fig. 27). As another evidence to illustrate the high resolution achieved by LZ-1105, 36-μm-wide tiny blood vessel and six lymph vessels with the diameters from 90 to 161 μm were visualized clearly after LZ-1105 injection, which was far superior to the broad signal distribution measured with ICG (Supplementary Fig. 28 and Fig. 1k). Furthermore, the widths of the hindlimb blood vessels at the same position were measured as 130 μm with NIR-II imaging and 126 μm with white light optical photograph, whereas ultrasound imaging could not discern any vessel, demonstrating the precise imaging ability of LZ-1105 in NIR-II window (Supplementary Fig. 29).

**In vivo pharmacokinetics and blood retention**. After intravenous injection (i.v.) of LZ-1105 (5 mg $kg^{-1}$), NIR-II imaging of mice abdominal, brain, and hindlimb at various time points were performed (Fig. 2a–c). The vasculature was clearly visualized within 12 h post injection (p.i.), and almost no fluorescence signal could be observed in the bladder and liver. Although the fluorescence signal of vasculature gradually decreased with time, the

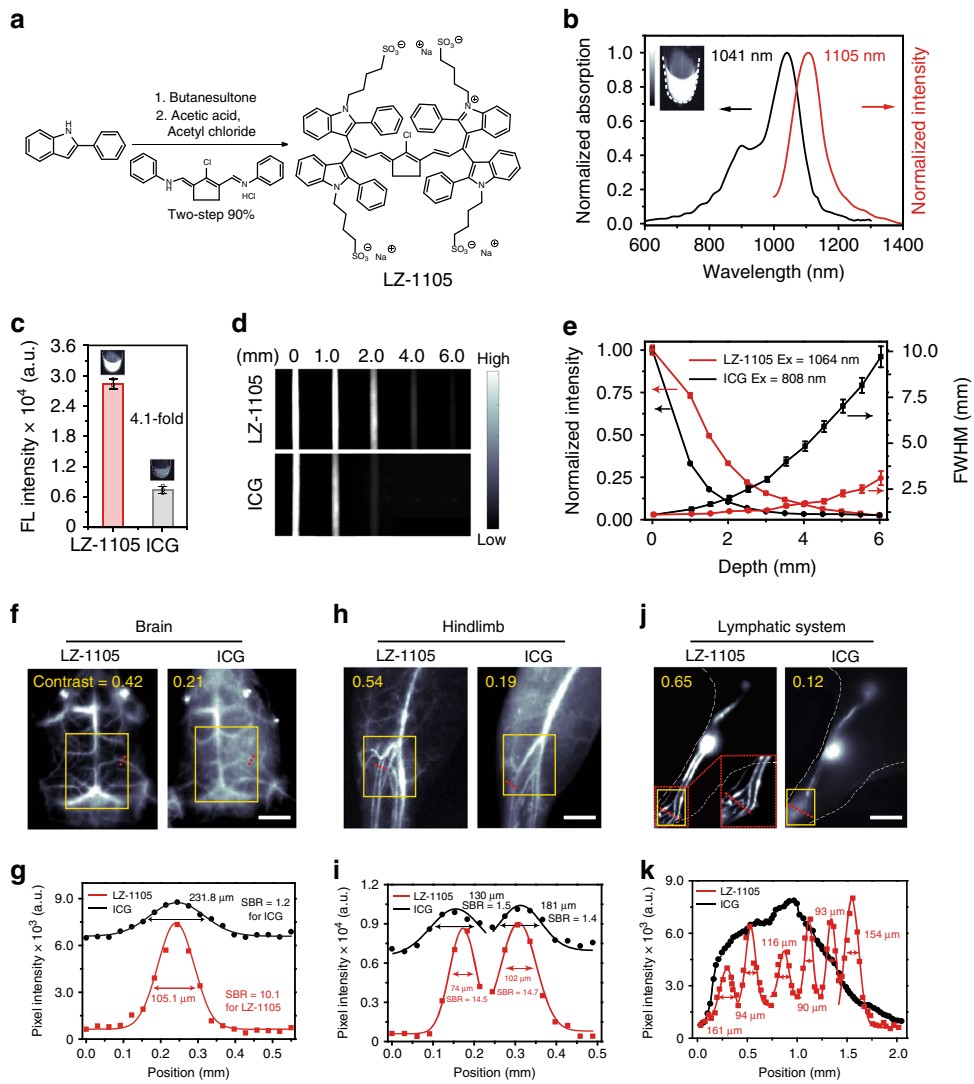

**Fig. 1 Optical characterization of LZ-1105 and in vivo NIR-II imaging comparision by LZ-1105 and ICG. a** Synthetic route of LZ-1105. **b** Normalized absorption and fluorescence intensity of LZ-1105 in PBS, demonstrating an absorbance peak at 1041 nm and an emission peak at 1105 nm. The fluorescent emission spectrum was obtained under 1064 nm laser excitation. Inset: An NIR-II fluorescence image of LZ-1105 (10 μM) in PBS. **c** The fluorescence intensity of LZ-1105 and ICG in mice blood under 1064 nm (30 mW cm$^{-2}$, 1400 nm long-pass filter) and 808 nm (30 mW cm$^{-2}$, 1300 nm long-pass filter) laser excitation, respectively. Inset: NIR-II fluorescence image of LZ-1105 and ICG in mice blood ([LZ dyes] = [ICG] = 10 μM). NIR-II images (**d**), normalized signal intensity, and full width at half maximum (FWHM) (**e**) of LZ-1105 (top, $\lambda_{ex}$ = 1064, 1400 nm long-pass filter) and ICG (bottom, $\lambda_{ex}$ = 808, 1300 nm long-pass filter) through various thicknesses of 1% Intralipid solution. Equal fluorescent intensities of LZ-1105 and ICG at 0 mm were obtained by adjusting the laser's working power density. Noninvasive NIR-II fluorescence images of brain (**f**), hindlimb (**h**), and lymphatic system (**j**) in nude mice (brain) and shaved ICR mice (*n* = 3) (lymphatic system and hindlimb) i.v. injected with LZ-1105 (1400 long-pass filter, $\lambda_{ex}$ = 1064 nm, 300 ms) or ICG (1300 long-pass filter, $\lambda_{ex}$ = 808 nm, 300 ms) (inset in **j** shows a magnified view of the red grid, contrast was calculated in the yellow box). **g**, **i**, **k** The fluorescence intensity profiles (dots) and Gaussian fit (lines) along the red dashed line in brain (**f**), hindlimb (**h**), and lymphatic system (**j**). Data point with its error bar stands for mean ± s.d. derived from *n* = 3 independent experiments. Scale bars in **f**, **h**, and **j** represent 3 mm. Source data underlying **c**, **e**, **g**, **i**, and **k** are provided as a Source Data file.

SBR remained above 5.0 even at 4 h p.i., surpassing the Rose criterion to distinguish image features with high certainty[35] (Fig. 2d). Therefore, the blood vessels of the abdomen, brain, and hindlimb could be obviously distinguished from the background noise, ensuring the long-term observation of dynamic vascular structures within 4 h. In comparison, the blood vessels of abdomen, brain, and hindlimb could only be visualized within 5 min after injection of ICG (5 mg kg$^{-1}$), and almost no fluorescence signal of vasculature could be observed at 10 min, which could not be utilized for long-term vascular imaging by single injection.

To investigate the pharmacokinetics difference between LZ-1105 and ICG in living mice after i.v. injection, venous blood

samples at various time points were collected to measure the dye contents. Long blood half-life time of 195.4 min for LZ-1105 and a short α half-life time of 3.1 min for ICG were obtained (Fig. 2e and Supplementary Figs. 30–31). In addition, clearance pathways were also different between ICG and LZ-1105. ICG was rapidly captured by the liver within 30 min and remained for 6 h followed by hepatobiliary clearance[37], while only weak fluorescence signal could be detected in urine of LZ-1105 injected mouse. Unexpectedly, after mixing with mice blood, an 11.3-fold signal enhancement was observed in urine (Supplementary Fig. 32), demonstrating the free state of LZ-1105 molecule in urine. Meanwhile, stability of LZ-1105 in blood and urine was illustrated

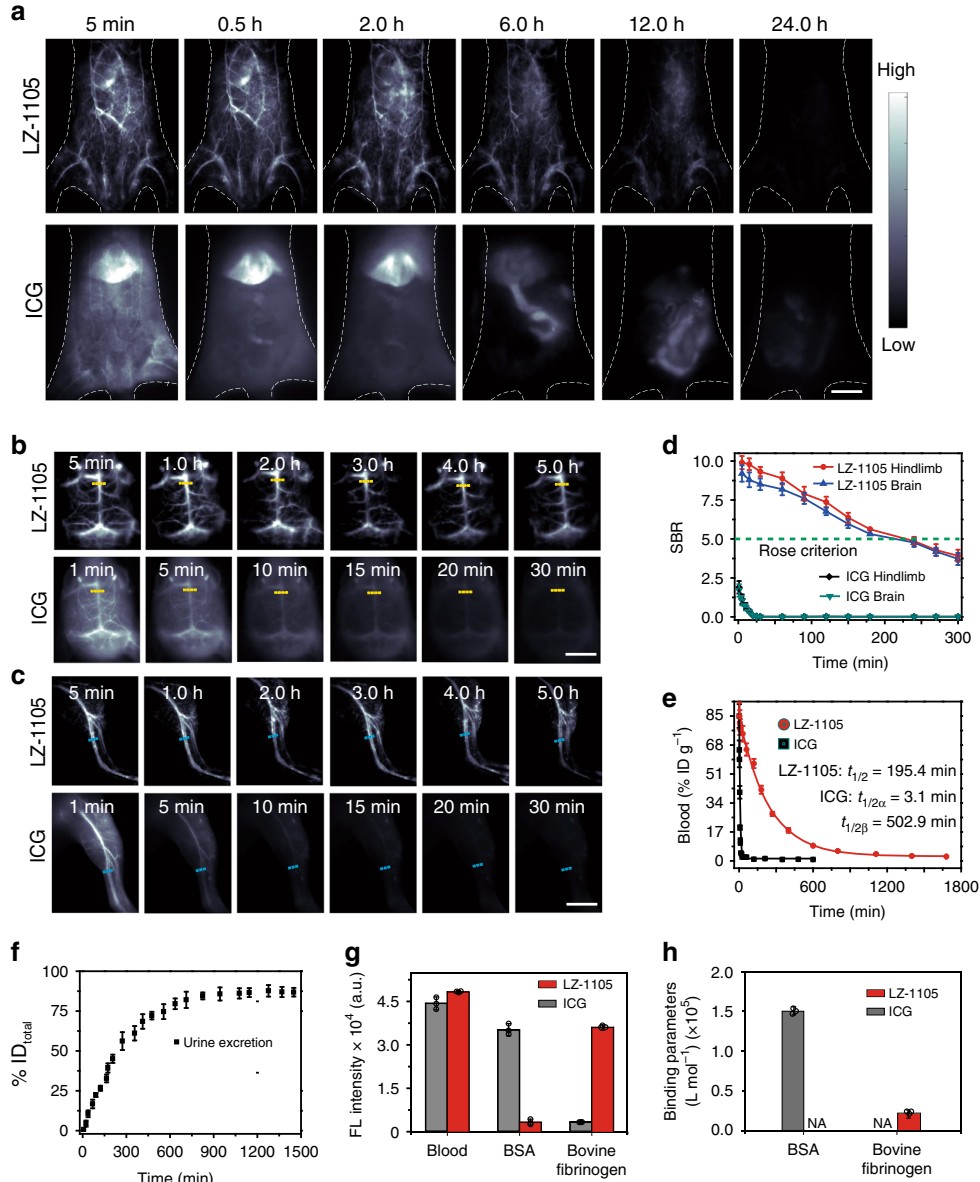

**Fig. 2 Pharmacokinetics and blood retention of LZ-1105 and ICG.** NIR-II bioimaging of mice body (**a**), brain (**b**), and hindlimb (**c**) after i.v. injected with LZ-1105 (1400 nm long-pass, $\lambda_{ex} = 1064$ nm, 300 ms) and ICG (1300 nm long-pass, $\lambda_{ex} = 808$ nm, 300 ms). **d** The corresponding signal-to-background ratio (SBR) of LZ-1105 and ICG administrated mice along the yellow and cyan dashed line in **b** and **c** as a function of time. The green dotted line indicates the Rose criterion. **e** Blood circulation ($\%ID\ g^{-1}$) of LZ-1105 and ICG administrated mice as a function of time. **f** Cumulative urine excretion curve for LZ-1105 injected mice within 24 h p.i. **g** NIR-II signal intensity of LZ-1105 and ICG with different excitations and long-pass filters. (1064 nm excitation and 1400 nm long-pass filters for LZ-1105, 808 nm excitation and 1300 nm long-pass filters for ICG. Excitation power was adjusted to obtain comparable NIR-II signal in mice blood. [LZ-1105] = [ICG] = 10 μM) **h** The binding parameters of LZ-1105 and ICG with bovine serum albumin (BSA) and bovine fibrinogen by isothermal titration calorimetry (ITC) at 20 °C. N/A means not measureable by ITC. $n = 3$ independent mice experiments for **a**–**c**. Data point with its error bar stands for mean ± s.d. derived from $n = 3$ independent experiments for **d**–**h**. Scale bars in **a**, **b**, and **c** represent 4 mm. Source data underlying **d**, **e**, **f**, **g**, and **h** are provided as a Source Data file.

by reversed-phase high performance liquid chromatographic (RP-HPLC) (Supplementary Fig. 33). Around 86% of the injected dose (% ID) of LZ-1105 was found in urine within 24 h p.i. with an elimination rate constant of 0.079 h⁻¹ (Fig. 2f and Supplementary Figs. 34–35). Besides, ex vivo images of main organs including the liver, heart, spleen, kidney, lung, stomach, intestine, skin, muscle, brain, and bone of LZ-1105 injected mice at 1 to 24 h p.i. showed extremely low organ uptake (Supplementary Fig. 36).

To explore the possible clearance path of LZ-1105, ICG and LZ-1105 were respectively mixed with mice blood, bovine serum albumin (BSA), bovine fibrinogen, and other media (Fig. 2g and

Supplementary Fig. 37). Under excitation, NIR-II signals were collected by different wavelength long-pass filters. ICG exhibited a 23-fold signal enhancement in BSA solution than that of in PBS, and the signal intensity was comparable with that of in mice blood. These results illustrated that after mixing with mice blood, binding with serum albumin was the main reason for ICG signal increase, which was also consistent with previous report[37]. Interestingly, fluorescent signals of LZ-1105 only increased obviously in the presence of bovine fibrinogen. Meanwhile, after hydrolyzing the fibrinogen in blood sample by plasmin, fluorescent signal of LZ-1105-blood mixture decreased about

fourfold compared with that of the untreated blood. Besides, LZ-1105 was further demonstrated to have little interaction with HUVEC and mouse blood cells including red blood cells, platelets, neutrophils, and lymphocytes (Supplementary Figs. 38 and 39). Sodium dodecyl sulfate polyacrylamide gel electrophoresis was used for plasma analysis. And the electrophoresis testing exhibited that only fibrinogen related proteins could be observed with a NIR-II InGaAs camera, demonstrating the NIR-II signal enhancement of LZ-1105 was mainly due to the interaction between fibrinogen and LZ-1105 (Supplementary Fig. 40). To further illustrate the interaction between LZ-1105 and fibrinogen, isothermal titration calorimetry (ITC) was utilized to measure its binding ability (Fig. 2h, Supplementary Figs. 41, 42, and Supplementary Table 4). In contrast to the obviously binding feature of ICG with BSA, scarcely any interaction between LZ-1105 and BSA could be observed. On the contrary, LZ-1105 tended to bind with bovine fibrinogen and no interaction between ICG and bovine fibrinogen could be observed. These results were consistent with the fluorescence imaging results (Supplementary Fig. 37), illustrating the different binding type between LZ-1105 and ICG with blood. These results might be used to explain the long-term blood circulation feature of LZ-1105. Furthermore, due to the weak binding parameter between LZ-1105 and bovine fibrinogen (18,140 L mol$^{-1}$, which was approximately sevenfold lower than that of ICG and BSA (127,113 L mol$^{-1}$)), the separation between LZ-1105 and fibrinogen was speculated to be easier than that of ICG and serum albumin. During the circulation, separation between LZ-1105 and fibrinogen might occur in kidney, leading to the renal exertion of LZ-1105 molecules with low NIR-II signal intensity. With long-term blood circulation, low cytotoxicity and the superior SBR, LZ-1105 provided a promising method for continuously real-time imaging of vascular structural changes and physiological process in small animals.

**Real-time imaging of the ischemic reperfusion in hindlimbs**. Ischemia usually occurs in fracture and other peripheral arterial diseases in limbs[41]. After ectopic replantation and functional reconstruction, real-time observation of ischemic reperfusion is of critical importance to evaluate the degree of recovery. To observe the ischemic reperfusion in limbs, we induced hindlimb ischemia by ligation of the femoral vein and artery using hemostatic clip for different durations (Fig. 3a)[42,43]. LZ-1105 was then i.v. injected into hindlimb ischemia mice as contrast agent. The clips were removed at varied timepoint and the ischemic reperfusion in hindlimbs was evaluated for 30 min (Fig. 3b, c). Three regions of interest (ROI, named as front, middle and end sites marked by yellow circles in Fig. 3b) were chosen to observe and evaluate the ischemic reperfusion process. Ischemic reperfusion could be visualized by the NIR-II bioimaging signals from the front site to the end site of occluded veins. By calculating the recovery time and blood flow velocity (BFV), real-time fluorescence bioimaging revealed a remarkable delay of signals in ischemic hindlimbs with longer clipping time (Supplementary Movies 1–4). The recovery time of the ischemic hindlimbs were 0.28, 1.63, and 18.67 min for 1, 4, and 8 h clipping groups, respectively, (The recovery time was defined as the duration from the timepoint of removing the clips to the timepoint of observing signal at the end site.). In addition, after 12 h clipping group, ischemic reperfusion was not observed at the middle and end site in 0.5 h and the signal at front site disappeared gradually from 5.3 min after removing the clip. Meanwhile, the branch vessels around the front site failed to reperfuse in the 12 h clipping group. These results demonstrate that complete reperfusion would not be realized if the blood supply is cut off for a long period of time. Furthermore, to

evaluate the ischemic reperfusion quantitatively, we calculated the BFV of three ROI (front, middle, and end site) at different time points after removing the clips with various clipping durations (Fig. 3b, c and Supplementary Figs. 43–46). At the same ROI, the BFV decreased with increasing clipping time. And the recovery time was extended with the extension of the clipping time (4.67, 7.33, 25.33, and 133.33 s at the front site for 1, 4, 8, and 12 h clipping, respectively) (Supplementary Tables 5–7). Meanwhile, the recovery time and BFV of branch vessels around the front site have shown the same trend for the middle and end sites (Supplementary Figs. 47–49 and Supplementary Table 8). All the results demonstrated the reduced ischemic reperfusion rate with the increasing clipping time. Therefore, NIR-II imaging performed with LZ-1105 could be used to monitor instantaneous BFV during the in vivo ischemic reperfusion with real-time feedback and high spatial resolution in living mice.

**Real-time monitoring of carotid artery thrombolysis**. Real-time and continuous monitoring of thrombolysis in small animals is of great significance in preclinical research[44–47]. Currently, recombinant tissue plasminogen activator (rt-PA), a glycoprotein which enhances the destruction of blood clots, has been widely used for thromboembolism[48,49]. The carotid arterial thrombosis model was induced by the FeCl$_3$ method[50] on the right carotid artery and verified by ultrasonography and Haematoxylin and Eosin staining (Fig. 4a inset and Supplementary Fig. 50). Then LZ-1105 was i.v. injected into the carotid arterial thrombosis mouse to observe the normal (left) and affected (right) side. NIR-II imaging of cervical region showed clear visualization of the normal carotid side, while the affected side was not observed due to the occlusion of blood flow (Fig. 4b). Then rt-PA was i.v. injected into the carotid thrombosis mouse and the thrombolysis process was monitored by the real-time NIR-II imaging (Fig. 4b and Supplementary Movie 5). NIR-II signal front of the affected side could be observed from 3.12 min and continuously increased until 12.12 min p.i. of rt-PA, giving the recovered blood flux rate in thrombolysis process of 0.1016 ± 0.0110 mL min$^{-1}$ (Fig. 4c). Through real-time NIR-II imaging, the thrombolysis process and blood flow recovery can be monitored in time in small animals.

**Real-time monitoring of the opening and recovery of BBB**. The BBB is a specialized structure of central nervous system formed by endothelial cells, it prevents most large drug molecules (≥400 Da) from entering the brain[51]. To evaluate the permeability of focused ultrasound (FUS)-induced BBB, several methods have been established including photoacoustic imaging, MRI, and two-photon microscopy[52,53]. However, most of them suffer from various disadvantages, including low sensitivity, long acquisition time, and invasive skull removing operation. Therefore, a noninvasive, highly sensitive, and fast acquisition method for monitoring the opening and recovery processes of BBB is in urgent demand. Herein, FUS combined with circulating microbubbles was used as a noninvasive approach to induce reversible and temporary opening of the BBB[52] (Fig. 5a). After i.v. injection of LZ-1105, noninvasive cerebrovascular imaging can be performed through the scalp and skull with a high degree of clarity in the NIR-II window. A large bright spot (violet circle) were observed immediately after 20 s sonication treatment owing to the leakage of LZ-1105 from the cerebral vessels, illustrating the FUS-induced BBB opening (Fig. 5b and Supplementary Movie 6). Meanwhile, a small bright spot (green circle) lighted up gradually and reached the maximum after 2 min. Afterward, NIR-II signal intensity of both spots decreased continuously with half-times of 5.4 and 21.5 min, respectively, demonstrating the gradual recovery of the BBB (Fig. 5c, d). Within the opening and recovery process of BBB,

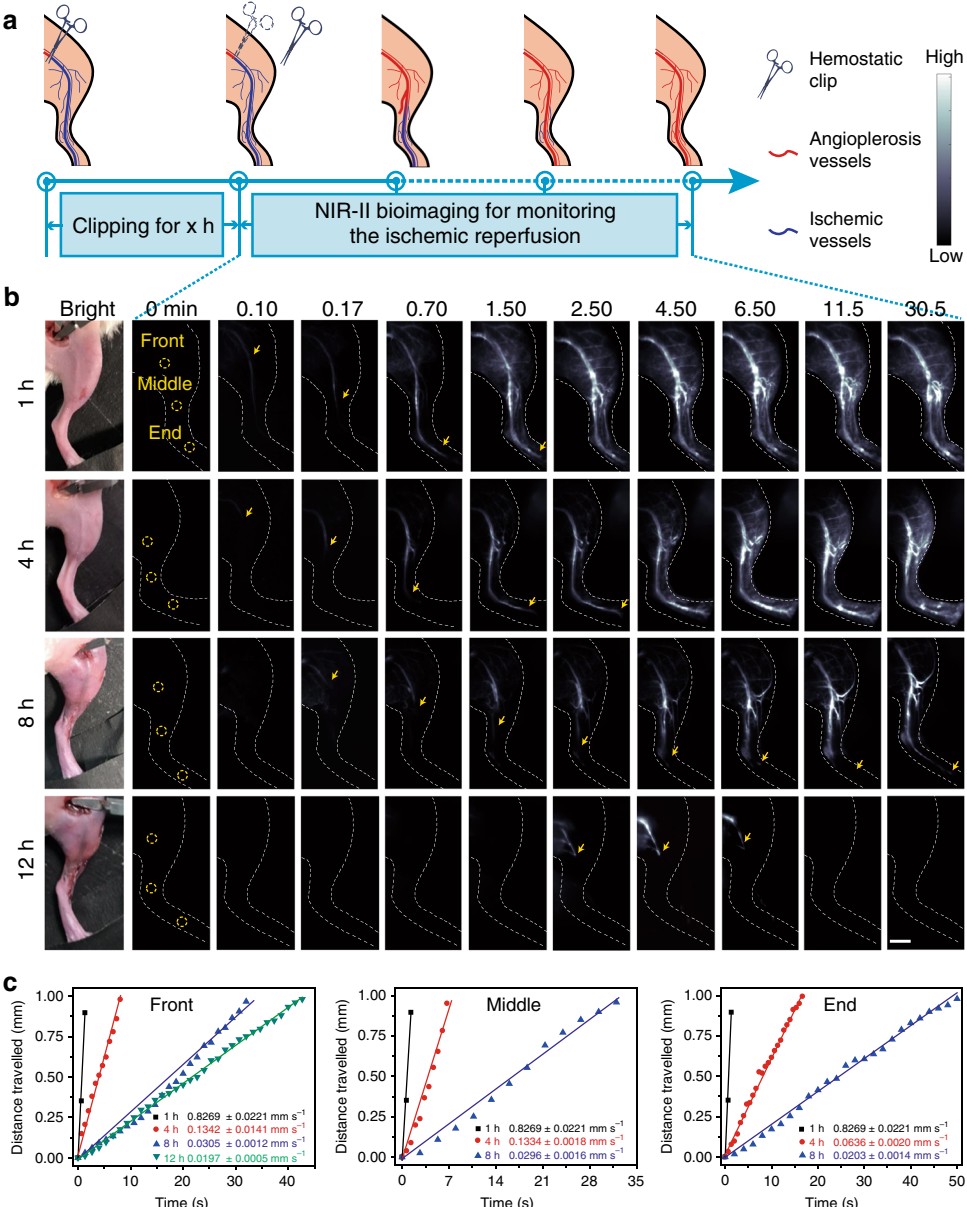

**Fig. 3 Real-time NIR-II imaging of the ischemic reperfusion in hindlimbs. a** Schematic illustration of the ischemic reperfusion process. **b** NIR-II bioimaging with LZ-1105 administration of ischemic reperfusion at different time points after various period clipping treatment (1400 nm long-pass, $\lambda_{ex} = 1064$ nm, 300 ms) as indicated. Three regions of interest (ROI) named as the front, middle, and end sites as shown with yellow circles were chosen to evaluate the ischemic reperfusion process. Yellow arrows indicate the flow front. **c** The corresponding distance traveled by the flow front at the front, middle, and end site as a function of time after 1, 4, 8, and 12 h clipping, respectively. The slope of the function was calculated as blood flow velocity (BFV) and indicats mean ± s.d. derived from $n = 3$ replicated measurements. Scale bar represents 4 mm. Source data underlying **c** is provided as a Source Data file.

LZ-1105 administration outlined the cerebral vascular structures clearly at a depth over 1.3 mm under the intact scalp and skull (Fig. 5b and Supplementary Fig. 26). These results suggested that LZ-1105-based long-term NIR-II imaging could facilitate real-time tracking of the temporary opening and recovery of the BBB in the mouse brain, providing an effective method to monitor the pathogenesis of brain diseases, and investigate the drug delivery and treatment strategies.

## Discussion

NIR-II imaging simultaneously provides anatomical and hemodynamic information thanks to reduced tissue scattering and nonradiative effect[3,5,10,35]. Continuous monitoring in small animal is helpful to broaden the understanding of

vascular dysfunction and recovery. To achieve high SBR and resolution in vivo, low tissue scattering is the main prerequisite for both excitation and emission wavelength. In general, under the same penetration depth, imaging resolution mainly depends on tissue scattering. According to the Mie theory, photon scattering experienced in biological tissue is inversely related to excitation and emission wavelength ($\mu_s' \sim \lambda^{-w}$, where $\mu_s'$ is the reduced scattering coefficient, $\lambda$ is the wavelength and the exponent w is determined by the scattering and ranges from 0.22 to 1.68 for different tissues)[54,55]. Thus, acquiring images beyond longer wavelength could achieve low background noise and high resolution. Therefore, we designed and synthesized LZ dyes with excitation/emission wavelength in NIR-II window.

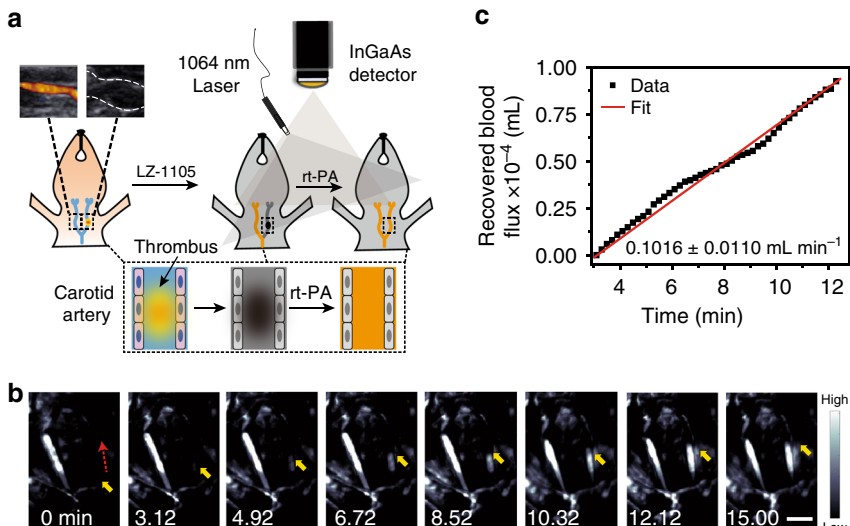

**Fig. 4 Real-time NIR-II imaging of the thrombolysis process in carotid artery. a** Schematic illustration of the thrombolysis process. Inset: Ultrasonic imaging of carotid artery without (left) and with (right) thrombus, demonstrating successful formation of a thrombus in the right carotid artery ($n = 3$ mice). **b** NIR-II bioimaging of the carotid artery performed by LZ-1105 at various time points p.i. of recombinant tissue plasminogen activator (rt-PA) administration (1400 nm long-pass, $\lambda_{ex} = 1064$ nm, 300 ms). The red dashed and yellow solid arrows indicate the blood flow direction and signal front, respectively. **c** Recovered blood flux during thrombolysis process as a function of time. The slope of the function was calculated as the recovered blood flux rate in thrombolysis and indicats mean ± s.d. derived from $n = 3$ replicated measurements. Scale bar represents 3 mm. Source data underlying **c** is provided as a Source Data file.

All the LZ dyes display the maximal absorptions and emissions in different solvents within the NIR-II window in the range of 1003–1118 nm. Such NIR-II absorption bands can be assigned to intramolecular charge transfer (ICT) between the electron-donating unit and the electron-withdrawing group, and can be further verified by their distinct solvatochromic effect (Supplementary Table 2).

To gain insight into the electronic properties of the LZ dyes, density functional theory calculations were carried out using the B3LYP method and 6–31 G (d) basis set[56]. As shown in Fig. 6a, b, both the highest occupied molecular orbitals (HOMOs) and the lowest unoccupied molecular orbitals (LUMOs) are delocalized over the heptamethine backbone for all the LZ dyes. However, the HOMOs and the LUMOs are partially separated on the indole and indolium moieties, which results in the ICT interactions and the NIR-II absorption transitions. Moreover, it can be found that upon the fusion of dioxolane groups on the indole moieties, the HOMOs are extendedly delocalized over the dioxoloindole units. Therefore, the extended π-conjugation and the enhanced ICT interactions bring about the bathochromically shifted absorption maxima of LZ-1092 and LZ-1118 in comparison to LZ-1060 and LZ-1105, respectively. Furthermore, when the hexane rings are replaced by pentene rings, a 13° bend can be observed for the heptamethine backbones of LZ-1105 and LZ-1118 due to the torsion of the introduced pentene (Fig. 6c). Consequently, the conjugation degree and the HOMO/LUMO delocalization slightly decrease in pentene-cored LZ dyes. As a result, the HOMOs are more localized on the electron-donating indole moieties while the LUMO are more localized on the electron-withdrawing indolium moieties. The enhanced charge separation explains the bathochromic shifts of the absorption maxima upon replacing the hexane rings by pentene rings. Similarly, all the LZ dyes display strong emission bands in the NIR-II window. The maximum emission wavelengths of the LZ dyes are consistent with the trend in their absorption maxima and can be explained likewise.

By extended π-conjugation and the enhanced ICT interactions, LZ-1105 was synthesized and used for NIR-II imaging with 1064 nm excitation and 1400 long-pass filter. LZ-1105 shows high spatial resolution and SBR in bioimaging. On the other hand, contrast agents with long blood circulation are of critical importance for the continuous observation of pathological or recovery processes. LZ-1105 can bind to fibrinogen and has the property of long-term blood retention with a half-life time of 3.2 h. With these angiography features, LZ-1105 is capable of quantifying femoral artery blood velocity in ischemic hindlimbs, monitoring thrombolysis in the context of carotid arterial thrombosis, and monitoring the reversible and controllable opening and recovery of the BBB in living mice.

Although real-time NIR-II optical bioimaging holds advantages of fast feedback and nonradiative effect, this noninvasive imaging method can only be used for superficial vessels or on small animals to monitor the vascular dynamics related diseases. In future works, improvements of the imaging equipment, such as larger range of detectors with higher sensitivity or endoscope techniques may permit the use of this technique in larger animal models. However, LZ-1105-based optical imaging provides advantages for vascular dynamic monitoring, which could be combined with other traditional imaging techniques for improved imaging. The characteristics of the LZ-1105 dye suggest applicability to a range of vascular physiological processes, which is likely to prove advantageous in the future.

In summary, this work describes a small molecular organic LZ-1105 to achieve rapid, noninvasive, high resolution and real-time imaging of vascular dynamics in vivo. With its long-term blood circulation and excitation and emission in the NIR-II window, LZ-1105 could be utilized for static or persistent imaging of dynamic vascular processes, including ischemic reperfusion in hindlimbs, thrombolysis in carotid artery and temporary opening and recovery of the BBB in mice. This method may be used to dynamically discover and monitor more vascular related phenomenon and enable new research into the pathogenesis of vasculature related diseases in small animals.

## Methods

**Optical characterization**. Absorption spectra of LZ dyes in different solvents (DMSO, Methanol, Ethanol and PBS (pH = 7.4) solutions) were taken on a

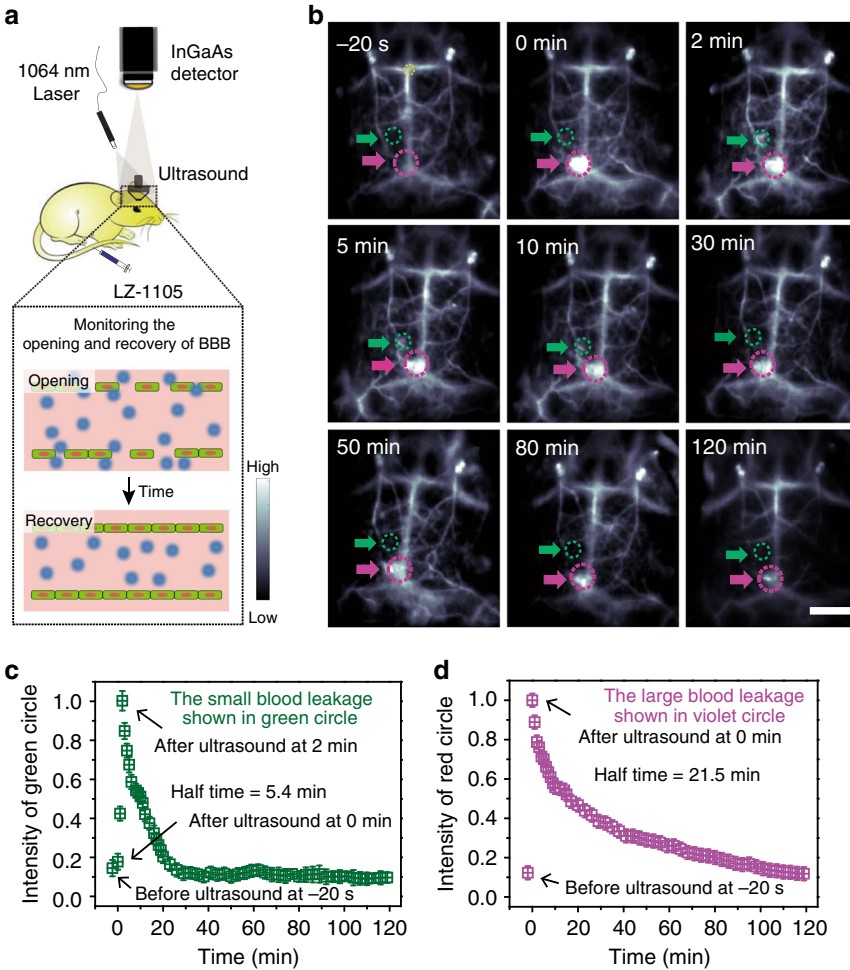

**Fig. 5 Real-time noninvasive NIR-II imaging to monitor the opening and recovery of the blood–brain barrier (BBB). a** Schematic illustration of the NIR-II imaging setup to monitor the opening and recovery of the BBB. **b** The NIR-II images of the mice brain before and after treated with focused ultrasound (FUS) and microbubbles at different time points ($n = 3$ mice). The green and violet circles and arrows indicate the opening and recovering points of cerebral vessels. Corresponding normalized mean signal intensity of green (**c**) and violet a (**d**) circles as a function of time. The fluorescence intensity region (noted as the yellow circle) of each images was normalized to measure the signal intensity decreasing process of the leakage on the brain vessels. Data point with its error bar stands for mean ± s.d. derived from $n = 3$ replicated measurements. Scale bar represents 4 mm. Source data underlying **c** and **d** are provided as a Source Data file.

PerkinElmer Lambda 750S UV–visible–NIR spectrometer. The NIR-II fluorescence emission spectra were obtained on an Edinburgh Instruments F-980 fluorescence spectrometer or home-built liquid-nitrogen-cooled InGaAs linear array detector (Princeton Instruments NIRvana640LN) with 808 nm laser or 1064 nm laser diode. The NIR-II fluorescence images of LZ dyes in blood, BSA, and PBS were measured using an InGaAs array detector (Princeton Instruments NIRvana640LN) under 1064 nm laser excitation (30 mW cm$^{-2}$). The emitted light was collected by the camera filtered through 1400-nm long-pass filter (LZ dyes) or 1300-nm long-pass filter (ICG) and focused onto the InGaAs NIRvana camera with a SWIR-35 hyperspectral lenses (Navitar). The exposure time for all images was 300 ms. RP-HPLC analyses were performed on an Waters equipped with a UV detector and an Atlantls T3 OBD-C18 RP ($10 \times 250$ mm) column, with methanol and $H_2O$ (0.1% of trifluoroacetic acid) as the eluent. Chemical structures were created using Chem-BioDraw Ultra 12.0. Statistical analyses were performed using OriginPro 8.0 and Microsoft Excel 2010 software. Images and movies were processed with the LightField imaging software 6.11 and MATLAB R2015b.

**Calculating the fluorescence quantum yield of LZ dyes**. The quantum yields of LZ-1060, LZ-1092, LZ-1105, and LZ-1118 in different solvents ($C_2H_5OH$, $CH_3OH$, DMSO, PBS (pH = 7.4)) were determined by using the dye IR-26 ($\Phi_f = 0.05\%$ in 1,2-dichloroehane (DCE)) as reference). Briefly, a serial dilution with OD < 0.1 at 980 nm of IR-26 and LZ dyes were performed in different solvents, and the fluorescent emission spectra were collected under excitation of 980 nm. The integrated emission intensity were plotted as a function of the OD value at 980 nm and fitted into a linear function. The different slopes of the linear fit between IR-26 and LZ dyes were obtained. The quantum yields of LZ dyes in different solvents were

calculated using the following Eq. (1):

$$QY_{sample} = QY_{ref} \times \frac{n_{sample}^2}{n_{ref}^2} \times \frac{Slope_{sample}}{Slope_{ref}}, \qquad (1)$$

where $QY_{sample}$ is the quantum yield of LZ dyes in different solvents, $QY_{ref}$ is the quantum yield of IR-26 in DCE, $n_{ref}$ and $n_{sample}$ are the refractive indices of IR-26 and LZ dyes in different solvents. $Slope_{sample}$ and $Slope_{ref}$ are the slopes obtained by linear fitting of the integrated emission spectra of IR-26 and LZ dyes in different solvents.

**Signal to noise, resolution using standardized measurements**. Signal-to-background ratio (SBR): When calculating SBR in NIR bioimaging, we draw a red dashed line in the region of interest (ROI), and the fluorescence intensity profiles across the line would be extracted and fitted by Gaussian fitting[4,10,19]. The highest intensity on the Gaussian fitted curve is selected as signal ($I_{signal}$) and the intensity of the baseline on Gaussian fitted curve is selected as background ($I_{background}$). SBR is calculated as Eq. (2):

$$SBR = \frac{I_{signal}}{I_{background}}. \qquad (2)$$

Full width at half maximum (FWHM): FWHM is a parameter to evaluate resolving ability of the images. When evaluating FWHM, we draw a line in the ROI, and the FWHM is extracted from the fluorescence intensity profiles across a line fitted by Gaussian fitting[3,10,21,32,35,37,39].

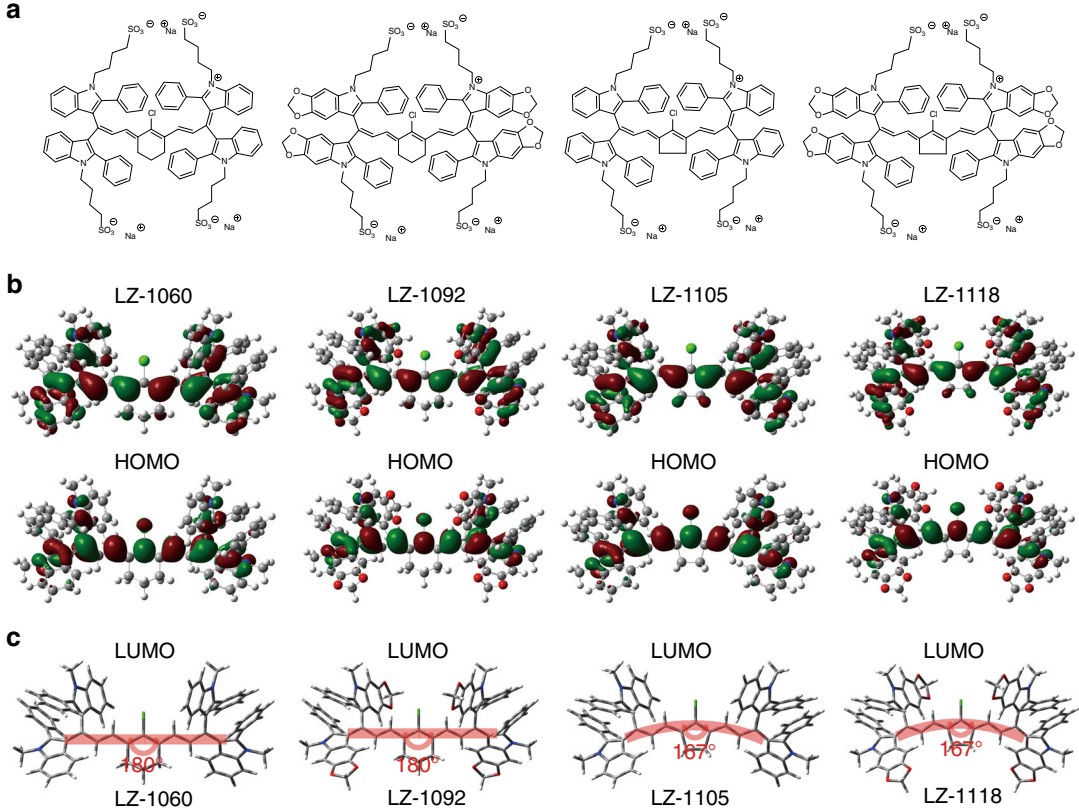

**Fig. 6 Density functional theory calculations for LZ dyes. a** The chemical structures of LZ dyes. **b** The HOMOs and LUMOs were plotted based on the optimized S1 geometries of the LZ dyes. **c** Optimized geometries of the LZ dyes in ground states.

Contrast: Contrast was also a parameter to evaluate resolving ability, which was defined as the standard deviation ($\sigma$) divided by the mean pixel intensity ($\mu$) in the red box[37,40] (Supplementary Fig. 10).

**Animal handling.** All the following animal procedures were agreed with the guidelines of the Institutional Animal Care and Use Committee of Fudan University and performed in accordance with the institutional guidelines for animal handling. All of the animal experiments were approved by the Shanghai Science and Technology Committee. Five-week-old female nude mice were obtained from Shanghai SLAC Laboratory animal CO. Ltd for brain vascular imaging and monitoring the opening and recovery of the BBB studies, five-week-old female ICR mice were obtained from Shanghai SLAC Laboratory animal CO. Ltd for excretion and hindlimb, lymphatic system vascular imaging studies, eight-week-old female ICR mice were obtained from Shanghai SLAC Laboratory animal CO. Ltd for monitoring ischemic reperfusion process and thrombolysis process studies. Mice were housing in independent ventilation cage system under specific-pathogen free condition. Housing temperature was 22–25 °C, humidity was 35–45%. The dark/light cycle was 12 h light/12 h dark. Before imaging, the mice were anaesthetized using rodent ventilator with air mixed with 4% isoflurane. The tail vein injection of contrast agents was carried out in the dark. The injected dose was a 150 μL bolus in a PBS (pH = 7.4) solution at specified concentration. During the time course of imaging the mouse was kept anaesthetized by a nose cone delivering air mixed with 4% isoflurane. Mice were randomly selected from cages for all experiments. No blinding was performed.

**Ultrasound for detecting the thrombus in carotid artery.** Ultrasound measurements were performed using the ACUSON S3000 ultrasound system (Siemens, Erlangen, Germany) equipped with a linear probe (18 MHz). The carotid artery was identified employing Duplex-ultrasonography (B-Mode and color Doppler).

**In vitro cytotoxicity assay.** HUVEC were purchased from Stem Cell Bank, Chinese Academy of Sciences. Cells were excluded mycoplasma contamination by mycoplasma Detection Kit. HUVEC were cultured in 1640 culture supplemented with 10 and 1% penicillin–streptomycin at 37 °C in a humidified atmosphere of 5% $CO_2$. The cells were seeded at a density of 10 000 cells per well (100 μL total volume per well) in 96-well assay plates for 12 h. Then, LZ-1105 at the indicated concentrations (0, 25, 50, 100, 150, 250, and 350 μM) were added to the cell culture medium. Cells were incubated with LZ-1105 for 12 and 24 h. To determine the toxicity, 10 μL of Cell Counting Kit-8 solution was added to each well of the

microtiter plate and the plate was incubated in the $CO_2$ incubator for additional 24 h. Enzyme dehydrogenase in living cells was oxidized by this kit to orange carapace. The quality was assessed calorimetrically by using a multireader (TECAN, Infinite M200, Germany). The measurements were based on the absorbance values at 450 nm. Following formula was used to calculate the viability of cell growth as Eq. (3):

$$\text{Viability}(\%) = (\text{mean absorbance value of treatment group} \\ /\text{mean absorbance value of control group}) \times 100. \tag{3}$$

**Titration calorimetry.** Isothermal titration calorimetry (ITC) was performed using a MicroCal VP-ITC system at 20.00 ± 0.01 °C. In individual titrations, injections of 2 μL of LZ-1105 (0.1 mM) or ICG (0.1 mM) were added from the computer-controlled 40 μL microsyringe at an interval of 2 min into the BSA (0.01 mM) or bovine fibrinogen (0.01 mM) (230 μL), while centrifuging at $20 \times g$.

**RP-HPLC measurement.** The LZ-1105 were analyzed by RP-HPLC using eluent A and eluent B (Eluent A: $H_2O$ containing 0.01% trifluoroacetic acid; eluent B: methanol. A/B = 40/60 to 20/80 for 42 min, 4 mL min⁻¹, detect wavelength: 294 nm; samples were filtered by 0.45 μm membrane). Blood sample was acquired at 4 h after LZ-1105 administration. Methanol (200 μL) was added into plasma (50 μL) to precipitate the proteins. The mixture was vortexed and subsequently centrifuged at $13,500 \times g$ for 10 min. The supernatant was mixed with an equal volume of deionized water and subjected to RP-HPLC. Urine same was collected within 5 h after LZ-1105 administration.

**Vascular imaging in the NIR-II window.** Mice were anaesthetized and placed on a stage with a venous catheter for injection of LZ-1105 (5 mg kg⁻¹) or ICG (5 mg kg⁻¹). All NIR-II images were collected on a 640 × 512 pixel InGaAs NIRvana640LN camera. For LZ-1105, the excitation laser was an 1064 nm laser diode at a power density of ~70 mW cm⁻² (lower than the safe exposure limit of 1.0 W cm⁻² determined by the International Commission on Nonionizing Radiation Protection), and emission was collected with 1400-nm long-pass filter (Thorlabs). For ICG, the excitation laser was an 808 nm laser diode at a power density of ~70 mW cm⁻², and emission was collected with 1300-nm long-pass filter (Thorlabs). Images and movies were processed with the LightField imaging software 6.11 and MATLAB R2015b. The hindlimb and lymphatic system vasculature, noninvasive brain vessels imaging were performed in mice using the home-built InGaAs array detector. The exposure time for all images was 300 ms. All images performed by

ICG or LZ-1105 were processed under the same fluorescence intensity range (0–10,000).

**In vivo pharmacokinetics and blood retention**. Five-week-old female ICR mice were used for pharmacokinetics, excretion, and biodistribution studies with three mice for each experiment group. All mice were intravenously injected with LZ-1105 or ICG at a dose of 5 mg kg$^{-1}$. The blood was collected at needed time points using a capillary tube from the retro-orbital sinus. The concentration of LZ-1105 or ICG in blood was estimated by the fluorescent intensity using an InGaAs array detector. For LZ-1105, the excitation laser was a 1064 nm laser diode at a power density of ~ 70 mW cm$^{-2}$, and emission was collected with 1400-nm long-pass filter. For ICG, the excitation laser was an 808 nm laser diode at a power density of ~70 mW cm$^{-2}$, and emission was collected with 1300-nm long-pass filter. The percentage of the LZ-1105 or ICG in blood was calculated as Eq. (4):

$$\%ID\,g^{-1} = \frac{F_t - F_{control}}{(F_{injected} - F_{control}) \times M_t} \times 100\%, \quad (4)$$

where $F_t$ is the fluorescent intensity of collected blood as measured with a 1400-nm long-pass filter, $F_{control}$ is the fluorescent intensity of control blood, $F_{injected}$ is the fluorescent intensity of the injected LZ-1105 (mixed with blood), $M_t$ is the mass of the collected blood. The kinetic study for LZ-1105 and ICG was fitted by Pharmacokinetic simulation software Drug and Statistics 2.0. Biodistribution was performed after 1, 6, and 24 h i.v. Main organs including liver, heart, spleen, kidney, lung, stomach, intestine, skin, muscle, brain, and bone were collected for NIR-II fluorescence imaging. For excretion experiments, urine was collected for 24 h after injection of LZ-1105. The fluorescence intensity of LZ-1105 remained stable after incubated with urine at 37 °C for 180 min, illustrating the photostability of LZ-1105 in urine. Mice were placed in plastic cages with water, and urine was collected in a 50 µL pipette. Fresh mice blood was added into each sample. The mixture was drawn up into a capillary tube and the fluorescence measured (1400-nm long-pass filter, 70 mW cm$^{-2}$, 300 ms, InGaAs array detector), as well as the minimal background fluorescence in control urine and capillary tube. The amount of urine was calculated as Eq. (5):

$$\%ID = \frac{\sum_{m=1}^{m=n_f} ((I_m - I_{control}) \times V_m)}{(I_{injected} - I_{control}) \times V_{injected}}, \quad (5)$$

where $n$ is the number of a urine timepoint from mice ($n = 3$), $I$ is the average fluorescent intensity as measured with a 1400-nm long-pass filter, $I_{control}$ is the fluorescent intensity of control urine, $I_{injected}$ is the fluorescent intensity of the injected dose, and $V$ is the volume of the urine, and $V_{injected}$ is the volume of the injected LZ-1105.

**NIR-II imaging of ischemic reperfusion in hindlimbs**. Five-week-old female ICR mice were shaved and anaesthetized using rodent ventilator with air mixed with 4% isoflurane. In the anesthetized state, unilateral hindlimb ischemia was induced by ligation of the left vein and femoral artery[42,43]. NIR-II imaging was performed on ischemic hindlimbs using LZ-1105 (5 mg kg$^{-1}$). For movie-rate imaging, the camera was set to continuously expose using the LightField imaging software 6.11 with a 300 ms exposure time (1.52 frame per second). Images and movies were processed with the LightField imaging software 6.11 and MATLAB R2015b. When LZ-1105 was i.v. injected into hindlimb ischemia mice, the clips were removed and the ischemic reperfusion in hindlimbs was evaluated for 30 min. To calculate the absolute BFV of the hindlimb, the position of the front of the fluorescence signal was extracted from each frame, and plotted as a function of time. The BFV was defined as the instantaneous blood velocity of the timepoint at which the bioimaging signal front reached the three ROI. The distance traveled by blood front showed a linear increase versus time, and the slope of the linear fit was the BFV in terms of mm s$^{-1}$.

**Real-time NIR-II imaging of the thrombolysis process in carotid artery**. Five-week-old female ICR mice were shaved and anaesthetized using rodent ventilator with air mixed with 4% isoflurane. In the anesthetized state, the carotid thrombosis model was induced by the FeCl$_3$ method[50] on the right carotid artery. Real-time NIR-II imaging was performed on carotid artery using LZ-1105 (5 mg kg$^{-1}$) before rt-PA (50 µL, 0.15 mg mL$^{-1}$) injection.

**Real-time NIR-II imaging of the opening and recovery of BBB**. Five-week-old female nude mice were anaesthetized using rodent ventilator with air mixed with 4% isoflurane. In the anesthetized state, the opening and recovery of BBB was induced by FUS and microbubbles[52]. A FUS transducer was used to temporarily open the BBB of mice, driven by a function generator connected to a power amplifier. A removable cone filled with ultrasonic coupling agent was employed to hold the transducer and guide the US beam into the brain. The acoustic parameters used were 0.6 MPa acoustic pressure, 0.5 MHz frequency, 1 ms pulse interval, and 20 s sonication duration. A total of 50 µL of microbubbles (mean diameter of about 2 µm and concentration of about $1 \times 10^9$ bubbles mL$^{-1}$) were intravenously injected into mice before sonication. Real-time NIR-II imaging was performed using LZ-1105 (5 mg kg$^{-1}$). The fluorescence intensity region (noted as the yellow

circle in the Fig. 5b) of each images was normalized to measure the signal intensity decreasing process of the leakage on the brain vessels

**Density functional theory calculations for LZ dyes**. Density function theory calculations were performed using Gaussian 03 revision C.02 software and B3LYP method and 6–31G* basis set[56]. The geometries were optimized using the default convergence criteria without any constraints.

**Reporting summary**. Further information on research design is available in the Nature Research Reporting Summary linked to this article.

## Data availability

The data that support the findings of this study are available from the corresponding authors upon reasonable request. The source data underlying Figs. 1c, e, g, i, k, 2d–h, 3c, 4c, and 5c, d and Supplementary Figs. 6, 7, 9, 12b–h, 13b–g, 14b–e, 15b–d, 16, 17c, 18b–d, 19, 20b–d, 21, 22b–d, 23–25, 27b, 28a, b, 29b, 30b, d, f, 31c, d, 32, 34a, 35, 36b, c, 37a, b, 38, 39b, 41, 42, 43b, 44b, 45b, 46b, 47b, 48b, and 49b are provided in a Source Data file. Source data are provided with this paper.

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

## Acknowledgements

This work was supported by the National Key R&D Program of China (2017YFA0207303), National Natural Science Foundation of China (21725502 and 51961145403), and Key Basic Research Program of Science and Technology Commission of Shanghai Municipality (17JC1400100 and 19490713100).

## Author contributions

B.L., M.Z., L.F., C.D., S.D., G.Z., L.L., H.Z., F.C., and C.J. were performed the experiments. B.L., M.Z., L.F., C.D., G.Z., and L.L. analyzed the data. B.L. and M.Z. wrote the paper and were assisted by G.Z., X.L., and F.Z. G.L., S.Z., C.J., Y.W., Y.C., D.Z., and F.Z. provided guidance on the study design.

## Competing interests

The authors declare no competing interests.
