## [Peer Review File · Nature Communications]

Reviewers' comments:

Reviewer #1 (Remarks to the Author):

Long-term blood circulation NIR-II molecular probe with absorption and emission beyond 1000 nm for in vivo dynamics imaging

Li, et al.

Summary: This is one of a number of recent contributions to explore the use of the NIR-II wavelength range in qualitative preclinical imaging. This contribution lacks the traceable measurements that are needed to draw the conclusions and speculations on clinical application that are made throughout the contribution. Herein the authors report a NIR-II agent LZ-1105 with excitation at 1064 nm and collection of emission light >1400 nm and test it in preclinical studies of ischemic reperfusion, thrombolysis in the carotid artery, and transient permeability of the BBB. Comparison is made with preclinical tests of ICG similarly dosed, but excited at 808 with emission collected (non-optimally) at >1300 nm. The authors make several broad conclusions that are not substantiated by the results presented as pointed out below. In addition, the manuscript is in need of grammatical review and editing to remove jargon and clarify meaning.

The contribution is interesting, but not convincing as more rigorous testing using traceable measurements on standardized materials are needed.

Major points:

1. In the abstract, the applicants state that the advantages of NIR-II are deep penetration depth, high resolution, and fast feedback. Unfortunately, the increased water absorption in the NIR-II ranges limits its application to shallow depths. Indeed in this contribution, the depth of 1.3 mm is repeated (definitely not deep by medical imaging standards); there is no discussion of a traceable resolution measurement in the contribution; and the exposure times of 300 ms are quite long compared to video rate NIR –I and –II measurements routine found in the literature.
2. Line 69-71 states that NIR-II has higher resolution contrast (sub – 10um) than visible and NIR-I ranges. But this depends upon the depth and system studied. There is no study demonstrating improved microscopy at these wavelengths that this reviewer is aware of (nor any that are referenced) and the statement is inaccurate. A careful study of contrast, signal to noise, resolution using standardized measurements in absolute traceable units (i.e., not AU) will be needed before such statements are made (or repeated).

3. NIR-II has limited penetration depth due to the increased water absorption as compared to NIR-I and to this reviewer's knowledge, no published study has been successfully performed on an animal model larger than a rodent. Finally the NIR-I wavelength range is 780-900 nm as below 780 nm, light is visible.

4. While ICG has a short blood half-life due to hepatobiliary clearance, it associates with albumin, making it a good blood pooling agent that does not extravasate. (By the way, there are several efforts to prolong blood half-life by incorporating ICG in liposomes.). Yet the small molecule LZ-1105 is stated NOT to associated with blood proteins, yet why does it not extravasate ? This is a significant question.

5. The addition of BSA to ICG reduces quenching caused by the stacking of the planar ICG structure and the supplementary Fig 15 really has little to do with interactions with blood proteins as the authors claim. Is LZ-1105 subject to quenching at high concentrations as ICG is?

6. Line 88: There was no traceable measures of image resolution or depth presented using ICG and LZ-1105 that justifies this statement that LZ-1105 is far superior to ICG. In fact, one might argue that the ICG imaging was performed in a disadvantaged way, in that the emission from 808 nm excitation should have collected with a long pass 1064 nm filter. Indeed, the ICG emission collection missed the majority of signal, which make this an unfair test. Line 366 should be struck.

7. Line 92: The authors state that LZ-1105 is capable of real-time quantification of femoral artery blood flow, when in fact, they simply use the advancing front of the dye to assess a velocity with a bolus administration and no volumetric flow is quantitated. This is not a viable means of measuring blood flow rate as currently available, for example with Doppler US. In addition, the limitations in depth penetration caused by water absorption will restrict this application from clinical use, even if there were some application that could be found for assessing superficial assessment of perfusion. Line 97 should be struck as there is no evidence that sufficient depths can be imaged to make NIR-II clinically viable. Line 350 should be struck as it has not been shown in this manuscript that one can use the technology to detect changes in blood flow in response to drugs or stimuli without having to inject contrast agents repeatedly.

8. Line 117: The term "Imaging accuracy" is used incorrectly. Imaging accuracy is determined by sensitivity and specificity in a receiver-operator characteristic study. This should be removed. Indeed, the resolution (which is what is being probed here) is dependent upon the emission level, i.e., what is the pW/sr/cm² of emission light collected ? Clearly, there is limited ICG emission collected due to the sub-optimal filtering and this result will clearly change when ICG emission is collected optimally.

9. Lines 135-137 Lymphatic vessels are not visible without contrast. Hence it is difficult to understand how lymphatic vessel diameter can be determined non-invasively or invasively from white light photography.

10. What is the stability of LZ-1105 ? None is shown in serum at 37C for the time-frame of the animal studies. How is renal clearance occurring if the urine and bladder are dimly fluorescent?

11. Cytotoxicity could be associated with triplet states, especially with the closeness of π orbitals. An assessment of lifetime decays would be most appropriate.

12. Line 171 -173 makes little sense. Instability of LZ-1105 needs to be performed in a stability study, not from assessing shift of fluorescence spectra.

13. Line 217-219 claiming superior and abundant information needs to be struck. Line 240-242 claiming use of the technology in clinical practice should be struck. Line 274-275, “. . . surpasses current medical modalities” is grandstanding and not appropriate for a scientific journal. Line 341 claiming the technique allows clinicians to visualize real-time treatment efficacy of drugs should be struck.

14. Lines 277: Long blood half life is needed ONLY for blood pooling agents. The authors confuse their application with the rest of medical imaging that requires short blood half lives for maximum contrast.

15. Line 290: The shorter excitation of ICG eliminates water absorption and enables greater penetration, albeit at reduced resolution to due greater scattering. Indeed, this reviewer finds that ICG, when optimally used in NIR-I windows (as used in some clinical applications) outperforms ICG when probed in NIR-II windows --- when compared with traceable measurements.

16. Paragraph starting on line 329 is not pertinent to the manuscript.

Other points:

1. In the abstract LZ-1105 is termed a molecular agent, when it is actually used as a blood pooling agent without any targeting moiety used. Short blood half-lives are desired for molecular agents since this increases target to background ratios. The authors should not refer to LZ-1105 as a molecular agent.

2. Line 78: ICG is clinically used, but not approved for biomedical optical imaging.

3. Line 471: meaning? Perhaps grammatical corrections would convey meaning.

Reviewer #2 (Remarks to the Author):

Bioimaging in the NIR window using organic fluorophores is of significant importance in the field of cancer diagnosis and therapy. This study is especially attractive because NIR-II imaging has the potential to overcome the current limitation of reflectance-based imaging, i.e., spatial resolution and tissue penetration depth. By designing hydrophilic small-molecule NIR-II organic dyes from commercially available benzopyrrole derivatives could obtain a long-term blood circulation (3.2 h)

dye LZ-1105 for the dynamic vascular imaging. This is the first small-molecule NIR-II probe that resulted in an impressive dynamic vascular process, incorporating the ischemic reperfusion in hindlimbs, thrombolysis in the carotid artery and opening and recovery of the blood-brain barrier. The manuscript is well organized, results are mostly supported by solid experimental data, and conceptually novel for the NIR-II organic probe design and bioapplications. Although the overall quality and presented data are solid, however, the data have several problems that need to be addressed:

- 1) Excitation with longer wavelengths may cause thermal effects, which could be a serious concern for their future clinical application. It would be helpful to discuss this issue.
- 2) Although clear images were obtained utilizing 1400 nm long-pass filter, the fluorescence intensity would be very low >1400 nm according to the emission spectra. The author should discuss the NIR-II bioimaging results using various long-pass filters (850-, 1,000-, 1,200-, 1,300-, and 1,400-nm).
- 3) The author had only focused on the photostability of LZ-1105 in blood, PBS and DI. The physicochemical and optical properties of new fluorophore need to be analyzed in serum-containing warm media. Biodistribution (% ID/g) and quantification need to be reconsidered with considering photo-decay (photo-ablation by laser), thickness and optical properties (scattering coefficient) of each organ, and other pharmacokinetic parameters.
- 4) The authors have shown that the bioimaging resolution and penetration depth with LZ-1105 was far superior to that of ICG for non-invasive imaging of mouse cerebral vasculatures, hindlimb vasculatures and lymphatic vasculatures. How about the penetration depth for these in vivo imaging experiment. Please provide supporting data for the relationship between the bioimaging resolution and penetration.
- 5) Up to now, several dyes have also been reported as NIR-II contrast agents with absorption and emission beyond 1000 nm for in-vivo imaging. The authors should add these papers in the references (J. Med. Chem. 2019, 62, 2049; Adv. Optical Mater. 2019, 1900229, Chem. Sci., 2019, 10, 1219, Adv. Healthcare Mater. 2018, 7, 1800589; Chem. Sci. 2017, 8, 3489), which may support the argument.
- 6) The manuscript also still requires revision with respect to the language used. A professional language editing service to assist the authors in correcting the spelling, grammar, word use, and punctuation throughout the manuscript. To help with this process the authors may also ask a native English speaker or equivalent.
- 7) In Figure 1h-k, signal to background ratio (SBR) of hindlimb vessels and lymphatic images were not given.
- 8) In Figure 2e, the tiny blood vessels were not clear at 2 h after injection. However, they are very clear in Figure 5b. The authors should give a reasonable explanation.
- 9) The scale bars should be provided for all figures and the key parameters such as filters, exposure time should be provided in the text and SI.

10) In Figure S10, the spatial resolution between the bright field image and the NIR-II image with the clinical imaging method should be provided.

11) In Figure 6, the key parameters of these programs should be provided.

12) Probably it's better to use velocity measurement with the highest spatial resolution that could be the real advantages for in vivo kinetic study, especially to understand alpha phase half-life.

Point-to-point Response to Reviewers:

Reviewer #1 (Remarks to the Author):

Long-term blood circulation NIR-II molecular probe with absorption and emission beyond 1000 nm for in vivo dynamics imaging

Li, et al.

Summary: *This is one of a number of recent contributions to explore the use of the NIR-II wavelength range in qualitative preclinical imaging. This contribution lacks the traceable measurements that are needed to draw the conclusions and speculations on clinical application that are made throughout the contribution. Herein the authors report a NIR-II agent LZ-1105 with excitation at 1064 nm and collection of emission light >1400 nm and test it in preclinical studies of ischemic reperfusion, thrombolysis in the carotid artery, and transient permeability of the BBB. Comparison is made with preclinical tests of ICG similarly dosed, but excited at 808 nm with emission collected (non-optimally) at >1300 nm. The authors make several broad conclusions that are not substantiated by the results presented as pointed out below. In addition, the manuscript is in need of grammatical review and editing to remove jargon and clarify meaning.*

The contribution is interesting, but not convincing as more rigorous testing using traceable measurements on standardized materials are needed.

Response: Thanks for the comments. We appreciate the reviewer for the useful suggestions. Related revisions have been made in the revised manuscript according to the comments, including the appropriate description and detailed measurement methods.

Major points:

Comment 1. *In the abstract, the applicants state that the advantages of NIR-II are deep penetration depth, high resolution, and fast feedback. Unfortunately, the increased water absorption in the NIR-II ranges limits its application to shallow depths. Indeed in this contribution, the depth of 1.3 mm is repeated (definitely not deep by medical imaging standards); there is no discussion of a traceable resolution measurement in the contribution; and the exposure times of 300 ms are quite long compared to video rate NIR –I and –II measurements routine found in the literature.*

Response: Thanks for the comments. We agree with the reviewer that water absorption increases gradually with the increasing wavelength, however there are still some optimal penetration window with lower water absorption in NIR-II region, such as ~1100 nm and ~1300 nm (**Figure R1-1**). Furthermore, besides tissue/water absorption, there are more factors which would influence the penetration ability, such as scattering and tissue autofluorescence (shown as the following **Figure R1-2**). As mentioned in Line 280 in our original manuscript, the tissue penetration depth depends on the absorption and scattering of the excitation and emission light. According to a proposed theoretical model: $\delta = [3\mu_a(\mu_a + \mu_s')]^{-1/2}$, where δ is penetration depth, μ_a is the optical absorption extinction coefficient, μ_s' is the reduced scattering coefficient. As for reduced scattering coefficient μ_s' , empirical measurements of scattering in tissues by light with different wavelengths reveal that the degree of scattering for almost all biological tissues follows an inversely

proportional wavelength relationship ($\mu_s' \sim \lambda^{-6}$, ω ranges from 0.22 to 1.68 for different tissues), **which means that the light scattering decreases with the increasing wavelength (Figure R1-3)**. In addition, tissue autofluorescence causes high background during imaging, and the excitation or emission light would not be distinguished from the background noise, leading to the decreased signal-to-background ratio (SBR) and penetration depth. The wavelength dependency of autofluorescence from vital organs and body fluids has been found to show progressively decreasing level of autofluorescence with increasing detection wavelength, and the autofluorescence could be hardly detected beyond 1300 nm (**Figure R1-4**). **Thus, penetration depth is a comprehensive result of tissue absorption, scattering and tissue autofluorescence.**

Figure R1-1. Absorption spectrum of water through a 1-mm-long path. OD, optical density. (Cited from: Hong, G., Antaris, A.L. & Dai, H. Near-infrared fluorophores for biomedical imaging. *Nat. Biomed. Eng.* **1**, 0010 (2017).)

Figure R1-2. Light-tissue interactions resulting from impinging excitation light. Interface reflection, scattering, absorption and autofluorescence all contribute to the loss of fluorescence signal and the gain of noise. (Refer to: Hong, G., Antaris, A.L. & Dai, H. Near-infrared fluorophores for biomedical imaging. *Nat. Biomed. Eng.* **1**, 0010 (2017).)

Figure R1-3. Reduced scattering coefficients of different biological tissues and of intralipid scattering tissue phantom as a function of wavelength in the 400-1700 nm region, which covers the visible, NIR-I and NIR-II windows. They all show reduced scattering at longer wavelengths. (Cited from: Hong, G., Antaris, A.L. & Dai, H. Near-infrared fluorophores for biomedical imaging. *Nat. Biomed. Eng.* **1**, 0010 (2017).)

Figure R1-4. Autofluorescence spectrum of ex vivo mouse liver, spleen and heart tissue under 808 nm excitation light, showing the absence of autofluorescence in the >1,500 nm NIR-II window. Inset: Autofluorescence spectra at high wavelengths. The dashed denote the values at three standard deviations from the baseline. (Cited from: Diao, S. *et al.* Biological imaging without autofluorescence in the second near-infrared region. *Nano Res.* **8**, 3027-3034 (2015) ; Hong, G., Antaris, A.L. & Dai, H. Near-infrared fluorophores for biomedical imaging. *Nat. Biomed. Eng.* **1**, 0010 (2017).)

Besides, tissue scattering and autofluorescence would also influence the resolution. When evaluating resolution in NIR bioimaging, we draw a line in the region of interest (ROI), and the fluorescence intensity profiles across a red line would be extracted and fitted by Gaussian fitting. The full width at half-maximum (FWHM) was hired to evaluate the imaging resolution (as we mentioned in **Comment 2, Figure R1-8** and Supplementary Fig. 10 in the revised supplementary information). This is also a common method to measure the resolution in NIR bioimaging (Antaris, A.L. *et al.* A small-molecule dye for NIR-II imaging. *Nat. Mater.* **15**, 235-242 (2016); Hong, G. *et al.* Multifunctional in vivo vascular imaging using near-infrared II fluorescence. *Nat. Med.* **18**, 1841-1846 (2012); Carr, J.A. *et al.* Shortwave infrared fluorescence imaging with the clinically approved near-infrared dye indocyanine green. *Proc. Natl. Acad. Sci. USA* **115**, 4465-4470 (2018);

Diao, S., *et al.* Fluorescence imaging in vivo at wavelengths beyond 1500 nm. *Angew. Chem. Int. Ed.* **54**, 14758-14762 (2015); Ma, Z., *et al.* Near-infrared IIb fluorescence imaging of vascular regeneration with dynamic tissue perfusion measurement and high spatial resolution. *Adv. Funct. Mater.* **28**, 1803417 (2018). As mentioned above, scattering and autofluorescence decreases with the increasing wavelength, thus, resolution in NIR-II region is higher than that of visible and NIR-I region.

Actually, there has been some works which illustrate that increased wavelength has deeper penetration and higher resolution (Figure R1-5 and R1-6).

Figure 1 NIR-I and NIR-II fluorescence imaging of blood vessels in the mouse. (a) A schematic showing that upon excitation by a 785-nm laser, the SWNT-IRDye-800 conjugate emits at the ~800 nm NIR-I region from IRDye-800 dye and the 1.1–1.4 μm NIR-II region from the SWNT backbone. (b) Absorption spectrum of the SWNT-IRDye-800 conjugate (black dashed line), emission spectrum of IRDye-800 dye (green line) and SWNTs (red line). (c) A digital camera photograph (left) and a NIR-II fluorescence image of injected solution containing 0.10 mg ml⁻¹ SWNT-IRDye-800 conjugates. (d) Schematic of the imaging setup for simultaneous detection of both NIR-I and NIR-II photons using silicon and InGaAs cameras. A zoomable lens set was used for adjustable magnifications. (e–g) NIR-I fluorescence images (top) and cross-sectional fluorescence intensity profiles (bottom) along red-dashed lines of a mouse injected with the SWNT-IRDye-800 conjugates. Gaussian fits to the profiles are shown in red dashed curves. (h–j) NIR-II fluorescence images (top) and cross-sectional fluorescence intensity profiles (bottom) along red-dashed bars of a mouse injected with the SWNT-IRDye-800 conjugates. Gaussian fits to the profiles are shown in red dashed curves.

Figure R1-5. NIR-I and NIR-II fluorescence imaging of blood vessels in the mouse. (Cited from: Hong, G. *et al.* Multifunctional in vivo vascular imaging using near-infrared II fluorescence. *Nat. Med.* **18**, 1841-1846 (2012))

Figure R1-6. (a) An image of a whole mouse body and fluorescence images of lymph vessels and a lymph node of the mouse. BSA-coated QDs (CdSe/ZnS; 520 nm emission, CdSeTe/CdS; 720 nm emission, and PbS/CdS; 1100, 1300, and 1500 nm emission) were injected into the footpad shown in the whole body image. The white lines in the fluorescence images are used for the line profile analysis in (b). Line profile analysis of the fluorescence intensities of the QDs in the lymph system. The values of S/B express signal to background ratios in the fluorescence intensities. (Cited from: Tsukasaki, Y. *et al.* A short-wavelength infrared emitting multimodal probe for non-invasive visualization of phagocyte cell migration in living mice. *Chem. Commun.* **50**, 14356-14359 (2014))

Meanwhile, there has been some works reported that increased wavelength has deeper penetration and higher resolution as following.

An analysis of photon-tissue interactions (Box 1) reveals that photon scattering in biological tissue scales with $\lambda^{-\alpha}$, where λ is the photon's wavelength and the exponent $\alpha = 0.2-4$ for biological tissues. This indicates that, at a fixed tissue depth, longer NIR-II imaging wavelengths should facilitate higher imaging clarity than achievable in the NIR-I window. The traditionally known 'biologically transparent window' of 700–900 nm with low tissue absorbance can extend past 1,000 nm, allowing for high imaging-penetration depths, although wavelengths in the 1,400–1,500-nm range are typically avoided due to the presence of a water overtone absorbance peak¹²². Additionally, autofluorescence induced by laser or light-emitting diode (LED) excitation sources decreases exponentially when imaging in the NIR-II window under a typical 808-nm excitation, reaching undetectable levels past 1,500 nm¹²³⁻¹²⁵. The benefits of much reduced photon scattering and autofluorescence can greatly outweigh the slightly higher light absorption in the NIR-II window over that in the NIR-I window. In recent years, it has been shown that the combination of minimal scattering, still low-photon absorbance and diminished tissue autofluorescence in the NIR-II optical window can dramatically increase feature clarity, penetration depths and SBR ratios.

Cited from : Hong, G., Antaris, A.L. & Dai, H. Near-infrared fluorophores for biomedical imaging. *Nat. Biomed. Eng.* **1**, 0010 (2017), page 11.

Fluorescence imaging is a method of real-time molecular tracking in vivo that has enabled many clinical technologies. Imaging in the shortwave IR (SWIR; 1,000–2,000 nm) promises higher contrast, sensitivity, and penetration depths compared with conventional visible and near-IR (NIR) fluorescence imaging. However, adoption

Cited from: Carr, J.A. *et al.* Shortwave infrared fluorescence imaging with the clinically approved near-infrared dye indocyanine green. *Proc. Natl. Acad. Sci. USA* **115**, 4465-4470 (2018), page 1.

The minimal autofluorescence of biological tissue in the SWIR region leads to increased sensitivity^{2,3}, while the significantly reduced light attenuation from scattering and from absorption by the blood and other structures enables imaging with high spatio-temporal resolution and penetration depth³⁻⁹. Consequently, large organisms such as a whole mouse may be rendered translucent when imaged using SWIR fluorescence^{4,8,10}.

Cited from: Bruns, O.T. *et al.* Next-generation in vivo optical imaging with short-wave infrared quantum dots. *Nat Biomed Eng* **1**, 0056 (2017) , page 1.

visible and NIR imaging (Fig. 1a and Fig. S4 in ESI†). Lymph node images by SWIR fluorescence imaging, especially at 1100 and 1300 nm, are much clearer than those by visible and NIR imaging due to the higher penetration of the SWIR light in tissue. The

Cited from: Tsukasaki, Y. *et al.* A short-wavelength infrared emitting multimodal probe for non-invasive visualization of phagocyte cell migration in living mice. *Chem. Commun.* **50**, 14356-14359 (2014), page 14356.

Because of the low photon scattering, autofluorescence, deeper tissue penetration, and improved signal–background ratio (SBR) in the second near-infrared (NIR-II, 1000–1700 nm) window, NIR-II optical imaging is considered as an ideal imaging modality than traditional NIR-I (650–900 nm) optical imaging.^{1–4} The NIR-II window is further divided

Cited from: Ding, B., *et al.* Polymethine Thiopyrylium Fluorophores with Absorption beyond 1000 nm for Biological Imaging in the Second Near-Infrared Subwindow. *J. Med. Chem.* **62**, 2049–2059 (2018), page 2049.

Stokes shift up to ~400 nm. Compared to the NIR-I window, which has been extensively explored for *in vitro* and *in vivo* imaging^{13–18}, the longer wavelength emission in NIR-II makes SWNTs advantageous for imaging owing to reduced photon absorption^{19,20}, reduced scattering by tissues^{8,21}, negligible tissue autofluorescence and, thus, deeper tissue penetration^{7,8,22}. This allows for unprecedented fluorescence-

Cited from: Hong, G. *et al.* Multifunctional *in vivo* vascular imaging using near-infrared II fluorescence. *Nat. Med.* **18**, 1841–1846 (2012), page 1841.

Recently, we and others have shown that biological imaging with carbon-nanotube and quantum-dot fluorescent agents in the long NIR region (1.0–1.7 μm , termed the NIR-II region) can benefit from the reduced tissue scattering and autofluorescence in that wavelength range, reaching deeper penetration depths *in vivo* than traditional NIR (750–900 nm, termed NIR-I) imaging techniques^{17–29}. Photon scattering scales as $\lambda^{-\alpha}$, where λ is the

Cited from: Hong, G. *et al.* Through-skull fluorescence imaging of the brain in a new near-infrared window. *Nat. photonics* **8**, 723–730 (2014), page 723.

900–1700 nm.^[1,7] The SWIR biomedical imaging has demonstrated much improvement compared to conventional NIR imaging by virtue of the far more

reduced light scattering, less tissue autofluorescence, deeper tissue penetration, and higher spatial resolution.^[8–10] Nevertheless,

Cited from: Qi, J. *et al.* Real-Time and High-Resolution Bioimaging with Bright Aggregation-Induced Emission Dots in Short-Wave Infrared Region. *Adv. Mater.* **30**, 1706856 (2018), page 1–2.

Fluorescence imaging, a proven method for tracking biological targets with high sensitivity and in real-time, recently has been expanded to the near-infrared II (NIR-II, 1000–1700 nm) region, with higher contrast and deeper penetration compared with the well-established visible to NIR-I (400–900 nm) fluorescence imaging systems.^[1,2] Therefore, the majority

Cited from: Zhu, S. *et al.* Repurposing Cyanine NIR-I Dyes Accelerates Clinical Translation of Near-Infrared-II (NIR-II) Bioimaging. *Adv. Mater.* **30**, 1802546 (2018), page 1.

As for exposure time during bioimaging, frames per second, i.e. fps, is usually hired to evaluate the frame rate for video. To explore the proper exposure time, we chose 1 ms (fps 20.1) and 300 ms (fps of 1.52) exposure time to acquire the NIR-II bioimages of ICR mouse hindlimb under the excitation of 1064 nm (70 mW/cm²) with 1400 LP filter (**Figure R1-7**). Signal-to-background ratio (SBR), contrast and full width at half-maximum (FWHM) were hired to evaluate the clarity

of the images. Comparable contrast was found in images with 1 ms (0.42) and 300 ms (0.49) exposure time. SBR result with exposure time of 300 ms (8.82) was almost 1.5-fold than that of 1 ms (6.01). Moreover, the FWHM result with exposure time of 300 ms (345 μm) was about a quarter short than that of 1 ms (445 μm). These results illustrated that short exposure time could also be used for the bioimaging at the expense of SBR, contrast and FWHM reduction. **In our experiment, due to the relatively slow dynamic vascular changing, we chose 300 ms as exposure time to collect higher clarity of images using lower laser power density.**

Figure R1-7. a) NIR-II bioimages of shaved ICR mouse hindlimb under the excitation of 1064 nm (70 mW/cm^2) with 1400 LP filter. Exposure time was 1 ms (fps 20.1, left) and 300 ms (fps 1.52, right), respectively. b) Cross-sectional fluorescence intensity profiles along red-dashed line of the mouse hindlimb injected with LZ-1105 in (a). The profiles of images with 1ms and 300 ms exposure time were presented with black and red dots, respectively. Gaussian fits to the profiles are shown in red curves. c) Contrast of the red box in the LZ-1105 administrated mouse hindlimb images (a). Scale bar represents 4 mm.

Comment 2. Line 69-71 states that NIR-II has higher resolution contrast (sub – 10 μm) than visible and NIR-I region. But this depends upon the depth and system studied. There is no study demonstrating improved microscopy at these wavelengths that this reviewer is aware of (nor any that are referenced) and the statement is inaccurate. A careful study of contrast, signal to noise, resolution using standardized measurements in absolute traceable units (i.e., not AU) will be needed before such statements are made (or repeated).

Response: Thanks for the comments, we agree with the reviewer that the higher resolution

contrast of NIR-II bioimaging depends upon the depth and system studied. We have revised the corresponding sentence in the original manuscript from “Recent development in noninvasive fluorescence imaging in the second near-infrared window (NIR-II; 1000-1700 nm) has received considerable attention because of higher resolution contrast (sub-10 μm) and deeper tissue penetration depth (\sim 5-30 mm) compared to traditional visible light and first near-infrared window (400-900 nm) imaging techniques.” into “Recent development in noninvasive fluorescence in vivo imaging in the second near-infrared window (NIR-II; 1000-1700 nm) has received considerable attention because of deeper tissue penetration depth compared to traditional near-infrared window (780-900 nm) imaging techniques.”

To make it clear to the readers, we define the measurement of SBR, FWHM and contrast as follows (**Figure R1-8**).

Figure R1-8. Measurement method of signal-to background ratio, resolution and contrast in NIR-II bioimages. Scale bar represents 4 mm.

We also added related description in the “*Synthesis and optical characterization of the LZ series dyes for in vivo vasculature imaging*” section in the revised manuscript as follow:

“The full width at half maximum (FWHM), contrast and signal-to-background ratio (SBR) were used to measure the resolution and clarity of the images (see *Method* section and Supplementary Fig. 10).”

Meanwhile, we also added the related description in “*Method*” section in the revised manuscript and Supplementary Fig. 10 in the revised supplementary information).

(1). Signal-to-background ratio (SBR): When calculating SBR in NIR bioimaging, we draw a red dashed line in the region of interest (ROI), and the fluorescence intensity profiles across the line would be extracted and fitted by Gaussian fitting. The highest intensity on the Gaussian fitted curve is selected as “signal” (I_{signal}) and the intensity of the baseline on Gaussian fitted curve is selected as “background” ($I_{\text{background}}$). SBR is calculated as follow:

$$SBR = \frac{I_{\text{signal}}}{I_{\text{background}}}$$

(Refer to: Diao, S., *et al.* Fluorescence Imaging In Vivo at Wavelengths beyond 1500 nm. *Angew. Chem. Int. Ed.* **54**, 14758-14762 (2015); Wang, F. *et al.* Light-sheet microscopy in the near-infrared II window. *Nat. Methods* **16**, 545-552 (2019); Ding, B., *et al.* Polymethine Thiopyrylium Fluorophores with Absorption beyond 1000 nm for Biological Imaging in the

Second Near-Infrared Subwindow. *J. Med. Chem.* **62**, 2049-2059 (2018); Zhong, Y. *et al.* Boosting the down-shifting luminescence of rare-earth nanocrystals for biological imaging beyond 1500 nm. *Nat. Commun.* **8**, 737 (2017); Tsukasaki, Y. *et al.* A short-wavelength infrared emitting multimodal probe for non-invasive visualization of phagocyte cell migration in living mice. *Chem. Commun.* **50**, 14356-14359 (2014).)

(2). Full width at half-maximum (FWHM): FWHM is a parameter to evaluate resolving ability of the images. When evaluating FWHM, we draw a line in the ROI, and the FWHM is extracted from the fluorescence intensity profiles across a red line fitted by Gaussian fitting. (Refer to: Hong, G. *et al.* Multifunctional in vivo vascular imaging using near-infrared II fluorescence. *Nat. Med.* **18**, 1841-1846 (2012); Diao, S., *et al.* Fluorescence Imaging In Vivo at Wavelengths beyond 1500 nm. *Angew. Chem. Int. Ed.* **54**, 14758-14762 (2015); Ding, B., *et al.* Polymethine Thiopyrylium Fluorophores with Absorption beyond 1000 nm for Biological Imaging in the Second Near-Infrared Subwindow. *J. Med. Chem.* **62**, 2049-2059 (2018); Carr, J.A. *et al.* Shortwave infrared fluorescence imaging with the clinically approved near-infrared dye indocyanine green. *Proc. Natl. Acad. Sci. USA* **115**, 4465-4470 (2018); Qi, J. *et al.* Real-Time and High-Resolution Bioimaging with Bright Aggregation-Induced Emission Dots in Short-Wave Infrared Region. *Adv. Mater.* **30**, 1706856 (2018); Antaris, A.L. *et al.* A small-molecule dye for NIR-II imaging. *Nat. Mater.* **15**, 235-242 (2016); Ma, Z., *et al.* Near-Infrared IIb Fluorescence Imaging of Vascular Regeneration with Dynamic Tissue Perfusion Measurement and High Spatial Resolution. *Adv. Funct. Mater.* **28**, 1803417 (2018); Li, B., Lu, L., Zhao, M., Lei, Z. & Zhang, F. An Efficient 1064 nm NIR-II Excitation Fluorescent Molecular Dye for Deep-Tissue High-Resolution Dynamic Bioimaging. *Angew. Chem. Int. Ed.* **57**, 7483-7487 (2018).)

(3). Contrast: contrast was also a parameter to evaluate resolving ability, which was defined as the standard deviation (σ) divided by the mean pixel intensity (μ) in the red box. (Refer to: Carr, J.A. *et al.* Shortwave infrared fluorescence imaging with the clinically approved near-infrared dye indocyanine green. *Proc. Natl. Acad. Sci. USA* **115**, 4465-4470 (2018); Pawley, J.B. *Handbook of Biological Confocal Microscopy Ch. 4* (Springer, New York, 2006).)

There are several works in which the researchers have illustrated a higher SBR, resolution and shorter FWHM in NIR-II window than that of visible and NIR-I region. Please see **Comment 1** for details.

In addition, signal intensity in NIR-II bioimaging is the value of the analog-to-digital conversion of the circuit. Thus we use a.u. as the unit of the pixel signal intensity in NIR-II bioimaging results. This is also a common method to measure the signal intensity in NIR-II bioimaging results, as mentioned in the previous works (Refer to: Ma, Z. *et al.* Near-Infrared IIb Fluorescence Imaging of Vascular Regeneration with Dynamic Tissue Perfusion Measurement and High Spatial Resolution. *Adv. Funct. Mater.* **28**, 1803417 (2018); Ding, B., *et al.* Polymethine Thiopyrylium Fluorophores with Absorption beyond 1000 nm for Biological Imaging in the Second Near-Infrared Subwindow. *J. Med. Chem.* **62**, 2049-2059 (2018); Zhu, S. *et al.* Repurposing Cyanine NIR-I Dyes Accelerates Clinical Translation of Near-Infrared-II (NIR-II) Bioimaging. *Adv. Mater.* **30**, 1802546 (2018); Zhu, S., *et al.* Molecular imaging of biological systems with a clickable dye in the broad 800-to 1,700-nm near-infrared window. *Proc. Natl. Acad. Sci. USA* **114**, 962-967 (2017).)

Comment 3. NIR-II has limited penetration depth due to the increased water absorption as

compared to NIR-I and to this reviewer's knowledge, no published study has been successfully performed on an animal model larger than a rodent. Finally the NIR-I wavelength range is 780-900 nm as below 780 nm, light is visible.

Response: As we explained in *Comment 1*, besides tissue/water absorption, there are more factors which would influence the penetration ability including scattering and tissue autofluorescence. Moreover, previous reports have illustrated that NIR-II bioimaging has deeper penetration depth in the same imaging system (Cited from: Hong, G. *et al.* Multifunctional in vivo vascular imaging using near-infrared II fluorescence. *Nat. Med.* **18**, 1841-1846 (2012); Tsukasaki, Y. *et al.* A short-wavelength infrared emitting multimodal probe for non-invasive visualization of phagocyte cell migration in living mice. *Chem. Commun.* **50**, 14356-14359 (2014).).

Although the penetration of NIR-II does not allow to penetrate much thicker tissue in animal model larger than a rodent, this window enables non-invasive detection of fluorophores at both the cortical and subcortical level in small animal models, which can broaden the understanding of the mechanisms underlying impaired neurovascular and neurometabolic regulation.

Comment 4. While ICG has a short blood half-life due to hepatobiliary clearance, it associates with albumin, making it a good blood pooling agent that does not extravasate. (By the way, there are several efforts to prolong blood half-life by incorporating ICG in liposomes.). Yet the small molecule LZ-1105 is stated NOT to associate with blood proteins, yet why does it not extravasate? This is a significant question.

Response: Thanks for the comments, we apologize for our inappropriate statement of LZ-1105 state in blood. In our previous isothermal titration calorimetry (ITC) experiment, we illustrated that LZ-1105 have no interaction with bovine serum albumin (BSA). While in fact, it has interaction with blood (as we have mentioned in Figure 2d in our original manuscript). Moreover, we noticed that after mixing with mouse blood, the fluorescence intensity of LZ-1105 increased about 23-fold than that of in PBS (**Figure R1-9** and **Figure 2g** in the revised manuscript). Then we found that the fluorescence intensity of LZ-1105 also increased about 19-fold than that of in PBS after mixing with bovine fibrinogen, demonstrating LZ-1105 have interaction with fibrinogen, which might explain the intensity enhancement after adding LZ-1105 into mouse blood. ITC experiment was then taken to evaluate the interaction between LZ-1105 and bovine fibrinogen. The binding parameter of LZ-1105 with bovine fibrinogen was 15413 L/mol, which was comparable with that of LZ-1105 with blood (**Figure R1-10** and **Figure 2h** in the revised manuscript). Thus, LZ-1105 would also act as a blood pooling agent and would not extravasate.

Figure R1-9. Optical characterization of ICG and LZ-1105 in various media. The fluorescence intensity of ICG and LZ-1105 in mice blood detected by the InGaAs charge-coupled device (CCD) (808 nm excitation for ICG, 38 mW/cm², 1300 nm long-pass filter; 1064 nm excitation for LZ-1105, 30 mW/cm², 1400 nm long-pass filter). Insets: NIR-II fluorescence images of ICG and LZ-1105 with different excitations and long-pass filters. ([LZ-1105] = [ICG] = 10 μM)

Figure R1-10. Measurement of the LZ-1105 and ICG binding ability with bovine fibrinogen. The isothermal titration calorimetry (ITC) results of LZ-1105 and ICG with bovine fibrinogen at 20°C.

Figure R1-11 (Fig. 2 in the revised manuscript). Pharmacokinetics and blood retention of LZ-1105 and ICG. **a-c** NIR-II bioimaging of mice body (**a**), brain (**b**) and hindlimb (**c**) after i.v. injected with LZ-1105 (1400 nm long-pass, $\lambda_{ex}=1064$ nm) and ICG (1300 nm long-pass, $\lambda_{ex}=808$ nm). **d** The corresponding signal-to-background ratio (SBR) of LZ-1105 and ICG administrated mice along the yellow and cyan dashed line in (**b**) and (**c**) as a function of time. The blue dotted line indicates the Rose criterion. **e** Blood circulation ($\%ID\ g^{-1}$) of LZ-1105 and ICG administrated mice as a function of time. **f** Cumulative urine excretion curve for LZ-1105 injected mice within 24 h p.i.. **g** NIR-II signal intensity of LZ-1105 and ICG with different excitations and long-pass filters. (1064 nm excitation and 1400 nm long-pass filters for LZ-1105, 808 nm excitation and 1300 nm long-pass filters for ICG. Excitation power was adjusted to obtain comparable NIR-II signal in mice blood. $[LZ-1105] = [ICG] = 10\ \mu M$) **h** The binding parameters of LZ-1105 and ICG

with mice blood, bovine serum albumin (BSA) and bovine fibrinogen by isothermal titration calorimetry (ITC) at 20°C. N/A means not measurable by ITC. Error bars: mean \pm s.d. (n=3). Scale bars in Fig2a, b and c represent 4 mm.

To make it clear for the readers, we added related description to explain the interaction between LZ-1105 and mice blood, moreover, we reorganized the “*In vivo pharmacokinetics and blood retention*” section in Page 7-9 in the revised manuscript as follow:

“After intravenous injection (i.v.) of LZ-1105 (5 mg/kg), abdominal, brain and hindlimb NIR-II imaging of mice at various time points were performed (Fig. 2a-c). The vasculature was clearly visualized within 6 h post-injection (p.i.), and almost no fluorescence signal could be observed in the bladder and liver. Although the fluorescence signal of vasculature gradually decreased with time, the SBR remained above 5.0 even at 3 h p.i., surpassing the Rose criterion to distinguish image features with 100% certainty²⁵ (Fig. 2d). Therefore, the blood vessels of the abdomen, brain and hindlimb could be obviously distinguished from the background noise, ensuring the long-term observation of dynamic vascular structures within 3 h. In comparison, the blood vessels of abdomen, brain and hindlimb could only be visualized within 5 min after injection of ICG (5 mg/kg), and almost no fluorescence signal of vasculature could be observed at 10 min, which could not be utilized for long-term vascular imaging by single injection.

Then the pharmacokinetics in living mice was investigated by measuring the LZ-1105 and ICG contents in venous blood samples at various time points, which exhibited a long blood half-life time of 195.4 min for LZ-1105 and a short α half-life time of 3.1 min for ICG (Fig. 2e and Supplementary Fig. 29-30). In addition, clearance pathways were different between ICG and LZ-1105. ICG was rapidly captured by the liver within 30 min and remained for 6 h followed by hepatobiliary clearance²⁹, while only weak fluorescence signal could be detected in urine of LZ-1105 injected mouse. Unexpectedly, after mixing with mice blood, an 11.3-fold signal enhancement was observed in urine (Supplementary Fig. 31), demonstrating the free state of LZ-1105 molecule in urine. Around 86% of the injected dose (% ID) of LZ-1105 was found in urine within 24 h p.i. with an elimination rate constant of 0.079 h⁻¹ (Fig. 2f and Supplementary Fig.32-33). Besides, ex vivo images of main organs including the liver, heart, spleen, kidney, lung, stomach, intestine, skin, muscle, brain and bone of LZ-1105 injected mice at 1 to 24 h p.i. showed extremely low organ uptake (Supplementary Fig. 34).

To explore the possible reason for long-term blood circulation property of LZ-1105, mice blood, bovine serum albumin (BSA), bovine fibrinogen and other medias were mixed with ICG and LZ-1105 (Fig. 2g and Supplementary Fig. 35). Under excitation, NIR-II signal were collected by different wavelength long-pass filters. ICG exhibited a 23-fold signal enhancement in BSA solution than that of in PBS, and the signal intensity was comparable with that of in mice blood. These results illustrated that after mixing with mice blood, binding with BSA was the main reason for ICG signal increase, which was also consistent with previous report²⁹. Interestingly, for LZ-1105, fluorescent signal only increased obviously in the presence of bovine fibrinogen. To further illustrate the interaction between LZ-1105 and bovine fibrinogen, isothermal titration calorimetry (ITC) was utilized to measure its binding ability (Fig. 2h, Supplementary Fig. 36-38 and Supplementary Table.4). Both LZ-1105 and ICG had interaction with blood, but in contrast to the obviously binding feature of ICG with BSA, scarcely any interaction between LZ-1105 and BSA could be observed. Meanwhile, the binding parameter of LZ-1105 with bovine fibrinogen

was 18140 L mol^{-1} , which was comparable with that of LZ-1105 with blood. No interaction between ICG and bovine fibrinogen could be observed. These were consistent with the fluorescence imaging results (Supplementary Fig. 35), illustrating the different binding type between LZ-1105 and ICG with blood. These results might be used to explain the long-term blood circulation feature of LZ-1105. The binding parameter of LZ-1105 with blood (15413 L/mol) was approximately 15.5-fold lower than that of the ICG (238708 L/mol). Thus the separation between LZ-1105 and blood was speculated to be easier than that of ICG due to the weaker binding with blood. During the circulation, separation between LZ-1105 and fibrinogen might occur in kidney, leading to the renal exertion of LZ-1105 molecules. With long-term blood circulation, low cytotoxicity and the superior SBR and resolution, LZ-1105 provided a promising method for continuously real-time imaging of vascular structural changes and physiological process.”

Comment 5. The addition of BSA to ICG reduces quenching caused by the stacking of the planar ICG structure and the supplementary Fig 15 really has little to do with interactions with blood proteins as the authors claim. Is LZ-1105 subject to quenching at high concentrations as ICG is?

Response: Thanks for the comments. As the reviewer mentioned, we did observe that ICG would be quenched at high concentration in water phase and the fluorescence intensity reached the maximum when the concentration was $4 \mu\text{M}$. After mixing with blood, maximum fluorescence intensity was obtained when the concentration reached $13 \mu\text{M}$. **Similar with ICG, LZ-1105 tends to be quenched in water phase at high concentration.** The fluorescence spectra of LZ-1105 at various aqueous concentration were shown as **Figure R1-12** and Supplementary Fig. 8 in the revised supplementary information ($\lambda_{\text{ex}}=1064 \text{ nm}$). In deionized water, LZ-1105 fluorescence intensity decreased when the concentration was beyond $12 \mu\text{M}$. However, after mixing with blood, LZ-1105 would bind to bovine fibrinogen and the maximum fluorescence intensity was observed at the concentration of $27 \mu\text{M}$. This may be due to the binding interaction between LZ-1105 and fibrinogen, which reduced the quenching effect by the stacking of the planar LZ-1105 structure.

Figure R1-12. Fluorescence spectra of ICG and LZ-1105 at various in different media. Fluorescence spectra of ICG and LZ-1105 at various concentration in water (ICG (a) and LZ-1105(e)) and blood (ICG (c) and LZ-1105 (g)). Fluorescence intensity at 794 nm for ICG in water (b) and in blood (d). Fluorescence intensity at 1105 nm for LZ-1105 in water (f) and in blood (h) as a function of incubation time.

To make it clear to the readers, we added related description in the “*Synthesis and optical characterization of the LZ series dyes for in vivo vasculature imaging*” section in the revised manuscript as follow:

“Meanwhile, similar with ICG, after mixing LZ-1105 with blood the quenching effect is reduced by the stacking of planar structure (Supplementary Fig. 8)”

6. Line 88: *There was no traceable measures of image resolution or depth presented using ICG and LZ-1105 that justifies this statement that LZ-1105 is far superior to ICG. In fact, one might argue that the ICG imaging was performed in a disadvantaged way, in that the emission from 808 nm excitation should have collected with a long pass 1064 nm filter. Indeed, the ICG emission collection missed the majority of signal, which make this an unfair test. Line 366 should be struck.*

Response: Thanks for the comments. To make it clear to the readers, in the revised manuscript various long-pass filters (850-, 1000-, 1100-, 1200-, 1300- and 1400 nm) were used for in-vitro and in-vivo ICG imaging.

First, single ICG-filled (mixed with blood, [ICG] = 10 μ M) capillary immersed into 1% Intralipid was excited at 808 nm and the emission was collected by different long-pass filters at varied Intralipid depths (**Figure R1-13** and Supplementary Fig. 11 in the revised supplementary information). The imaging signal intensity with different long-pass filters at 0 mm penetration depth was almost the same by adjusting the laser's working power density. With increase of penetration depth, attenuation of signal intensity and blurring of capillary profiles were observed for all images, and only images acquired beyond 1300 nm resolved sharp edges of the capillary at a depth up to 5.5 mm. 1300 nm long-pass filter group (19.0) exhibited a 10.6-fold signal-to-background ratio (SBR) than that of 850 nm long-pass filter group (1.8) under 1.0 mm depth. Besides, FWHM, which was used to evaluate the resolving ability, was observed to increase more obviously by using 850 nm long-pass filter. **These results illustrated that ICG imaging acquired beyond 1300 nm exhibited higher signal-to-background ratio, resolution and deeper penetration compared to that of 850 nm in the presence of biotissue.**

Figure R1-13. Deep penetration of ICG-filled single capillary in mimic tissue. a) Fluorescence images of capillaries filled with ICG in blood ($[ICG] = 10 \mu M$) immersed in 1% Intralipid with varying depth. Imaging signals were collected with various long-pass filters under 808 excitation. b-f) Profiles measured at same position in the capillary in various long-pass group at different

depth. g) Measured SBR of capillary images as a function of depth. h) Wavelength-dependent FWHM of cross-sectional profiles in capillary images as a function of depth. The bars represent mean \pm s.d. derived from the uncertainty in the Gaussian fitting of feature width.

To further investigate the resolving ability in the presence of biotissue with 1300 nm long-pass filter, two ICG-filled (mixed with blood, [ICG] = 10 μ M) capillaries were chosen to mimic adjacent blood vessels in vivo. Two capillaries immersed into 1% Intralipid was excited at 808 nm and the emission was collected by different long-pass filters at varied depths (**Figure R1-14** and Supplementary Fig. 12 in the revised supplementary information). Similar with the single capillary imaging, attenuation of image intensities and blurring of capillary profiles were observed for all images with increase of penetration depth. Contrast of images acquired by various long-pass filters under different depth was calculated. The resolving ability could also be evaluated by contrast, which has been defined in *Comment 2*. Contrast of various long-pass filters groups was comparable when penetration depth was 0 mm. However, contrast of 850 nm long-pass group dropped below 0.06 under 3.0 mm depth, and the two capillaries could not be resolved from each other. In contrary, contrast of 1300 nm long-pass group stay above 0.2 even under 6.5 mm depth, demonstrating that images acquired by 1300 nm long-pass filter exhibited higher resolution in the presence of biotissue.

Figure R1-14. Deep penetration of ICG-filled double capillaries in mimic tissue. a) Fluorescence images of capillaries filled with ICG in blood ([ICG] = 10 μ M) immersed in 1% Intralipid with varying depth. Imaging signals were collected with various long-pass filters under 808 excitation. b) Wavelength-dependent contrast of capillary images as a function of depth. c-g) Profiles measured at same position in the capillary in various long-pass groups at different depth.

Next, mice skin was used to repeat the above experiment and similar results could be obtained

that ICG bioimaging has higher SBR and contrast by using longer wavelength long-pass filter (**Figure R1-15** and Supplementary Fig. 13 in the revised supplementary information).

Figure R1-15. Deep penetration of ICG-filled single capillary in mice skin. a) Fluorescence images of capillaries filled with ICG in blood ($[ICG] = 10 \mu M$) immersed in mice skin. Imaging signals were collected with various long-pass filters under 808 excitation. b) Wavelength-dependent contrast of capillary images with or without mice skin. c) Wavelength-dependent SBR of capillary images with or without mice skin. d-e) Profiles measured along the red dashed line in the capillary in various long-pass group with or without mice skin.

Encouraged by the above results, we further investigated the SBR, resolution and contrast of ICG for in vivo imaging with various long-pass filters. Different 808 nm laser power density was used during the imaging to obtain the same intensity at same signal ROI. As a result, background noise and FWHM decreased while SBR and contrast increased when using the longer long-pass filter, illustrating the superior bioimaging results could be achieved by long-pass filter. When 1400 nm long-pass filter was used, SBR, contrast and FWHM reached the optimum (SBR: 2.5, contrast: 0.29, FWHM: 132 μm) (**Figure R1-16-R1-17** and Supplementary Fig. 14-15 in the revised supplementary information). However, the laser power density of 808 nm reached 500 mW/cm^2 ,

which could lead to strong heating effect of the tissue (**Figure R1-18** and Supplementary Fig. 16 in the revised supplementary information). Same results were obtained in the bioimaging of hindlimb (**Figure R1-19-R1-20** and Supplementary Fig. 17-18 in the revised supplementary information). Thus 1300 nm long-pass filter was chosen as the proper long-pass filter for ICG imaging.

Figure R1-16. NIR bioimaging of ICG injected mice brain with various long-pass filters. a) NIR bioimaging of balb/c nude mice brain with various long-pass filters by ICG administration (up row: images were acquired under 808 nm excitation with same power density; bottom row: images were acquired under 808 nm excitation with different power density to obtain the same signal intensity, contrast was annotated as red letters). b-d) SBR (b), contrast (c) and FWHM (d) of cross-sectional profiles in balb/c nude mice brain images with various long-pass filters. Scale bar represents 4 mm.

Figure R1-17. FWHM of ICG injected mice brain obtained by various long-pass filters. The fluorescence intensity profiles (dots) and Gaussian fit (lines) along the red-dashed line in brain with various long-pass filters.

Figure R1-18. Thermal effect experiments. Thermal effect experiments of 808 nm laser (70 or 500 mW/cm²) irradiation on balb/c nude mice back at 23.5 °C using FLIR A315 thermal imaging camera. a) Photograph of the experimental setup. b) Photothermal images of mice back with different laser working power density for different irradiation time. The radiation area was shown with yellow circle. c) Thermal effect for mice skin during 30 min irradiation of laser as a function of time. Scale bar represents 3 mm.

Figure R1-19. NIR bioimaging of ICG injected mice hindlimbs with various long-pass filters. a) NIR bioimaging of balb/c nude mice hindlimb with various long-pass filters by ICG administration (up row: images were acquired under 808 nm excitation with same power density; bottom row: images were acquired under 808 nm excitation with different power density to obtain the same signal intensity, contrast was annotated as red letters). b-d) SBR (b), contrast (c) and FWHM (d) of cross-sectional profiles in balb/c nude mice hindlimb images with various long-pass filters. Scale bar represents 4 mm.

Figure R1-20. FWHM of ICG injected mice brain obtained by various long-pass filters. The fluorescence intensity profiles (dots) and Gaussian fit (lines) along the red-dashed line in hindlimb with various long-pass filters.

Furthermore, there is a similar work which reported that higher SBR and contrast in NIR-II imaging was obtained with longer filters after administration of ICG (**Figure R1-21**, refer to: Carr, J.A. *et al.* Shortwave infrared fluorescence imaging with the clinically approved near-infrared dye indocyanine green. *Proc. Natl. Acad. Sci. USA* **115**, 4465-4470 (2018) ; Starosolski, Z., *et al.* Indocyanine green fluorescence in second near-infrared (NIR-II) window. *PLOS ONE* **12**, e0187563 (2017).).

Figure R1-21. The contrast, defined as the SD (σ) divided by the mean (μ) pixel intensity, was quantified to compare NIR and SWIR imaging of the brain vasculature and hindlimb vasculature

of a mouse. The contrast was found to be 0.20 for NIR-imaged brain vasculature and 0.29 for SWIR-imaged brain vasculature. The contrasts were 0.12 and 0.19 for NIR- and SWIR-imaged hind-limb vasculature, respectively. (Cited from: Carr, J.A. *et al.* Shortwave infrared fluorescence imaging with the clinically approved near-infrared dye indocyanine green. *Proc. Natl. Acad. Sci. USA* **115**, 4465-4470 (2018).)

In addition, the proper (1400 nm) long-pass filter for LZ-1105 was also selected by the above method. The data were shown as **Figure R1-22-R1-23** and Supplementary Fig. 19 and 22 in the revised supplementary information. Thus, the proper long-pass filters were used for both ICG (1300 nm long-pass filter) and LZ-1105 (1400 nm long-pass filter) during the imaging. The comparison between ICG and LZ-1105 is fair, and LZ-1105 imaging is superior to ICG for in-vivo bioimaging, as we mentioned in our original manuscript.

Figure R1-22. NIR-II bioimaging of LZ-1105 injected mice brain with various long-pass filters. a) NIR bioimaging of balb/c nude mice brain with various long-pass filters by LZ-1105 administration under 1064 nm excitation. b) SBR of balb/c nude mice brain imaging with various long-pass filters. c) Contrast of brain vessels acquire by various long-pass filters. d) FWHM of Gaussian fitted fluorescence intensity profiles of brain vessels acquire by various long-pass filters. Scale bar represents 4 mm.

Figure R1-23. NIR-II bioimaging of LZ-1105 injected mice hindlimbs with various long-pass filters. a) NIR bioimaging of balb/c nude mice hindlimb with various long-pass filters by LZ-1105 administration under 1064 nm excitation. b) SBR of balb/c nude mice hindlimb imaging with various long-pass filters. c) Contrast of hindlimb vessels acquire by various long-pass filters. d) FWHM of Gaussian fitted fluorescence intensity profiles of hindlimb vessels acquire by various long-pass filters. Scale bar represents 4 mm.

To make it clear for the readers, we added the related description to explain the reason of choosing 1300 nm- and 1400 nm long-pass filters for ICG and LZ-1105, respectively, in “*Synthesis and optical characterization of the LZ series dyes for in vivo vasculature imaging*” section in the revised manuscript as follow:

“Before administrating ICG and LZ-1105 for NIR-II imaging, signal collection condition was optimized in vitro and in vivo. 1300 nm and 1400 nm long-pass filters were finally selected to acquire optimal NIR-II imaging for ICG and LZ-1105, respectively (Supplementary Fig. 11-22).”

Besides, we added “Optimizing the signal collection for ICG and LZ-1105 NIR imaging” section in the revised supplementary information in Page 12.

7. Line 92: *The authors state that LZ-1105 is capable of real-time quantification of femoral artery blood flow, when in fact, they simply use the advancing front of the dye to assess a velocity with a bolus administration and no volumetric flow is quantitated. This is not a viable means of measuring blood flow rate as currently available, for example with Doppler US. In addition, the limitations in depth penetration caused by water absorption will restrict this application from clinical use, even if there were some application that could be found for assessing superficial assessment of perfusion. Line 97 should be struck as there is no evidence that sufficient depths can be imaged to make NIR-II clinically viable. Line 350 should be struck as it has not been shown in this manuscript that one can use the technology to detect changes in blood flow in response to drugs or stimuli without having to inject contrast agents repeatedly.*

Response: Thanks for the comments. In the ischemia reperfusion model, the advancing front of the dye is able to represent the blood flow front. This is also a common method to measure the blood flow velocity, as mentioned in the previous works (Real-Time and High-Resolution Bioimaging with Bright Aggregation-Induced Emission Dots in Short-Wave Infrared Region. *Adv. Mater.* **30**, 1706856 (2018); Wan, H., et al. A bright organic NIR-II nanofluorophore for three-dimensional imaging into biological tissues. *Nat. Commun.* **9**, 1171 (2018); Zhong, Y. *et al.* Boosting the down-shifting luminescence of rare-earth nanocrystals for biological imaging beyond 1500 nm. *Nat. Commun.* **8**, 737 (2017); Hong, G., et al. Multifunctional in vivo vascular imaging using near-infrared II fluorescence. *Nat. Med.* **18**, 1841-1846 (2012).).

Besides, Doppler Ultrasound (US) and NIR-II bioimaging by LZ-1105 administration were both hired to measure the blood flow velocity of healthy shaved ICR mice (**Figure R1-24**). Blood flow velocity was detected as $5.87 \pm 0.12 \text{ cm s}^{-1}$ by Doppler US, which is consistent with the NIR-II bioimaging results ($5.3684 \pm 0.0025 \text{ cm s}^{-1}$), illustrating the reliability of blood flow velocity detection through NIR-II bioimaging by LZ-1105. Furthermore, blood flow velocity of ischemia reperfusion hindlimb was low and beyond the detection of Doppler US, demonstrating the NIR-II bioimaging proved a broader dynamic range of blood flow velocity measurements in vivo. These results were consistent with the previous report. (Hong, G., *et al.* Multifunctional in vivo vascular imaging using near-infrared II fluorescence. *Nat. Med.* **18**, 1841, (2012).)

Figure R1-24. a) Ultrasound measurement of blood flow velocity in shaved ICR mouse hindlimb (left) based on integration of three outlined pulses (right). b) Time course NIR-II bioimages showing the flow front (marked by yellow arrows) in the hindlimb. c) Distance traveled by the flow front as a function of time. The slope of the function is calculated as the blood flow velocity.

In order to avoid misleading to readers, we have revised Line 97 from “Thus, a single NIR-II

imaging modality with the long-term angiography probe enables multifunctional imaging capable of accomplishing what is difficultly done by several traditional biomedical imaging techniques.” into “Thus, NIR-II imaging modality with the long-term angiography probe might help researchers to understand some dynamic vascular changing process on the basis of the traditional biomedical imaging techniques.” in the revised manuscript.

We admit that the NIR-II bioimaging method could only measure the blood flow velocity when the blood flow appears after injury. Just as the reviewer commented, this method cannot be used to detect changes in blood flow in response to drugs or stimuli without multiple injection. Thus, we revised Line 350 from “Our long-term blood circulation dye like LZ-1105 may allow quantification and tracking of certain areas of body by detecting changes in blood flow in response to external stimuli, which would allow researchers to monitor the animals over time without the need to inject contrast agents intermittently.” into “Our long-term blood circulation dye may allow researchers to monitor the animals over time without the need to inject contrast agents intermittently during some long-time reperfusion experiment.” in the revised manuscript.

8. Line 117: The term “Imaging accuracy” is used incorrectly. Imaging accuracy is determined by sensitivity and specificity in a receiver-operator characteristic study. This should be removed. Indeed, the resolution (which is what is being probed here) is dependent upon the emission level, i.e., what is the pW/sr/cm² of emission light collected? Clearly, there is limited ICG emission collected due to the sub-optimal filtering and this result will clearly change when ICG emission is collected optimally.

Response: Thanks for the comments. We have deleted related description of “imaging accuracy” and revised Line 114-117 from “When capillary tubes filled with LZ-1105 or ICG solution were immersed in 1 % Intralipid solution at increased phantom depth, the sharp tube edges for LZ-1105 demonstrated a better imaging accuracy even at a 6 mm immersion depth compared to ICG.” into “When capillary tubes filled with LZ-1105 or ICG solution were immersed in 1 % Intralipid solution at increased phantom depth, bioimaging results of LZ-1105 resolve sharper edges of the capillary at a depth up to 6 mm than that of ICG.” in the revised manuscript.

pW/sr/cm² is the signal intensity unit for imaging in visible region. For NIR bioimaging, emission light was collected by the InGaAs charge-coupled device (CCD), signal intensity for NIR-II bioimaging is the value of the analog-to-digital conversion of the circuit. Thus we use a.u. as the unit of the pixel signal intensity. The signal intensity depends on the excitation power density and exposure time. As we have indicated in **Figure R1-7, R1-16, R1-19 and R1-22-R1-23**, the emission intensity did decrease when using longer wavelength long-pass filter or longer exposure time. The reviewer commented it is the limited ICG emission collected that caused the low resolution. However, besides the emission intensity, the resolution is also influenced by the tissue scattering and the autofluorescence. The stronger intensity obtained by 850 nm long-pass filter will be attenuated in the presence of tissue induced scattering and autofluorescence. As we explained in the **Comment 6**, resolution of ICG in vivo imaging can reach the maximum only when using 1300 nm long-pass filter.

9. Lines 135-137 Lymphatic vessels are not visible without contrast. Hence it is difficult to

understand how lymphatic vessel diameter can be determined non-invasively or invasively from white light photography.

Response: Thanks for the comments. We apologize for the unclear expression in our original manuscript. We have revised Line 135-137 from “Furthermore, the widths of the vessels at the same position were measured as 130 μm with NIR-II imaging and 126 μm with white light optical photograph, respectively, demonstrating the precise imaging ability of LZ-1105 in NIR-II window (Supplementary Fig.10).” into “Furthermore, the widths of the hindlimb blood vessels at the same position were measured as 130 μm with NIR-II imaging and 126 μm with white light optical photograph, respectively, demonstrating the precise imaging ability of LZ-1105 in NIR-II window (Supplementary Fig. 28).”

10. What is the stability of LZ-1105? None is shown in serum at 37C for the time-frame of the animal studies. How is renal clearance occurring if the urine and bladder are dimly fluorescent?

Response: Thanks for the comments. **The stability of LZ-1105 in serum and other media at 37 °C was measured for 180 min and the results were shown as Figure R1-25** and Supplementary Fig. 9 in the revised supplementary information. The fluorescence intensity of LZ-1105 remained stable after incubated with serum at 37 °C for 180 min, illustrating the photostability of LZ-1105 in serum.

The fluorescence signal intensity of the bladder was weak during the NIR-II imaging and the urine collected also presented weak fluorescence signal intensity. However, when blood was added in the urine collected at post-injection of LZ-1105, the fluorescence intensity increased about 11.3-fold, which was in consistent with the 19.4-fold increase when LZ-1105 was mixed with blood (Supplementary Fig. 31 and 35 in the revised supplementary information), illustrating that LZ-1105 has not been destructed and was excreted from the body by renal clearance.

As we mentioned in *Comment 4*, LZ-1105 binds with fibrinogen immediately after i.v. injection. Then the blood intensity decreased continuously with time. We presumed that during the blood circulation, the LZ-1105 molecules separated with fibrinogen in kidney and excreted from the body in urine.

Figure R1-25. Photostability of LZ-1105 and ICG. LZ-1105 and ICG in a variety of biological media under continuous 1064 nm and 808 nm lasers excitation exposure for 180 min (330 mW/cm²), respectively.

To make it clear to the readers, we revised “Moreover, superior photostability of LZ-1105 was observed by exposing ICG and LZ-1105 in water, PBS (pH = 7.4) and mice blood to the continuous laser irradiation for 3 h, respectively (Supplementary Fig.8).” into “Moreover, superior photostability of LZ-1105 was observed by exposing ICG and LZ-1105 in blood, PBS (pH = 7.4), deionized (DI) water, serum and urine to the continuous laser irradiation for 3 h, respectively (Supplementary Fig. 9).” in the revised manuscript.

11. Cytotoxicity could be associated with triplet states, especially with the closeness of π orbitals. An assessment of lifetime decays would be most appropriate.

Response: Thanks for the comments. The lifetime decay of LZ-1105 in water was measured as 0.15 ns under the excitation of 1064 nm laser (Figure R1-26 and Supplementary Fig. 24 in the revised supplementary information). The lifetime of LZ-1105 was in nanoseconds rather than microseconds, indicating the triplet state was not involved in the luminescence process at 1105 nm. Thus LZ-1105 presented low phototoxicity caused by the generation of singlet oxygen under irradiation of 1064 nm.

Figure R1-26. Lifetime of LZ-1105 in water phase. Luminescence decay curve measured at 1105 nm under the excitation of 1064 nm.

To make it clear to the reviewers and the readers, we added the related description in the “*Synthesis and optical characterization of the LZ series dyes for in vivo vasculature imaging*” section in the revised manuscript as follow:

“Moreover, the lifetime of LZ-1105 in water was measured as 0.15 ns under the excitation of 1064 nm laser (Supplementary Fig. 24). The nanosecond lifetime illustrated that the triplet state was not involved in the luminescence process at 1105 nm, and the phototoxicity caused by the generation of singlet oxygen could be excluded.”

12. Line 171 -173 makes little sense. Instability of LZ-1105 needs to be performed in a stability study, not from assessing shift of fluorescence spectra.

Response: Thanks for the comments. Stability of LZ-1105 in urine was investigated in Figure R1-28. The fluorescence intensity of LZ-1105 remained stable after incubated with urine at 37 °C for 180 min, illustrating the stability of LZ-1105 in urine.

To make it clear to the reviewers and the readers, we deleted “Fluorescence emission peak shift of the urine was not detected, indicating that LZ-1105 molecule was not destructed or degraded in the elimination process (Supplementary Fig.16).” Besides, we added related description in *Method* section as follow:

“Moreover, superior photostability of LZ-1105 was observed by exposing ICG and LZ-1105 in blood, PBS (pH = 7.4), deionized (DI) water, serum and urine to the continuous laser irradiation for 3 h, respectively (Supplementary Fig. 9)”

13. Line 217-219 claiming superior and abundant information needs to be struck. Line 240-242 claiming use of the technology in clinical practice should be struck. Line 274-275, “. . . surpasses current medical modalities” is grandstanding and not appropriate for a scientific journal. Line 341 claiming the technique allows clinicians to visualize real-time treatment

efficacy of drugs should be struck.

Response: Thanks for the comments. We have revised Line 217-219 from “Therefore, LZ-1105 performed NIR-II imaging proved a superior observation and abundant information of *in vivo* ischemic reperfusion with real-time feedback and high spatial resolution.” into “**Therefore, LZ-1105 performed NIR-II imaging could prove instantaneous BFV during the *in vivo* ischemic reperfusion with real-time feedback and high spatial resolution.**” in our revised manuscript.

We have deleted Line 240-242 “This real-time vessel monitoring method could be utilized in clinical practice for precise thrombolysis and other vascular diseases treatment which require continuous observation.” in our revised manuscript.

We have revised Line 273-276 from “NIR-II imaging technique simultaneously provides anatomical and hemodynamic information and surpasses current medical imaging modalities, owing to reduced tissue scattering, deeper anatomical penetration and non-radiative effect.” into “**NIR-II imaging technique simultaneously provides anatomical and hemodynamic information owing to reduced tissue scattering, deeper anatomical penetration and non-radiative effect.**” in our revised manuscript.

We have deleted Line 340-342 “Our approach allows clinicians to visualize the real-time treatment efficiency of drugs and provide proper dose for treatment.” in our revised manuscript.

14. Lines 277: Long blood half-life is needed ONLY for blood pooling agents. The authors confuse their application with the rest of medical imaging that requires short blood half-lives for maximum contrast.

Response: Thanks for the comments. We have revised Line 276-278 from “For *in vivo* monitoring the dynamic physiological process in NIR-II imaging window, two main prerequisites of the contrast agents should be achieved: long absorption/emission and long blood circulation.” into “**For *in vivo* monitoring the dynamic vascular-related physiological process in NIR-II imaging window, two main prerequisites of the contrast agents should be achieved: long absorption/emission and long blood circulation.**” In the revised manuscript.

15. Line 290: The shorter excitation of ICG eliminates water absorption and enables greater penetration, albeit at reduced resolution to due greater scattering. Indeed, this reviewer finds that ICG, when optimally used in NIR-I windows (as used in some clinical applications) outperforms ICG when probed in NIR-II windows --- when compared with traceable measurements.

Response: Thanks for the comments. We agree with the reviewer that in some clinical applications, ICG performs better properties in NIR-I window. However, as we mentioned in **Comment 6**, when ICG is used for *in vivo* bioimaging, NIR-II window is more appropriate due to the higher resolution and SBR. Thus we have revised Line 288-291 from “Although the only clinically approved ICG may function as NIR-II contrast agent with emission tail extending into NIR-II region, it still suffers from shorter excitation, moreover, the signal intensity of the emission tail is too low to obtain images with high SBR and resolution.” into “**Although the only clinically approved ICG may function as NIR-II contrast agent with emission tail extending into NIR-II**

region, the signal intensity of the emission tail is too low to obtain images with high SBR and resolution.” in the revised manuscript.

16. Paragraph starting on line 329 is not pertinent to the manuscript.

Response: Thanks for the comments. We have deleted this paragraph in our revised manuscript.

Other points:

1. In the abstract LZ-1105 is termed a molecular agent, when it is actually used as a blood pooling agent without any targeting moiety used. Short blood half-lives are desired for molecular agents since this increases target to background ratios. The authors should not refer to LZ-1105 as a molecular agent.

Response: Thanks for the comments. We have revised “molecular agent” into “molecule” in our revised manuscript.

2. Line 78: ICG is clinically used, but not approved for biomedical optical imaging.

Response: Thanks for the comments. We have revised Line 77-80 from “Up to now, indocyanine green (ICG) is the only clinically approved dye for in vivo NIR bioimaging, and it has been recently reported for NIR-II imaging with emission tails extending into NIR-II region” into “Indocyanine green (ICG), a clinically approved dye, has been recently reported for NIR-II imaging with emission tails extending into NIR-II region.” in our revised manuscript.

3. Line 471: meaning? Perhaps grammatical corrections would convey meaning.

Response: Thanks for the comments. We have revised Line 471-472 from “The percentage of the LZ-1105 or ICG in blood was calculated by subtracting the background from the blood of healthy mice.” into “The percentage of the LZ-1105 or ICG in blood was calculated as follow:

$$\%ID / g = \frac{F_t - F_{control}}{(F_{injected} - F_{control}) \times M_t} \times 100\%$$

Where F_t is the fluorescent intensity of collected blood as measured with a 1400-nm long-pass filter, $F_{control}$ is the fluorescent intensity of control blood, $F_{injected}$ is the fluorescent intensity of the injected LZ-1105 (mixed with blood), M_t is the mass of the collected blood” In our revised manuscript.

Reviewer #2 (Remarks to the Author):

Bioimaging in the NIR window using organic fluorophores is of significant importance in the field of cancer diagnosis and therapy. This study is especially attractive because NIR-II imaging has the potential to overcome the current limitation of reflectance-based imaging, i.e., spatial resolution and tissue penetration depth. By designing hydrophilic small-molecule NIR-II organic dyes from commercially available benzopyrrole derivatives could obtain a long-term blood circulation (3.2 h) dye LZ-1105 for the dynamic vascular imaging. This is the first small-molecule NIR-II probe that resulted in an impressive dynamic vascular process, incorporating the ischemic reperfusion in hindlimbs, thrombolysis in the carotid artery and opening and recovery of the blood-brain barrier. The manuscript is well organized, results are mostly supported by solid experimental data, and conceptually novel for the NIR-II organic probe design and bioapplications. Although the overall quality and presented data are solid. However, the data have several problems that need to be addressed:

1) Excitation with longer wavelengths may cause thermal effects, which could be a serious concern for their future clinical application. It would be helpful to discuss this issue.

Response: Thanks for the comments. We have added the thermal effect experiments in **Figure R2-1** (Supplementary Fig. 16 in the revised supporting information), which demonstrated that the 1064 nm NIR-II excitation has negligible thermal effect on the tissue under the power density (70 mW/cm^2) which we used for in vivo bioimaging. In order to make this point clearer for the readers, we have added the related description in the revised supplementary information in Page 13 as following “Meanwhile, the 1064 nm NIR-II excitation has negligible thermal effect on the tissue under the power density (70 mW/cm^2) used for in vivo bioimaging (Supplementary Fig. 16).”

Figure R2-1. Thermal effect experiments. Thermal effect experiments of 1064 nm laser (70 mW/cm^2) irradiation on balb/c nude mice back using FLIR A315 thermal imaging camera. a)

Photograph of the experimental setup. b) Photothermal images of mice back with different laser working power density for different irradiation time. The radiation area was shown with yellow circle. c) Thermal effect for mice skin during 30 min irradiation of laser as a function of time. Scale bar represents 3 mm.

2) Although clear images were obtained utilizing 1400 nm long-pass filter, the fluorescence intensity would be very low >1400 nm according to the emission spectra. The author should discuss the NIR-II bioimaging results using various long-pass filters (850-, 1,000-, 1,200-, 1,300-, and 1,400-nm).

Response: thanks for the comments. NIR bioimaging results using various long-pass filters were shown as **Figure R2-2-5** and Supplementary Fig.19-22 in the revised supplementary information. The SBR and resolving ability of NIR images using different long-pass filters increased obviously with the increased wavelength of the long-pass filters. Thus, 1400 nm long-pass filter was selected to obtain NIR-II bioimaging for LZ-1105.

Figure R2-2. NIR-II bioimaging of LZ-1105 injected mice brain with various long-pass filters. a) NIR bioimaging of balb/c nude mice brain with various long-pass filters by LZ-1105 administration under 1064 nm excitation. b) SBR of balb/c nude mice brain imaging with various long-pass filters. c) Contrast of brain vessels acquired by various long-pass filters. d) FWHM of Gaussian fitted fluorescence intensity profiles of brain vessels acquired by various long-pass filters.

Figure R2-3. FWHM of LZ-1105 injected mice brain obtained by various long-pass filters. The fluorescence intensity profiles (dots) and Gaussian fit (lines) along the red-dashed line in brain with various long-pass filters.

Figure R2-4. NIR-II bioimaging of LZ-1105 injected mice hindlimbs with various long-pass filters. a) NIR bioimaging of balb/c nude mice hindlimb with various long-pass filters by LZ-1105 administration under 1064 nm excitation. b) SBR of balb/c nude mice hindlimb imaging with various long-pass filters. c) Contrast of hindlimb vessels acquire by various long-pass filters. d) FWHM of Gaussian fitted fluorescence intensity profiles of hindlimb vessels acquire by various

long-pass filters.

Figure R2-5. FWHM of LZ-1105 injected mice brain obtained by various long-pass filters. The fluorescence intensity profiles (dots) and Gaussian fit (lines) along the red-dashed line in hindlimb with various long-pass filters.

To make it clear for the reviewer and the readers, we added the related description to expound the reason of choosing 1400 nm long-pass filters when acquire NIR-II bioimages using LZ-1105 in “*Synthesis and optical characterization of the LZ series dyes for in vivo vasculature imaging*” section in the revised manuscript as follow:

“Before administrating ICG and LZ-1105 for NIR-II imaging, signal collection was optimized in vitro and in vivo. 1300 nm and 1400 nm long-pass filters were finally selected to acquire optimal NIR-II imaging for ICG and LZ-1105, respectively (Supplementary Fig. 11-22).”

Besides, we added “Optimizing the signal collection for ICG and LZ-1105 NIR imaging” section in the revised supplementary information in Page 12.

3) The author had only focused on the photostability of LZ-1105 in blood, PBS and DI. The physicochemical and optical properties of new fluorophore need to be analyzed in serum-containing warm media. Biodistribution (% ID/g) and quantification need to be reconsidered with considering photo-decay (photo-ablation by laser), thickness and optical properties (scattering coefficient) of each organ, and other pharmacokinetic parameters.

Response: Thanks for the comments. The stability of LZ-1105 in serum and other media at 37 °C was measured for 180 min and the results were shown as **Figure R2-6** and Supplementary Fig.9 in

the revised supplementary information. The fluorescence intensity of LZ-1105 remained stable after incubated with serum at 37 °C for 180 min, illustrating the photostability of LZ-1105 in serum. To measure the biodistribution of LZ-1105, organs (heart, liver, spleen, lung, kidney, brain, skin, bone) were homogenized and measured under InGaAs charge-coupled device (CCD). (Figure R2-7 and Supplementary Fig. 34 in the revised supplementary information).

Figure R2-6. Photostability of LZ-1105 and ICG. LZ-1105 and ICG in a variety of biological media under continuous 1064 nm and 808 nm lasers excitation exposure for 180 min (330 mW/cm^2), respectively.

Figure 2-7. LZ-1105 distribution in main organs and tissues. a) Ex vivo images of main organs of LZ-1105 treated mice after 1 h, 6 h and 24 h p.i.. b-c) Signal intensity of organs and tissues measured by InGaAs CCD before (b) and after (c) homogenizing.

To make it clear to the reviewer and the readers, we revised “Moreover, superior photostability of LZ-1105 was observed by exposing ICG and LZ-1105 in water, PBS (pH = 7.4) and mice blood to the continuous laser irradiation for 3 h, respectively (Supplementary Fig.8).” into “Moreover, superior photostability of LZ-1105 was observed by exposing ICG and LZ-1105 in blood, PBS (pH = 7.4), deionized (DI) water, serum and urine to the continuous laser irradiation for 3 h, respectively (Supplementary Fig. 9).” on Page 5 in the revised manuscript.

Besides, we revised “Main organs including liver, heart, spleen, kidney, lung, stomach, intestine, skin, muscle, brain and bone were collected for NIR-II fluorescence imaging.” into “Besides, ex vivo images of main organs including the liver, heart, spleen, kidney, lung, stomach, intestine, skin, muscle, brain and bone of LZ-1105 injected mice at 1 to 24 h p.i. showed extremely low organ uptake (Supplementary Fig. 34).” on Page 8 in the revised manuscript.

4) The authors have shown that the bioimaging resolution and penetration depth with LZ-1105 was far superior to that of ICG for non-invasive imaging of mouse cerebral vasculatures, hindlimb vasculatures and lymphatic vasculatures. How about the penetration depth for these *in vivo* imaging experiment. Please provide supporting data for the relationship between the bioimaging resolution and penetration.

Response: Thanks for the comments. The penetration depth of mouse cerebral vasculatures, hindlimb vasculatures and lymphatic vasculatures were measured precisely by vernier caliper at the corresponding imaging position. Precise penetration depth of brain, lymph vessels and hindlimb vessels measured to be about 1.31 mm, 1.31 mm and 0.50 mm, respectively (**Figure R2-8** and Supplementary Fig. 25 in the revised supplementary information).

Figure R2-8. Penetration depth of brain, lymphatic and hindlimb vasculatures. A repeated measure of the precise penetration depth of brain, lymph vessels and hindlimb vessels by vernier caliper after sacrificing the imaging mice. The average thickness were 1.31 mm, 1.31 mm and 0.50 mm, respectively.

To make it clear for the readers, we added the related description in “*Synthesis and optical characterization of the LZ series dyes for in vivo vasculature imaging*” section on Page 6 in the revised manuscript as follow:

“The penetration depths were measured as 1.31, 1.31 and 0.50 mm for brain, hindlimb and lymphatic system (Supplementary Fig. 25).”

To discuss the relationship between bioimaging resolution and penetration, the FDA-approved dye, ICG, was chosen as the model contrast agent.

First, single ICG-filled (mixed with blood, [ICG] = 10 μ M) capillary immersed into 1% Intralipid was excited at 808 nm and the emission was collected by different long-pass filters at varied depths (**Figure R2-9** and Supplementary Fig. 11 in the revised supplementary information).

The imaging signal intensity with different long-pass filter at 0 mm penetration depth was almost the same by adjusting the laser's working power density. With increase of penetration depth, attenuation of signal intensity and blurring of capillary profiles were observed for all images, and only images acquired beyond 1300 nm resolved sharp edges of the capillary at a depth up to 5.5 mm. 1300 nm long-pass filter group (19.0) exhibited a 10.6-fold signal-to background ratio (SBR) than that of 850 nm long-pass filter group (1.8) under 1.0 mm depth. Besides, FWHM, which was hired to evaluate the resolving ability, was observed to increase more obviously by using 850 nm long-pass filter. These results illustrated that ICG imaging acquired beyond 1300 nm exhibited higher signal-to-background ratio, resolution and deeper penetration compared to that of 850 nm.

Figure R2-9. Deep penetration of ICG-filled single capillary in mimic tissue. a) Fluorescence images of capillaries filled with ICG in blood ($[ICG] = 10 \mu M$) immersed in 1% Intralipid with varying depth. Imaging signals were collected with various long-pass filters under 808 excitation. b-f) Profiles measured at same position in the capillary in various long-pass group at different

depth. g) Measured SBR of capillary images as a function of depth. h) Wavelength-dependent FWHM of cross-sectional profiles in capillary images as a function of depth. The bars represent mean \pm s.d. derived from the uncertainty in the Gaussian fitting of feature width.

To further investigate the resolving ability achieved by 1300 nm long-pass filter, two ICG-filled (mixed with blood, [ICG] = 10 μ M) capillaries were chosen to mimic adjacent blood vessels *in vivo*. The two capillaries immersed into 1% Intralipid were excited at 808 nm and the emission was collected by different long-pass filters at varied depths (**Figure R2-10** and Supplementary Fig. 12 in the revised supplementary information). Similar with the single capillary imaging, attenuation of image intensities and blurring of capillary profiles were observed for all images with increase of penetration depth. Contrast of images acquired by various long-pass filters under different depth was calculated. The resolving ability could also be evaluated by contrast. Contrast of various long-pass filters groups was comparable when penetration depth was 0 mm. However, contrast of 850 nm long-pass group dropped below 0.06 under 3.0 mm depth, and the two capillaries could not be resolved from each other. In contrary, contrast of 1300 nm long-pass group stay above 0.2 even under 6.5 mm depth, demonstrating that images acquired by 1300 nm long-pass filter exhibited higher resolution.

Figure R2-10. Deep penetration of ICG-filled double capillaries in mimic tissue. a) Fluorescence images of capillaries filled with ICG in blood ([ICG] = 10 μ M) immersed in 1% Intralipid with varying depth. Imaging signals were collected with various long-pass filters under 808 excitation. b) Wavelength-dependent contrast of capillary images as a function of depth. c-g) Profiles measured at same position in the capillary in various long-pass groups at different depth.

To make it clear for the reviewer and the readers, we added the related description in Discussion section on Page 14 in the revised manuscript as follow:

“Then, we try to discuss and get a better understand of the relationship between resolution and

penetration. Resolution is limited by the following three factors: tissue scattering caused wider signal diameter, tissue absorption of excitation/emission light caused low signal intensity and tissue scattering and autofluorescence caused high background noise. Firstly, although scattering exponent (w) only depends on the species of tissue, the deeper tissue penetration would increase the frequency of scattering and lead to the broader FWHM and higher background noise. Second, under deeper tissue penetration, absorption caused excitation/emission light attenuation decreased the signal intensity. Thus, under the same experimental parameter, resolution tends to attenuate penetration along with the increased penetration (as we have illustrated in the “Optimizing the signal collection for ICG and LZ-1105 NIR imaging” section in supplementary information). The key points to improve the resolution under deep penetration are to decrease the tissue scattering, absorption and autofluorescence. Fortunately, acquiring images beyond longer wavelength could achieve the above purpose, thus, within the acceptable excitation power density, contrast agent with longer excitation/emission wavelength and longer wavelength long-pass filter should be chosen during the imaging process.”

5) Up to now, several dyes have also been reported as NIR-II contrast agents with absorption and emission beyond 1000 nm for in-vivo imaging. The authors should add these papers in the references (J. Med. Chem. 2019, 62, 2049; Adv. Optical Mater. 2019, 1900229, Chem. Sci., 2019, 10, 1219, Adv. Healthcare Mater. 2018, 7, 1800589; Chem. Sci. 2017, 8, 3489), which may support the argument.

Response: Thanks for the comments. We have cited the related reference as Reference 10, 12, 13, 14 and 15 in the revised manuscript.

6) The manuscript also still requires revision with respect to the language used. A professional language editing service to assist the authors in correcting the spelling, grammar, word use, and punctuation throughout the manuscript. To help with this process the authors may also ask a native English speaker or equivalent.

Response: Thanks for the comments. Language editing has been improved as follows:

1. We revised “With the excitation and emission in the NIR-II window and long-term blood circulation, LZ-1105 holds a great promise for clinical transformation for assessing vasculature related diseases.” into “LZ-1105 provides a novel approach for researchers to assess vasculature related diseases due to the long excitation and emission wavelength and long-term blood circulation properties.” in *Abstract* section.
2. We revised “A healthy and functional vasculature is responsible for delivering nutrients to cells and protecting organs by tightly regulating tissue perfusion. However, a number of ischemic conditions (eg, cardiovascular ischemia, cerebrovascular ischemic stroke and peripheral arterial diseases) could lead to severe morbidity or mortality. Moreover, during the treatment of brain diseases, one big challenge is the efficient delivery of therapeutic agents across the blood-brain barrier (BBB) of cerebral vascular system. Diagnosing vascular dysfunction and the cerebral vascular permeability of the BBB requires more than taking a static measurement at a single time point, it requires the ability to noninvasively monitor the

functional dynamic processes in real-time with high spatial and temporal resolution.” into “A healthy and functional vasculature plays an important role of delivering nutrients to cells and protecting organs. Belated diagnose of vascular dysfunction may lead to severe morbidity or mortality. For instance, ischemia, injury induced vascular leakage and drug overdose caused damage could not be completely avoided in accidents and surgeries. Timely detection and treatment would decrease the risk of amputation, hemorrhage and other vascular related secondary damage. However, obtaining the vascular dynamic information requires more than taking a static measurement at a single time point, it also asks for the ability to noninvasively monitor the functional dynamic processes in real-time with high spatial and temporal resolution.” on Page 3 in our revised manuscript.

3. We revised “Magnetic resonance imaging (MRI), X-ray computed tomography (X-ray CT) or positron emission tomography (PET) can resolve features down to ~ 100 μm with deep penetration but are limited by long processing time and ionizing radiation. Vascular hemodynamics is usually detected by Doppler technology of micro-ultrasonography with high temporal resolution of up to 1000 Hz, but spatial resolution attenuates with increased depth of penetration.” into “Traditional techniques including magnetic resonance imaging (MRI), computed tomography (CT) and positron emission tomography (PET) hold an advantage of deep penetration depth with resolution of ~ 100 μm , while they are suffering from long processing time and ionizing radiation. Besides, although Doppler imaging used for vascular hemodynamics measurement exhibits high temporal resolution up to 1,000 Hz, it is limited by the attenuated spatial resolution under deep tissue penetration.” on Page 3 in our revised manuscript.
4. We revised “Owing to unknown long-term toxicity concerns, it is necessary to design organic small-molecular NIR-II fluorophores for in vivo imaging to facilitate clinical translation.” into “Owing to unknown long-term toxicity concerns, development of organic small-molecular NIR-II fluorophores for in vivo imaging is in urgent demand.” on Page 4 in our revised manuscript.
5. We revised “Herein, we report a biocompatible small molecule NIR-II probe LZ-1105 for long-term in vivo dynamic imaging of vascular structures with a home-made imaging system.” into “Herein, we report a small molecule LZ-1105 as NIR-II probe for long-term *in vivo* imaging of dynamic vascular structure changing with a home-made imaging system.” on Page 4 in our revised manuscript.
6. We revised “Key steps used in the synthesis process included alkylated reaction and Vilsmeier-Haack reaction” into “Alkylated reaction and Vilsmeier-Haack reaction played as key steps during the synthesis process.” on Page 5 in our revised manuscript.
7. We revised “They showed high molar extinction coefficients (ϵ_{max}) in the range of $1.01\text{-}1.99 \times 10^5 \text{ cm}^{-1}\text{mol}\cdot\text{L}^{-1}$, which were comparable to the clinically approved ICG” into “Besides, the high molar extinction coefficients (ϵ_{max} , $1.01\text{-}1.99 \times 10^5 \text{ cm}^{-1}\text{mol}\cdot\text{L}^{-1}$) were comparable to the clinically approved ICG, which ensured the signal brightness during NIR-II imaging.” on Page 5 in our revised manuscript.
8. We revised “The whole body NIR-II imaging, the smallest measurable vessel exhibited a Gaussian-fit diameter of only 36 μm (Supplementary Fig.9). Especially, six lymph vessels with the FWHM from 90 to 161 μm were visualized clearly after LZ-1105 injection, compared to the broad signal distribution measured with ICG (Fig. 1k).” into “As an

- evidence to illustrate the high resolution achieved by LZ-1105, 36 μm -wide tiny blood vessel was observed (Supplementary Fig. 27). Especially, six lymph vessels with the diameters from 90 to 161 μm were visualized clearly after LZ-1105 injection, which was far superior to the broad signal distribution measured with ICG (Fig. 1k).” on Page 7 in our revised manuscript.
9. We revised “Real-time observing of ischemic reperfusion after ectopic replantation and functional reconstruction is of critical importance in clinical practice.” into “After ectopic replantation and functional reconstruction, real-time observing of ischemic reperfusion is of critical importance to evaluate the degree of recovery.” on Page 10 in our revised manuscript.
 10. We revised “Besides, real-time bioimaging revealed a remarkable delay of fluorescence signals in ischemic hindlimbs with longer clipping time.” into “By calculating the recovery time and blood flow velocity (BFV), real-time fluorescence bioimaging revealed a remarkable delay of signals in ischemic hindlimbs with longer clipping time.” on Page 10 in our revised manuscript.
 11. We revised “Although the obvious real-time bioimaging advantages compared to other imaging techniques, this non-invasive bioimaging method can only be used for the preclinical evaluation of vascular dynamics related diseases at present. In the future work, improvements of the imaging equipment, such as larger range of detectors with higher sensitivity or endoscope techniques may permit the use of this technique in larger animal models or human patient. Promisingly, the generality of the LZ-1105 dye suggests applicability to a range of vascular physiological processes, which is likely to prove advantageous in the future.” into “Although real-time NIR-II optical bioimaging holds obvious advantages compared to other imaging techniques, this non-invasive imaging method can only be used for superficial vessels or on small animals to monitor the vascular dynamics related diseases. In the future work, improvements of the imaging equipment, such as larger range of detectors with higher sensitivity or endoscope techniques may permit the use of this technique in larger animal models. As least, LZ-1105 based optical imaging provides a novel method for vascular dynamic monitoring, which could be hired to combine with other traditional imaging techniques for improved imaging.” on Page 17 in our revised manuscript.
 12. We revised “This method may be broadly applicable for streamlining preclinical drug development processes and enabling new research into the pathogenesis of vasculature related diseases.” into “This method may be hired to dynamically discover and monitor more vascular related phenomenon and enable new research into the pathogenesis of vasculature related diseases.” on Page 18 in our revised manuscript.

7) In Figure 1h-k, signal to background ratio (SBR) of hindlimb vessels and lymphatic images were not given.

Response: Thanks for the comments. SBR of hindlimb vessels and lymphatic images were given in **Figure R2-11-R2-12**, Supplementary Fig.26 in the revised supplementary information and Figure 1i in the revised manuscript.

Figure R2-11. NIR-II bioimaging of lymphatic system in ICG and LZ-1105 injected mice. a) Non-invasive NIR-II fluorescence images of lymphatic system in shaved ICR mice i.v. injected with LZ-1105 (1400 long-pass filter, $\lambda_{ex}=1064$ nm, 300 ms) or ICG (1300 long-pass filter, $\lambda_{ex}=808$ nm, 300 ms). b) The fluorescence intensity profiles (dots) and Gaussian fit (lines) along the red-dashed line in lymphatic system. Scale bar represents 4 mm.

Figure R2-12. NIR-II bioimaging of hindlimb in ICG and LZ-1105 injected mice. a) Non-invasive NIR-II fluorescence images of hindlimb in shaved ICR mice i.v. injected with LZ-1105 (1400 long-pass filter, $\lambda_{ex}=1064$ nm, 300 ms) or ICG (1300 long-pass filter, $\lambda_{ex}=808$ nm, 300 ms). b) The fluorescence intensity profiles (dots) and Gaussian fit (lines) along the red-dashed line in hindlimb. Scale bar represents 4 mm.

To make it clear for the readers, the related description was added in “*Synthesis and optical characterization of the LZ series dyes for in vivo vasculature imaging*” section in the revised supplementary information as follows:

“Besides, blood vessels and lymphatic system of the hindlimb were also observed by NIR-II imaging with high SBR and resolution (Fig. 1h-k and Supplementary Fig. 26).”

8) In Figure 2e, the tiny blood vessels were not clear at 2 h after injection. However, they are very clear in Figure 5b. The authors should give a reasonable explanation.

Response: Thanks for the comments. In Figure 2e, bioimaging results of brain vessels were processed under the same fluorescence intensity range (0-10000). However, in Fig. 5b, the fluorescence intensity region (noted as the yellow circle in the **Figure R2-13** and Fig. 5b in the revised manuscript) of each images was normalized to measure the signal intensity decreasing process of the leakage on the brain vessels.

Figure R2-13 (Fig. 5 in the revised manuscript). Real-time non-invasive NIR-II imaging to monitor the opening and recovery of the blood-brain barrier (BBB). **a** Schematic illustration of the NIR-II imaging setup to monitor the opening and recovery of the BBB. **b** The NIR-II images of

the mice brain before and after treated with focused ultrasound (FUS) and microbubbles at different time points (n = 3 mice). The green and red circles and arrows indicate the opening and recovering points of cerebral vessels. **c, d** Corresponding normalized mean signal intensity of green (c) and red (d) circles as a function of time. **Note: The fluorescence intensity region (noted as the yellow circle) of each images was normalized to measure the signal intensity decreasing process of the leakage on the brain vessels. Scale bars in Fig 5b represents 4 mm.**

To make it clear for the reviewer and the readers, the related description was added in *Methods* section in the revised manuscript as follows:

“All images performed by ICG or LZ-1105 were processed under the same fluorescence intensity range (0-10000).”

“The fluorescence intensity region (noted as the yellow circle in the Fig. 5b in the revised manuscript) of each images was normalized to measure the signal intensity decreasing process of the leakage on the brain vessels.”

Meanwhile, the related description was added in fig.5 caption in the revised manuscript as follows:

Note: The fluorescence intensity region (noted as the yellow circle) of each images was normalized to measure the signal intensity decreasing process of the leakage on the brain vessels.

9) The scale bars should be provided for all figures and the key parameters such as filters, exposure time should be provided in the text and SI.

Response: Thanks for the comments. Scale bars were provided in Figure 1f-j, Figure 2a, b and c, Figure 3b, Figure 4b, Figure 5b and Supplementary Fig. 10, 14, 16, 17, 19, 21, 25-30, 39-46 in the revised manuscript and supplementary information. **In addition, filters, exposure time were provided in *Methods* section in the revised manuscript and the revised supplementary information.**

10) In Figure S10, the spatial resolution between the bright field image and the NIR-II image with the clinical imaging method should be provided.

Response: Thanks for the comments. Ultrasonic (US) imaging was chosen as the clinical imaging method to compare with our NIR-II bioimaging performed by LZ-1105 (**Figure R2-14** and Supplementary Fig. 28 in the revised supplementary information). US imaging failed to provide a complete and clear blood vessel image in the hindlimb while NIR-II bioimage exhibited precise vessel images which was comparable with the bright field (without skin).

Figure R2-14. The digital photograph, Ultrasound imaging and NIR-II imaging of hindlimb in LZ-1105 injected mice. a. The digital camera photograph of left leg in mouse with or without skin, Ultrasound imaging and NIR-II fluorescence imaging ($\lambda_{\text{ex}}=1064$ nm, 1400 nm long-pass filter, 300 ms) of the same area in hindlimb blood vessels. b. The fluorescence intensity profiles (dots) and Gaussian fit (lines) along the yellow-dashed line in NIR-II imaging. Scale bar represents 3 mm.

To make it clear for the readers, the related description was added in “*Synthesis and optical characterization of the LZ series dyes for in vivo vasculature imaging*” section in the revised manuscript as follows:

Furthermore, the widths of the hindlimb blood vessels at the same position were measured as 130 μm with NIR-II imaging and 126 μm with white light optical photograph, whereas ultrasound imaging could not discern any vessel, demonstrating the precise imaging ability of LZ-1105 in NIR-II window (Supplementary Fig. 28).”

11) In Figure 6, the key parameters of these programs should be provided.

Response: Thanks for the comments. Parameters of the programs were provided in *Methods* section in the revised manuscript as follow:

“Density functional theory calculations for LZ dyes.

Density function theory (DFT) calculations were conducted with the Gaussian 03 program using the B3LYP method and 6-31G* basis set. The geometries were optimized using the default convergence criteria without any constraints.”

12) Probably it’s better to use velocity measurement with the highest spatial resolution that could be the real advantages for in vivo kinetic study, especially to understand alpha phase half-life.

Response: Thanks for the comments. We apologize for the unclear expression of the kinetic study for LZ-1105. Fitted by Pharmacokinetic simulation software Drug and Statistics (DAS) 2.0, ICG performed two-compartment model, $t_{1/2\alpha} = 3.1$ min, $t_{1/2\beta} = 502.9$ min. While LZ-1105 performed one-compartment model, $t_{1/2} = 195.4$ min. Moreover, blood flow velocity was measured with the highest spatial resolution by ICG and LZ-1105 administration (Figure R2-15). The blood flow velocity measured by LZ-1105 and ICG were comparable, illustrating the signal front obtained from NIR-II images could present for the real blood front and our half-life measurement by measuring the signal intensity of blood at different time point was reliable.

Figure R2-15. a) Time course NIR-II bioimages showing the flow front (marked by yellow arrows) in the hindlimb by ICG and LZ-1105 administration. b) Distance traveled by the flow front as a function of time by ICG and LZ-1105 administration. The slope of the function is calculated as the blood flow velocity.

To make it clear for the readers, the related description was added in *Methods* section in the revised manuscript as follows:

“The kinetic study for LZ-1105 and ICG was fitted by Pharmacokinetic simulation software Drug and Statistics (DAS) 2.0.”

Reviewers' comments:

Reviewer #1 (Remarks to the Author):

Summary: This is a revised contribution in which the authors profusely responded to reviewer comments in lengthy rebuttal. The authors failed to address the point that the differences in performance contrast with ICG and LZ-1105 were due to differences in collection as the response even more strongly suggests. As such the claim that ICG is better than LZ-1105 in terms of resolution and depth still remain unsubstantiated. A fluorescent photon whether generated by ICG or LZ-1105 will behave identically in scattering media. If you look at DIFFERENT fluorescent photons (i.e. with different LP filters), then of course there will be differences in performance. Hence the rebuttal further supports that this contribution does not provide substantiated conclusions. The following highlights continuing major and minor points.

MAJOR POINTS (with reference to past major comments):

1. The authors continue to disregard that the limited penetration depth of SWIR fluorescence is unlikely to led to clinical applications. Despite stating as much at the later stages of the manuscript, the introduction still motivates the current study with an introduction of clinical imaging needs not met by clinical CT, PET, MRI, etc. For SWIR (or NIR-II), there is significant limitation in clinical applications and the introduction and the use of “deep” is misleading to readers. The response to my major point 1 was primarily a treatise of past studies, some of which also promotes SWIR in the same incorrect context of applications as PET, MRI, etc.

This begs the question of how much innovation (in light of the several other studies cited in response to my point #1) this contribution has for showing improved resolution at shallow depths (1.3 mm) from SWIR over that from NIRF. Clearly SWIR is limited in penetration depth over NIRF, but as shown by others, has greater resolution at shallow depths that are not clinically relevant for vascular imaging.

2. The statement that some works report increased (emission) wavelengths has deeper penetration should be balanced by other works showing that is as more shallow penetration in the SWIR as opposed to NIRF region. Since this manuscript does not compare NIR and SWIR, these statements need to be struck from the manuscript as they are misleading to the readership.

In response to my comment, a figure is provided (R1-8) stating contrast is measured. While contrast is not provided in the figure, it is noteworthy that its calculation as the standard deviation of pixel value divided by the mean is incorrect. This manuscript should not be published with the well known imaging performance metric of contrast defined in this erroneous manner.

3. No further comment on authors' response to prior major point 3.

4. In response to my comment asking what mechanism LZ-1105 was retained in the vasculature if it was "free", the authors have stated that LZ-1105 is not "free" after all. Without convincing evidence, they now state that LZ-1105 associates with fibrinogen in blood. Calorimetry studies are done attesting to "binding" to fibrinogen, but there could be other and multiple components in blood in which associated with LZ-1105. In addition calorimetry studies are done with blood and LZ-1105 and its unclear such an experiment could be done with the complexity of blood. IS it possible that LZ-1105 associated with a cellular compartment of blood (but not with blood endothelial cells?). Fibrinogen is generated in the liver and therefore, I would have expected the liver to be VERY bright is fibrinogen "binding" increasing the quantum yield. Yet the liver is dim. In addition, how exactly is fibrinogen increasing the quantum yield so much?

Figure R1.10 is suspicious since it shows kcal/mol of injection (Sic) for fibrinogen-LZ-1105 and fibrinogen-ICG based upon the number of dyes per BSA molecule???. How is this ITC results for binding of LZ-1105 or ICG with fibrinogen ?

Figure R1.11 Please review the caption as it does not correspond accurately with the figures.

As described by reviewer 2, the innovation of this contribution is LZ-1105 and hence the characterization and understanding of LZ-1105 this is an important component of the manuscript.

5. The authors state that blood reduces the stacking of LZ-1105 that causes static quenching (as it does with ICG, a planar molecule). I must have missed this, but is LZ-1105 a planar molecule and do you have analytical measurements showing that it is?

6. The Quantum efficiency of the NIRvana system is given in https://www.princetoninstruments.com/userfiles/files/assetLibrary/Datasheets/NIRvana_640_rev_P1-7-7-16.pdf. At 850 nm, the quantum efficiency of the camera falls off and therefore the results at 850 nm cannot be compared to longer wavelengths.

BTW: What does the scale bar of Figure R1-13a correspond to?

7. The authors changed blood flow rate to velocity, as requested, but at the end of the manuscript, reverted back to blood flow. The method does not measure blood flow, but rather velocity of a front and therefore can only provide a single time point (at time of injection) of blood velocity.

8. See response to comment 6.

9. No further comment.

10. Stability of an agent would typically be tested with HPLC. However, the photostability may suffice here.

11. No further comment.

12. See response to comment 10.

13, 14 No further comment

15. See comment 6

Other comments:

1. Figure R2-1 Its hard to imagine that 30 minutes of 70 mW/cm² of 1064 nm light causes no heating of tissues. Please check this.

2. The authors write that the NIRvana camera is a CCD. It is not. It is a focal plane array without the read out architecture of a CCD.

Reviewer #2 (Remarks to the Author):

Comments were well addressed and the revised version is suitable for publishing in Nature Communications.

Reviewers' comments:

Reviewer #1 (Remarks to the Author):

Summary: This is a revised contribution in which the authors profusely responded to reviewer comments in lengthy rebuttal. The authors failed to address the point that the differences in performance contrast with ICG and LZ-1105 were due to differences in collection as the response even more strongly suggests. As such the claim that ICG is better than LZ-1105 in terms of resolution and depth still remain unsubstantiated. A fluorescent photon whether generated by ICG or LZ-1105 will behave identically in scattering media. If you look at DIFFERENT fluorescent photons (i.e. with different LP filters), then of course there will be differences in performance. Hence the rebuttal further supports that this contribution does not provide substantiated conclusions. The following highlights continuing major and minor points.

Response: Thanks for the comments. The collection regions for ICG and LZ-1105 in our work were different. However, the appropriate excitations and the maximum emission of the two dyes were also different. So we chose the **optimal** collection region for ICG and LZ-1105, respectively. Actually, as for ICG, it has been reported in the literature that the higher SBR and resolution was obtained with 1300 nm long-pass filters in vivo (refer to: Carr, J.A. et al. Shortwave infrared fluorescence imaging with the clinically approved near-infrared dye indocyanine green. Proc. Natl. Acad. Sci. USA 115, 4465-4470 (2018)). We also demonstrated the signal collection for ICG and LZ-1105 via series of in vitro and in vivo experiments to emphasize this point in previous response to reviewer (comment 6, figure R13-21). **Fluorescence imaging needs to be acquired with high imaging quality (high resolution, signal-to-noise ratio)**, and we think the reviewer ignore this point.

MAJOR POINTS (with reference to past major comments):

1. The authors continue to disregard that the limited penetration depth of SWIR fluorescence is unlikely to lead to clinical applications. Despite stating as much at the later stages of the manuscript, the introduction still motivates the current study with an introduction of clinical imaging needs not met by clinical CT, PET, MRI, etc. For SWIR (or NIR-II), there is significant limitation in clinical applications and the introduction and the use of “deep” is misleading to readers. The response to my major point 1 was primarily a treatise of past studies, some of which also promotes SWIR in the same incorrect context of applications as PET, MRI, etc.

This begs the question of how much innovation (in light of the several other studies cited in response to my point #1) this contribution has for showing improved resolution at shallow depths (1.3 mm) from SWIR over that from NIRF. Clearly SWIR is limited in penetration depth over NIRF, but as shown by others, has greater resolution at shallow depths that are not clinically relevant for vascular imaging.

Response: Thanks for the comments. We agree with the reviewer that the fluorescence bioimaging has shallow tissue penetration compared to the clinical CT, PET, MRI, etc, especially for the preoperative detection. However, due to the advantages of fast-feedback and non-ionizing

radiation, actually the fluorescence bioimaging technology has been used for the imaging guided surgery (Cited from: Gotoh, K., *et al.* A novel image-guided surgery of hepatocellular carcinoma by indocyanine green fluorescence imaging navigation. *J. Surg. Oncol.* **100**, 75-79 (2009); Schaafsma, B.E., *et al.* The clinical use of indocyanine green as a near-infrared fluorescent contrast agent for image-guided oncologic surgery. *J. Surg. Oncol.* **104**, 323-332 (2011)). Furthermore, optical imaging in SWIR region provides a method to continuously monitor physical process of small animals. This technology can be used as a method for fundamental research. Thus, we have revised the introduction section as follows:

“A healthy and functional vasculature plays an important role of delivering nutrients to cells and protecting organs. Timely detection and treatment of vascular dysfunction would decrease the risk of amputation, hemorrhage and other vascular related secondary damage. However, obtaining the vascular dynamic information requires more than taking a static measurement at a single time point, it also asks for the ability to noninvasively monitor the functional dynamic processes in real-time with high spatial and temporal resolution. Optical bioimaging exhibits advantages of fast-feedback and non-ionizing radiation. Although the optical imaging penetration depth is limited to several centimeters, continuously monitoring dynamic physical process of small animals would be helpful to broaden the understanding of vascular dysfunction and recovery process. Compared to the traditional near-infrared window (780-900 nm), recent developed second near-infrared window (NIR-II; 1000-1700 nm) has received considerable attention for noninvasive in vivo imaging because of higher resolution and signal-to-background ratio (SBR). Unfortunately, widely used inorganic NIR-II contrast agents such as single-walled carbon nanotubes (SWNTs), quantum dots (QDs), and rare-earth doped downconversion nanoparticles (DCNPs) hold unknown long-term toxicity concerns. Thus development of organic small-molecular NIR-II fluorophores for in vivo imaging is in urgent demand. Several organic NIR-II contrast agents have been designed for in vivo bioimaging. However, due to the short blood circulation half-life time (5-60 min) (Supplementary Table.1), continuously monitoring of dynamic physical process would be impeded.

Herein, we report a small molecule LZ-1105 as NIR-II probe for long-term in vivo imaging of dynamic vascular structure changing in small animals. LZ-1105 exhibits a high aqueous solubility, a peak fluorescent emission at 1,105 nm and a molecular mass of 0.15 kDa. Meanwhile, LZ-1105 shows high resolution and SBR for non-invasive imaging of mouse cerebral vasculatures, hindlimb vasculatures and lymphatic vasculatures. Significantly, pharmacokinetics of LZ-1105 demonstrated a half-life of over 3 h in blood circulation, which provides a long imaging window for long-term monitor the dynamic vascular structure changing. LZ-1105 is capable of real-time quantifying femoral artery blood velocity in ischemic perfusion hindlimbs, thrombolysis process of carotid arterial thrombosis and the temporary opening and recovery of the blood-brain barrier (BBB) in living mice. Thus, NIR-II imaging modality with the long-term angiography probe in small animal model might help researchers to understand some dynamic vascular changing process.”

2. The statement that some works report increased (emission) wavelengths has deeper penetration should be balanced by other works showing that is as more shallow penetration in the SWIR as opposed to NIRF region. Since this manuscript does not compare NIR and SWIR, these statements need to be struck from the manuscript as they are misleading to the readership.

In response to my comment, a figure is provided (R1-8) stating contrast is measured. While contrast is not provided in the figure, it is noteworthy that its calculation as the standard deviation of pixel value divided by the mean is incorrect. This manuscript should not be published with the well known imaging performance metric of contrast defined in this erroneous manner.

Response: Thanks for the comments. We revised our introduction and discussion section substantially. We have highlighted the revised abstract, introduction and discussion section in the revised manuscript.

In the abstract section, our revision is listed as follows:

1. We have revised “Compared to the current methods, optical bioimaging in the second near-infrared (NIR-II) window provides advantages including deep penetration depth, high resolution and fast feedback.” into “Optical bioimaging in the second near-infrared (NIR-II) window provides advantages including high resolution and fast feedback.” On Page 2 in revised manuscript

In the introduction section, our revisions are listed as follows:

2. We have revised “Recent development in noninvasive fluorescence in vivo imaging in the second near-infrared window (NIR-II; 1000-1700 nm) has received considerable attention because of deeper tissue penetration depth compared to traditional near-infrared window (780-900 nm) imaging techniques.” into “Compared to the traditional near-infrared window (780-900 nm), recent developed second near-infrared window (NIR-II; 1000-1700 nm) has received considerable attention for noninvasive in vivo imaging because of higher resolution and signal-to-background ratio (SBR)” on Page 3 in revised manuscript.
3. We have revised “The bioimaging resolution and penetration depth with LZ-1105 was far superior to that of ICG for non-invasive imaging of mouse cerebral vasculatures, hindlimb vasculatures and lymphatic vasculatures.” into “Meanwhile, LZ-1105 shows high resolution and SBR for non-invasive imaging of mouse cerebral vasculatures, hindlimb vasculatures and lymphatic vasculatures.” on page 4 in revised manuscript.

In the discussion section, our revision is listed as follows:

4. We have revised “Current methodologies for in vivo assessing vasculature and hemodynamics in microvasculature are suboptimal because no single method provides enough information and essential parameters, including vascular structure and tissue perfusion, with high spatial and temporal resolution. NIR-II imaging technique simultaneously provides anatomical and hemodynamic information owing to reduced tissue scattering, deeper anatomical penetration and non-radiative effect.” into “NIR-II imaging technique simultaneously provides anatomical and hemodynamic information owing to reduced tissue scattering and non-radiative effect.” on page 13 in revised manuscript.

For figure R1-8 in original response to reviewer, we describe the test method of contrast based on previous reports (which are listed below). The specific values are marked on the corresponding images or histograms, such as Fig. 1 (f, h, g), Supplementary Fig.12 (b), Supplementary Fig.13 (b), Supplementary Fig.14 (c), Supplementary Fig.17 (c), Supplementary Fig.19 (c) and Supplementary Fig.21 (c) in our previous revised manuscript and supplementary information.

Meanwhile, we cannot agree with the reviewer’s comment about the definition of contrast.

Actually, there has been some works which describe the definition of contrast.

As James B. Pawley reported, contrast can be defined in many ways but usually it involves a measure of the variation of image signal intensity divided by its average value.

Contrast can be defined in many ways but usually it involves a measure of the variation of image signal intensity divided by its average value:

$$C = \frac{\Delta I}{I}$$

Contrast is just as essential to the production of an image as "resolution." Indeed, the two concepts can only be thought of in terms of each other. They are linked by a concept called the contrast transfer function (CTF), an example of which is shown in Figure 4.3.

(Cited from: Pawley, J.B. *Handbook of Biological Confocal Microscopy* Ch. 4 Points, Pixels, and Gray Levels: Digitizing Image Data (Springer, New York, 2006))

camera. We chose the 1,300- to 1,620-nm wavelength range, since as we show here and others have previously shown, contrast and resolution are maximized at wavelengths greater than 1,300 nm (13, 19, 52). We quantified the contrast within a region of interest in the NIR image and the SWIR image by calculating the coefficient of variation, defined as the standard deviation (SD) of pixel intensity normalized to the mean pixel intensity

(Refer from: Carr, J.A. *et al.* Shortwave infrared fluorescence imaging with the clinically approved near-infrared dye indocyanine green. *Proc. Natl. Acad. Sci. USA* **115**, 4465-4470 (2018))

3. No further comment on authors' response to prior major point 3.

Response: Thanks for the positive comments.

4. In response to my comment asking what mechanism LZ-1105 was retained in the vasculature if it was "free", the authors have stated that LZ-1105 is not "free" afterall. Without convincing evidence, they now state that LZ-1105 associates with fibrinogen in blood. Calorimetry studies are done attesting to "binding" to fibrinogen, but there could be other and multiple components in blood in which associated with LZ-1105. In addition calorimetry studies are done with blood and LZ-1105 and its unclear such an experiment could be done with the complexity of blood. IS it possible that LZ-1105 associated with a cellular compartment of blood (but not with blood endothelial cells?). Fibrinogen is generated in the liver and therefore, I would have expected the liver to be VERY bright is fibrinogen "binding" increasing the quantum yield. Yet the liver is dim. In addition, how exactly is fibrinogen increasing the quantum yield so much?

Figure R1.10 is suspicious since it shows kcal/mol of injection (Sic) for fibrinogen-LZ-1105 and fibrinogen-ICG based upon the number of dyes per BSA molecule???. How is this ITC results for binding of LZ-1105 or ICG with fibrinogen?

Figure R1.11 Please review the caption as it does not correspond accurately with the figures.

As described by reviewer 2, the innovation of this contribution is LZ-1105 and hence the characterization and understanding of LZ-1105 this is an important component of the manuscript.

Response: Thanks for the comments. In our previous revised manuscript, we have demonstrated the interaction between LZ-1105 and fibrinogen by fluorescence increase and ITC experiments. To further demonstrate the interaction between LZ-1105 and fibrinogen, we hydrolyzed the fibrinogen in blood by adding fibrinolytic enzyme (plasmin) into the fresh blood. Then LZ-1105 was added into the treated blood sample. Almost 4-fold decrease in brightness of the mixing solution could be observed compared to that of the blood sample without plasmin. Furthermore, we incubated LZ-1105 with human embryonic renal epithelial cells (HEK293T) and extremely low NIR-II signal could be observed (Supplementary Fig. 38 in revised Supplementary information). These results illustrated that the fluorescence increase of LZ-1105 was mainly due to the binding with fibrinogen.

To make it clear to the reviewer and the readers, we added the related description in “*In vivo pharmacokinetics and blood retention*” section on Page 9 in the revised manuscript as follow: “Meanwhile, we hydrolyzed the fibrinogen in blood sample by plasmin, after adding LZ-1105 into the treated blood sample, fluorescent signal decreased about 4-fold compared to that of the untreated blood. Besides, LZ-1105 was further demonstrated to have little interaction with human embryonic renal epithelial cells (HEK293T) (Supplementary Fig. 38).”

Supplementary Fig. 38. Optical characterization of LZ-1105 in different media. The fluorescence intensity of LZ-1105 was detected by the InGaAs camera in different media including fresh mice blood with or without plasmin, saline (0.9% NaCl solution), human embryonic renal epithelial cells (HEK293T) suspension and plasmin solution (1064 nm excitation, 30 mW/cm², 1400 nm long-pass filter). Insets: NIR-II fluorescence images of LZ-1105 in different media. ([LZ-1105] = 10 μM)

As for ITC experiment between LZ-1105 and blood, since nobody has done ITC experiment of whole blood, in order to make our manuscript rigorous, we deleted the ITC data of LZ-1105 and blood.

Through fluorescence changes and ITC experiments, we demonstrated the interaction between LZ-1105 and fibrinogen. Unfortunately, up to now, the metabolism mechanism of the LZ-1105 dye is still not clear, which still need more specific research to explain the phenomenon such as why the liver was dim. Since the innovation of the article is mainly focused on the novel NIR-II

molecular dye design with simple synthesis and excitation/emission in SWIR region for bioimaging on small animals, we will further research the metabolism mechanism of the LZ-1105 dye in the future work. In this work, we have exhibited the high imaging SBR and resolution by administration of LZ-1105. These results may allow researchers to monitor the small animals and would be helpful to broaden the understanding of vascular dysfunction and recovery process.

Interactions between organic dyes and proteins is a route to brilliantly enhance fluorescent dyes emission, such as the previously reported 4-fold increase in brightness of ICG when exposed to serum proteins and the 110-fold fluorescence increase of CH-4T when exposed to FBS proteins (Cite from: Benson, R. C. & Kues, H. A. Fluorescence properties of indocyanine green as related to angiography. *Phys. Med. Biol.* 23, 159–163 (1978); Antaris, A.L. *et al.* A high quantum yield molecule-protein complex fluorophore for near-infrared II imaging. *Nat. Commun.* 8, 15269 (2017)). The quantum yield of LZ-1105 in PBS is 0.03% and can be increased to 1.69% after mixing with blood (which was mainly due to the combining with fibrinogen). There may be two mainly reasons to explain the increased quantum yield. First, the conjugated aromatic structure of LZ-1105 would lead to aggregation in aqueous solution at high concentration through van der Waals forces. The aggregation would reduce the brightness through self-quenching. After binding with fibrinogen, aggregation will be hindered, leading to the increased quantum yield. Second, when binding with fibrinogen, the torsion of the molecule will be restricted, thus the quantum yield will be increased due to the rigidity caused minimized non-radiative transition.

Besides, we apologize for the writing error in Figure R1-10 and R1-11. The abscissa in Figure R1-10 should be “Dyes/fibrinogen (molar ratio)”, we had revised this error and reorganized caption of Fig. 2 in the revised manuscript and Supplementary Fig.39 in the revised supplementary information.

The basic principle of ITC is to measure the heat released or absorbed in a liquid sample after the addition of another liquid sample. The heat changing only occurs when there is interaction between organic molecule and protein. In our previous revised manuscript, heat changing only occurred after the addition of LZ-1105 solution into fibrinogen solution, while no heat changing was observed after the addition of ICG solution, illustrating the binding between LZ-1105 and fibrinogen.

5. The authors state that blood reduces the stacking of LZ-1105 that causes static quenching (as it does with ICG, a planar molecule). I must have missed this, but is LZ-1105 a planar molecule and do you have analytical measurements showing that it is?

Response: Thanks for the comments. Actually, ICG is not a complete planar molecule. ICG has four methyl groups. And there is a certain angle between the side chain containing sulfate groups and the polycyclic parts. (Cited from: Desmettre, T., Devoisselle, J. & Mordon, S. Fluorescence properties and metabolic features of indocyanine green (ICG) as related to angiography. *Survey of ophthalmology* 45, 15-27 (2000).) Instead, the molecular skeleton (aromatic conjugated methane chain structure) is planar. Thus the fluorescence intensity of ICG in water was quenched due to the π - π stacking. Similarly, the molecular skeleton of LZ-1105 (conjugated methane chain) is also planar, but due to the steric hindrance of terminal groups, the planarity of LZ-1105 is weaker than

that of ICG. To make it clear to the reviewer and the readers, we have revised “Meanwhile, similar with ICG, mixing with blood reduced the quenching effect by the stacking of planar LZ-1105 structure.” into “Meanwhile, after mixing LZ-1105 with blood, quenching effect is reduced (Supplementary Fig. 8).”

6. The Quantum efficiency of the NIRvana system is given in https://www.princetoninstruments.com/userfiles/files/assetLibrary/Datasheets/NIRvana_640_rev_P1-7-7-16.pdf. At 850 nm, the quantum efficiency of the camera falls off and therefore the results at 850 nm cannot be compared to longer wavelengths.

BTW: What does the scale bar of Figure R1-13a correspond to?

Response: Thanks for the comments. The emission spectrum of ICG was acquired by fluorescence spectrometer (Figure R1, red line). Meanwhile, by multiplying quantum efficiency and the fluorescence intensity, we can obtain the calibration emission spectrum of ICG (Figure R1, green line). Although the quantum efficiency of the camera falls off at 850 nm, the peak of calibrated ICG spectra appears at 873 nm, which is suitable for imaging measurements with 850 nm long-pass filter. Actually, the imaging intensity of ICG acquired by InGaAs camera with 850 nm long-pass filter is much higher than that of 1000 nm-, 1100 nm-, 1200nm-, 1300 nm long-pass filters (Figure R2). These results indicated that although the detector at 850 nm falls down, the amount of fluorescent photons collected beyond 850 nm is much higher than that of other long-pass filters groups. **However, the evaluation of optical imaging quality requires not only the amount of photons received by the camera, but also higher resolution and SBR.** In fact, in 850 nm long-pass filter group, background noise locates in the same collected region, which also falls off due to the low quantum efficiency. Thus the quantum efficiency of NIRvana system has little influence on the evaluation of imaging quality. **The low resolution and SBR acquired beyond 850 nm was due to the high tissue scattering, not fall-off quantum efficiency.**

Please note: The NIRvana system detector has a saturation value. Thus for 850 nm long-pass filter group, the power density of the excitation light cannot be arbitrarily increased. However, for the longer wavelength groups, due to the low fluorescence signal intensity, the power density of the excitation can be adjusted to increase the signal-to-noise ratio (the power density of the excitation should be within an acceptable range). Meanwhile, background noise would remain low because of decreased scattering signal in longer wavelength. Please see Figure R1-16, R1-18 and R1-19 in previous response. Therefore, superior fluorescence imaging quality (high resolution and SBR) of ICG could be achieved with 1300 nm long-pass filter at an acceptable power density of the excitation light (70 mW/cm²).

Figure R1. Quantum efficiency of the NIRvana system (black line), the emission spectrum of ICG acquired by fluorescence spectrometer (red line), and the ICG spectrum after calibration (green line).

Figure R2. Signal intensity of ICG-blood complex acquired by various long-pass filters.

In addition, there were not “scale bars” in Figure R1-13a in original response. The bright lines were used to represent capillaries with ICG-blood complex. The experimental setup is shown in Supplementary Figure 11 in the revised supplementary information. To make it clear for the reviewer and readers, we added the experimental setup illustration as Supplementary Figure 11 in the revised supplementary information.

Supplementary Figure 11. The experimental setup illustration for ICG-blood complex imaging under various penetration depth through Intralipid.

7. The authors changed blood flow rate to velocity, as requested, but at the end of the manuscript, reverted back to blood flow. The method does not measure blood flow, but rather velocity of a front and therefore can only provide a single time point (at time of injection) of blood velocity.

Response: Thanks for the comments. We have revised “blood flow” into “blood velocity” on Page 4, 16 in the revised manuscript.

8. See response to comment 6.

Response: Please see comment 6 for response.

9. No further comment.

Response: Thanks for the positive comment.

10. Stability of an agent would typically be tested with HPLC. However, the photostability may suffice here.

Response: Thanks for the comments. We have investigated the stability of LZ-1105 by HPLC in blood and urine.

LZ-1105 were analyzed by reversed-phase high performance liquid chromatographic (RP-HPLC) using eluent A and eluent B (Eluent A: H₂O containing 0.01% of trifluoroacetic acid; eluent B: methanol. A / B = 40 / 60 to 20 / 80 for 42 min, 4 ml/min, detect wavelength: 294 nm; samples were filtered by 0.45 μm membrane). An elution peak at 31.5 min corresponding to free LZ-1105 was observed in RP-HPLC trace of the solution. Blood sample was acquired at 4 h after LZ-1105 administration. Methanol (200 μl) was added into plasma (50 μl) to precipitate the proteins. The mixture was vortexed and subsequently centrifuged at 12000 rpm for 10 min. The supernatant was mixed with an equal volume of deionized water and subjected to RP-HPLC. An elution peak at 31.5 min was observed in RP-HPLC trace of the treated blood sample, illustrating the good stability of LZ-1105 in blood. Urine same was collected within 5 h after LZ-1105 administration.

An elution peak at 31.5 min was observed in RP-HPLC trace of the treated urine sample, demonstrating the good stability of LZ-1105 in urine (Supplementary Fig. 33 in revised Supplementary information). These results indicated the stability and renal clearance of LZ-1105 in vivo.

Supplementary Fig. 33. RP-HPLC results of LZ-1105, treated blood samples from control and LZ-1105 injected mice, treated urine samples from control and LZ-1105 injected mice.

In addition, we added the related description on Page 8 in the revised manuscript as follows:
“Meanwhile, stability of LZ-1105 in blood and urine was illustrated by reversed-phase high performance liquid chromatographic (RP-HPLC) (Supplementary Fig. 33).”

11. No further comment.

Response: Thanks for the positive comments.

12. See response to comment 10.

Response: Please see comment 10 for response.

13, 14 No further comment

Response: Thanks for the positive comments.

15. See comment 6

Response: Please see comment 6 for response.

Other comments:

1. Figure R2-1 Its hard to imagine that 30 minutes of 70 mW/cm² of 1064 nm light causes no heating of tissues. Please check this.

Response: Thanks for the comments. We have repeated the photothermal experiment and same results were obtained.

Meanwhile, previous work has also illustrated the low photothermal effect of 1064 nm (Cite from: Lin, H., Gao, S., Dai, C., Chen, Y. & Shi, J. A Two-Dimensional Biodegradable Niobium Carbide (MXene) for Photothermal Tumor Eradication in NIR-I and NIR-II Biowindows. Journal of the American Chemical Society 139, 16235-16247 (2017). Temperature increased only 5 °C during 10 min irradiation at power density of 1W/cm²).

Figure R3. Temperature elevations at the tumor sites of 4T1-tumor-bearing mice in groups of NIR-I, NIR-II (1064 nm, 1W/cm²), Nb₂C-PVP + NIR-I, and Nb₂C-PVP + NIR-II during laser irradiation. (Cite from: Lin, H., Gao, S., Dai, C., Chen, Y. & Shi, J. A Two-Dimensional Biodegradable Niobium Carbide (MXene) for Photothermal Tumor Eradication in NIR-I and NIR-II Biowindows. *Journal of the American Chemical Society* 139, 16235-16247 (2017))

2. The authors write that the NIRvana camera is a CCD. It is not. It is a focal plane array without the read out architecture of a CCD.

Response: Thanks for the suggestion. We have revised “InGaAs CCD” into “InGaAs camera” in the revised manuscript and supplementary information.

Reviewer #2 (Remarks to the Author):

Comments were well addressed and the revised version is suitable for publishing in *Nature Communications*.

Response: Thanks for the positive comments.

Reviewers' comments:

Reviewer #1 (Remarks to the Author):

SUMMARY: This is the third version of a manuscript which has some interesting features to it, but has inaccuracies that mislead readers not specialized in optical imaging. While the authors have substantively revised and improved the manuscript over two cycles with corrections (and most notably profound changes in conclusions on “free” LZ-1105), the rebuttals also argue key points non-succinctly. The following summarizes my continued reasons for recommending rejection. The authors can use these comments to succinctly revise their contribution and submit elsewhere.

1. Signal to noise (SNR) and contrast (C) is a function of both a device and contrast agent. Contrast and signal to background (or target to background) are related. The target to background or signal to background is intensity of target divided by the intensity of background. Contrast = (intensity of target)/ (intensity of background) – 1, hence Contrast = SBR – 1. These are definitions for medical imaging.

The signal to noise ratio is the difference in intensities between target and background divided by the standard deviation in the background. The authors should be focusing on SNR and C since SBR and C are functions of each other. The definition which the authors used is incorrect, regardless whether they are replicating an incorrect statement. Replicating an incorrect statement from PNAS does not make it correct.

It is noteworthy that the authors provide TWO different definitions of contrast in their reply. The first definition they provide is correct.

2. The authors responded strongly to my overall summary. If “fluorescence imaging needs to be acquired with the highest imaging quality (high resolution, high signal to noise),” then (1) SNR (not SBR) needs to be computed, and ICG should be imaged with the optimal wavelengths and detectors! The authors may wish to read PMID:31478842, which shows that they are comparing their LZ-1105 with a sub-optimal imaging of ICG (i.e., without the highest SNR possible!). NIRF-II imaging of ICG has lower SNR and is not as promising for medical applications as NIRF imaging of ICG (which is already in the clinic).

3. SWIR or NIRF-II is limited to mm, not cm as contained in page 3, line 58. NIRF-I can penetrate 2-4 cm (with optimized devices), but NIRF-II cannot go beyond a few mm. My comment on this aspect of poor penetration in the past two reviews was ignored.

4. The discussion of vascular dysfunction, amputation, and hemorrhage in the introduction is mute, since as described in the Discussion, NIRF-II is good for animal studies and does not have the penetration depth to look at these clinical scenarios. My comment on this aspect of poor penetration in the past two reviews was ignored.

5. Page 4, line 75. "Resolution" is not a function of the imaging agent, but rather the device. The statement made is incorrect and misleading.

6. Contrast and SNR is a function of the imaging device. Given that ICG was interrogated with 808 nm excitation and 1300 nm longpass (which collected the weakest part of the emission spectra) and LZ-1105 was collected with 1064 nm excitation and 1400 nm longpass. Hence these are different systems. The NIRF-II imaging of ICG is very poor (because of the way it is collected) and in accordance with the author's response to my overall summary, the comparison should have been made with NIRF imaging of ICG (which is in clinical and preclinical use!)

7. I am not at all convinced by the changes of the "free" LZ-1105 in the vasculature (first manuscript), and the "fibrinogen-bound" LZ-1105 in the vasculature to address my comment on permeability (second manuscript). In the third manuscript, the authors did not respond to my comment that fibrinogen is secreted by the liver and therefore the liver should be bright when their data shows it is not. Why fibrinogen and not some other entity is being bound to LZ-1105, is not known. Is it taken up in a cellular component? I think that the analytical tests and lack of blood analyses makes for an unconvincing argument. Its almost better to state that it remains unknown.

8. I did not see that my request for data evidencing that LZ-1105 is planar was responded to.

Reviewer #2 (Remarks to the Author):

I have carefully read the point-to-point response, the Review 1's third round comments as well as the reference mentioned by Reviewer 1 (PMID:31478842). Frankly speaking, I am surprised the

Reviewer 1 is still tangled with the author about the well-demonstrated results that whether NIR-II of ICG is superior to the NIR-I. I have an address in my last email. In fact, this point has already been investigated clearly before (Proc. Natl. Acad. Sci. USA 115, 4465-4470 (2018); Nat. Mater. 15, 235–242 (2016); Nat. Biomed. Eng. doi:10.1038/s41551-019-0494-0 (2019)). It has been demonstrated that even though the main emission spectra peak of ICG was 820 nm, the in-vivo imaging resolution and contrast beyond 1300 nm were higher than that of 850-, 1000-, 1100- and 1200 nm (Proc. Natl. Acad. Sci. USA 2018, 115, 4465; Nat. Biomed. Eng. doi:10.1038/s41551-019-0494-0 (2019)). Overall, regarding the controversy on the imaging of ICG, I still think the authors have addressed all of the comments properly in the last two round revision process. They don't have to revise this point again. For the metabolism of LZ-1105 (reviewer #1, point 7), the reviewer 1 just mentioned that "I am not at all convinced by the changes of the "free" LZ-1105 in the vasculature (first manuscript), and the "fibrinogen-bound" LZ-1105 in the vasculature". However, this reviewer did not present a detailed reason to suspect the authors' results. In my opinion, the metabolism and "fibrinogen-bound" results are supported by solid experimental data. I also think the authors don't have to revise this point again. I strongly to recommend to accept it.

I would like to go through the Reviewer 1's comments point-to-point as follows.

1. Firstly, I'd like to approve using SBR rather than SNR in the in vivo imaging. Because the definition of SNR only considers the scattering and absorption effect of excitation wavelength while ignores the scattering and absorption effect of tissue during the emission process. SBR could reflect the overall imaging quality. Secondly, Review 1 declares that Contrast = (intensity of target)/ (intensity of background) – 1 (which is from paper PMID: 31478842), and he/she agrees with the first definition in the author's point-to-point response, where contrast equals variation of image signal divided by the average intensity. However, these two definitions are different. This makes me confused. Moreover, two contrast definitions the author provided were the same. Thus I think the parameters the authors used are correct. They don't have to revise this point again.

2. I cannot agree with the reviewer 1 about the "ICG should be imaged with the optimal wavelengths and detectors!". For the optimal wavelength selection, I have an address in my last email. In fact, this point has already been investigated clearly before (Proc. Natl. Acad. Sci. USA 115, 4465-4470 (2018); Nat. Mater. 15, 235–242 (2016).). It has been demonstrated that that even though the main emission spectra peak of ICG was 820 nm, the in-vivo imaging resolution and contrast beyond 1300 nm were higher than that of 850-, 1000-, 1100- and 1200 nm (Proc. Natl. Acad. Sci. USA 2018, 115, 4465; Nat. Biomed. Eng. doi:10.1038/s41551-019-0494-0 (2019)). It's unnecessary to tangle with the author again about this well-demonstrated results for Reviewer 1. Furthermore, I can agree with Reviewer1 about the "optimal detectors" selection. There are different kind of detectors from a different company with a different model. It's very difficult to ask all of the authors to use the best and optimal detectors for every experiment. Furthermore, as I mentioned above, SBR (not SNR) should be used to evaluate imaging quality. Moreover, in paper PMID: 31478842, a solid tissue consist of polyurethane, TiO₂, and ICG was used to mimic tissue absorption/scattering and evaluate

imaging quality. This is not reasonable because the molecular state of ICG could be influenced in such solid. I'd like to approve using Intralipid as phantom tissue (which was used in this manuscript and most of the previous reports about the NIR-II bioimaging). Besides, all data were extracted from experiments out of living animals, which is not convincing. Moreover, after I read the reference suggested by the Reviewer1 (PMID: 31478842), I think this paper doesn't support his comments either for this manuscript. Such as the statement in the discussion part "The advantages of SWIR, namely reduced tissue scattering, a large Stokes shift that minimizes contributions from autofluorescence and excitation light leakage, and the larger bandwidth for fluorescent collection bode well for its deployment in biomedical optical imaging." These statements in the paper clearly demonstrated that NIR-II bioimaging is superior to the NIR-I. Thus, I think these comments don't have to be replied anymore.

3. On the premise of sacrificing resolution, tissue penetration of NIR-II bioimaging could also reach cm level (Angew. Chem. Int. Ed. 2014, 53, 12086). However, imaging without resolution is meaningless, Reviewer 1 continue to ignore the key point that tissue penetration should be considered with resolution together. Since this work is focused on the blood vessel dynamic imaging, the high resolution for the blood vessel is very important during considering the penetration depth. I think the authors' statements for tissue penetration detection are correct. This comment doesn't have to be replied anymore.

4. I think the author has responded to these comments before. They also added the related explanation in the revised introduction and discussion part before. In my opinion, ICG is a commercialized probe that has been reported a lot for the clinical application. However, the progress of science still needs more effort to design and discover more new matters. This study is attractive because of the novel hydrophilic small-molecule NIR-II organic dyes with long-term blood circulation features for the dynamic vascular imaging. The results are supported by solid experimental data. LZ-1105 is a good probe for fundamental researcher and pre-clinical research before the clinical permission. This comment doesn't have to be replied anymore.

5. Reviewer 1 may confuse the relationship between device resolving ability and bioimaging resolution. Device resolving ability is only related to the device while the imaging resolution is affected by both device and contrast agent. Actually, after I read the reference suggested by the Reviewer1 (PMID: 31478842), I think this paper doesn't support his comments for this manuscript. Such as the statement in the discussion part "Because this study was confined to the three separate systems studied, we would like to emphasize that we do not broadly conclude that NIRF imaging is advantageous over SWIR fluorescence imaging of ICG or any other potential fluorescent contrast agent that could be translated clinically for molecularly targeted detection of diseased tissues." Therefore, I think these comments don't have to be replied anymore.

6. In fact, in contrary to what the reviewer declares, research using ICG for liver tumor detection in NIR-I and NIR-II region showed intraoperative NIR-II imaging provided a higher tumor-detection sensitivity compared with NIR-I imaging (Nat. Biomed. Eng. 2019, doi.org/10.1038/s41551-019-0494-0). It's unnecessary to tangle with the author about this well-demonstrated results for Reviewer 1. This comment doesn't have to be replied anymore.

7. For the metabolism of LZ-1105 (reviewer #1, point 7), reviewer #1 just mentioned that "I am not at all convinced by the changes of the "free" LZ-1105 in the vasculature (first manuscript), and the "fibrinogen-bound" LZ-1105 in the vasculature". However, this reviewer did not present a detailed reason to suspect the authors' results. Actually, the author has provided solid data to illustrate the interaction between LZ-1105 and fibrinogen in the point-to-point response. In fact, some molecule has been reported to bind with protein, such as CH-4T and HSA (Nat. Commun. 2017, 8, 15269), ICG and HSA (Proc. Natl. Acad. Sci. USA 2018, 115, 4465). The interaction could be demonstrated by measuring the fluorescence of dye before and after mixing with protein. Of course, it would have been helpful if the authors would figure out why the liver is dim, but this is not the key point of this work, and the author may explore this in their future work. Thus, these comments doesn't have to be replied anymore.

8. Reviewer 1 requested for data to demonstrate the planar of LZ-1105, however, this could only be demonstrated by a single-crystal X-ray diffraction technique, which is extremely hard for LZ-1105. Thus, these comments doesn't have to be replied anymore.

Therefore, here are my opinions about the main controversy:

(1) The imaging collection of ICG and LZ-1105: I believe the collection range for ICG in this manuscript is correct, which is also supported by the solid data and other contributions.

(2) The penetration depth of imaging in NIR and SWIR region (in other words, NIR-I and NIR-II window): On the premise of sacrificing resolution, tissue penetration could reach cm level. However, imaging without resolution is meaningless, Reviewer 1 continue to ignore this.

(3) Definition of contrast, SBR, SNR, and which parameter should the author use to evaluate the imaging quality: SBR should be used for evaluating imaging quality, and the definition of contrast the author used in living animals was correct.

(4) The interaction between LZ-1105 and fibrinogen, and the planar of LZ-1105: the author has provided solid data to illustrate the interaction between LZ-1105 and fibrinogen. The requirement for the planar of LZ-1105 data is beyond the experimental operation because it's impossible to obtain the single crystal samples for the LZ-1105 organic molecule.

Reviewer #3 (Remarks to the Author):

The authors provided the abundant data to address the reviewer's concerns. However, some issues need to be further investigated carefully.

1. The authors claimed the LZ-1105 could be bound to the fibrinogen resulting in prolonged blood retention. However, the bovine fibrinogen rather than the mouse fibrinogen was employed for this assay, which greatly compromised the accuracy. For the better understanding potential mechanism of its long circulation, the multi-component analysis should be executed including blood cells and the plasma. For blood cell analysis, different types of cells could be separated by centrifugation or FACS. For the plasma analysis, 1D or 2D SDS-PAGE could be executed and further stained with the LZ-1105.

2. About vascular permeability assessment, the authors incubated LZ-1105 with the human embryonic renal epithelial cells (HEK293T), as in the scenario described above, HEK293T may not be the appropriate cell model. The vascular endothelial cells should be taken into account.

3. In order to achieve efficient comparative analysis, the optimized excitation and collection of ICG and LZ-1105 should not be ignored. The authors should conduct the experiments to compare the classic NIRF imaging of ICG with LZ-1105, and other NIR-II organic dyes could also serve as the controls.

Point-to-point response

Reviewer #3

The authors provided the abundant data to address the reviewer's concerns. However, some issues need to be further investigated carefully.

1. The authors claimed the LZ-1105 could be bound to the fibrinogen resulting in prolonged blood retention. However, the bovine fibrinogen rather than the mouse fibrinogen was employed for this assay, which greatly compromised the accuracy. For the better understanding potential mechanism of its long circulation, the multi-component analysis should be executed including blood cells and the plasma. For blood cell analysis, different types of cells could be separated by centrifugation or FACS. For the plasma analysis, 1D or 2D SDS-PAGE could be executed and further stained with the LZ-1105.

Response: Thanks for the useful comments. For blood cell analysis, different types of cells including red blood cells (RBC), platelets, neutrophils and lymphocytes were collected by density gradient centrifugation from the mouse whole blood. After incubating with LZ-1105, no NIR-II signal enhancement were observed for the blood cells under 1064 nm excitation, indicating that there is weak interaction between LZ-1105 and blood cells (Supplementary Fig. 39). To make it clear to the readers, we revised the related description in “*In vivo pharmacokinetics and blood retention*” section on Page 9 in the revised manuscript as follow: “Besides, LZ-1105 was further demonstrated to have little interaction with HUVEC and several kinds of mouse blood cells including red blood cells (RBC), platelets, neutrophils and lymphocytes (Supplementary Fig. 38-39).” And we added Supplementary Fig. 39 in the revised supplementary information.

Supplementary Fig. 39. (a) White light photos of mouse whole blood and blood cells before and after adding LZ-1105. (b) NIR-II imaging results of LZ-1105 in mouse whole blood, RBC, platelets, neutrophils, and lymphocytes. (1064 nm excitation, 30 mW/cm², 1400 nm long-pass filter). Insets: NIR-II fluorescence images of LZ-1105 in different media. ([LZ-1105] = 10 μM).

Furthermore, Sodium dodecyl sulfate polyacrylamide gel electrophoresis (SDS-PAGE) were hired for plasma analysis. As shown in Supplementary Fig. 40, plasma-LZ-1105 complex exhibited abundant protein bands. However, only fibrinogen related bands could be observed under InGaAs NIR-II camera, indicating the NIR-II signal enhancement of LZ-1105 in plasma was mainly due to the interaction between fibrinogen and LZ-1105 (the protein bands observed under InGaAs camera were in consistent with the previous report: Zauner, G. et al. Glycoproteomic Analysis of Human Fibrinogen Reveals Novel Regions of O-Glycosylation. *J. Proteome Res.* **11**, 5804-5814 (2012). This reference has been added in the revised Supplementary information as Ref 3). To make it clear to the reviewer and the readers, we added the related description in “*In vivo pharmacokinetics and blood retention*” section on Page 9 in the revised manuscript as follow: “Moreover, sodium dodecyl sulfate polyacrylamide gel electrophoresis (SDS-PAGE) were hired for plasma analysis. The electrophoresis testing exhibited that only fibrinogen related proteins could be observed under NIR-II InGaAs camera, demonstrating the NIR-II signal enhancement of LZ-1105 was mainly due to the interaction between fibrinogen and LZ-1105 (Supplementary Fig. 40).” And we added Supplementary Fig. 40 in the revised supplementary information.

Supplementary Fig. 40. Sodium dodecyl sulfate polyacrylamide gel electrophoresis (SDS-PAGE) results. Electrophoresis gel analysis of LZ-1105-plasma complexes and NIR-II imaging of the electrophoresis gel. Concentration of LZ-1105: 10 μM. Power density of 1064 nm laser: 30 mW cm⁻². NIR-II images were acquired by 1400 long-pass filter.

2. About vascular permeability assessment, the authors incubated LZ-1105 with the human embryonic renal epithelial cells (HEK293T), as in the scenario described above, HEK293T may not be the appropriate cell model. The vascular endothelial cells should be taken into account.

Response: Thanks for the useful comments. According to the reviewer's suggestion, we have tried human umbilical vein endothelial cells (HUVEC) to incubate with LZ-1105 instead of HEK293T. As shown in Supplementary Fig. 38, after incubating with LZ-1105, there was not NIR-II signal intensity enhancement for HUVEC suspension, which exclude the interaction between LZ-1105 and vascular endothelial cells.

Supplementary Fig. 38. Optical characterization of LZ-1105 in different media. The fluorescence intensity of LZ-1105 was detected by InGaAs camera in different media including fresh mice blood with or without plasmin, saline (0.9% NaCl solution), human umbilical vein endothelial cells (HUVEC) suspension and plasmin solution (1064 nm excitation, 30 mW/cm², 1400 nm long-pass filter). Insets: NIR-II fluorescence images of LZ-1105 in different media. ([LZ-1105] = 10 μM)

To make it clear to the readers, we revised the related description in “*In vivo pharmacokinetics and blood retention*” section on Page 9 in the revised manuscript as follow: “Besides, LZ-1105 was further demonstrated to have little interaction with human umbilical vein endothelial cells (HUVEC) and several kinds of blood cells including red blood cells (RBC), platelets, neutrophils and lymphocytes (Supplementary Fig. 38-39).”

In addition, we also used HUVEC to measure the cell viability of LZ-1105 instead of HEK293T. After 12 h and 24 h incubation with LZ-1105 under different concentrations, cell viabilities were above 95 % even with concentration of 350 μM , indicating the low cytotoxicity (Supplementary Fig. 24). To make it clear to the readers, we revised the related description in “*In vivo pharmacokinetics and blood retention*” section on Page 5 in the revised manuscript as follow: “The potential cytotoxicity of LZ-1105 was evaluated in human umbilical vein endothelial cells (HUVEC). The cells exhibited over 95 % viability after incubation with 350 μM of LZ-1105 for 24 h (Supplementary Fig.24).”

Supplementary Fig.24. Potential cytotoxicity of LZ-1105. Cell viability of human umbilical vein endothelial cells (HUVEC) after 12 h and 24 h incubation with LZ-1105 under different concentrations. The cell viabilities were above 95 % even with concentration of 350 μM . Error bars, mean \pm s.d. (n = 3).

REVIEWERS' COMMENTS:

Reviewer #2 (Remarks to the Author):

The manuscript contains a fairly complete set of experiments supporting their results. The comments have been well addressed and the revised version is suitable for publishing in Nature Communications.

All of the comments can be concluded into two points:

- 1) Whether the present molecule (LZ-1105) offers a technological advantage over ICG.
- 2) Whether the reason for long-term blood circulation time is the binding with fibrinogen.

For the first points, the comments were fully addressed in the last two round revision process.

For the second points, it has been further intensified by addressing the Reviewer #3's comments. Through blood cell multi-component analysis and vascular permeability assessment, the authors have demonstrated that the molecule LZ-1105 could not bind to blood cells or vascular endothelial cells. Furthermore, through plasma analysis (SDS-PADE), the authors illustrated that molecule LZ-1105 could only bind to fibrinogen.

Overall, in their revised manuscript, the authors have provided convincing evidence to address all of the comments. I strongly recommend to accept it for publication in Nature Communications.